# Enhancing the fairness of AI prediction models by Quasi-Pareto improvement among heterogeneous thyroid nodule population

Siqiong Yao[1,2,5], Fang Dai[1,5], Peng Sun[1], Weituo Zhang[3] ✉, Biyun Qian [3] ✉ & Hui Lu [1,2,4] ✉

Artificial Intelligence (AI) models for medical diagnosis often face challenges of generalizability and fairness. We highlighted the algorithmic unfairness in a large thyroid ultrasound dataset with significant diagnostic performance disparities across subgroups linked causally to sample size imbalances. To address this, we introduced the Quasi-Pareto Improvement (QPI) approach and a deep learning implementation (QP-Net) combining multi-task learning and domain adaptation to improve model performance among disadvantaged subgroups without compromising overall population performance. On the thyroid ultrasound dataset, our method significantly mitigated the area under curve (AUC) disparity for three less-prevalent subgroups by 0.213, 0.112, and 0.173 while maintaining the AUC for dominant subgroups; we also further confirmed the generalizability of our approach on two public datasets: the ISIC2019 skin disease dataset and the CheXpert chest radiograph dataset. Here we show the QPI approach to be widely applicable in promoting AI for equitable healthcare outcomes.

AI has made substantial progress as a clinical diagnostic tool for a variety of diseases[1–3]. Previous studies primarily reported results on the overall population, but there is a growing concern regarding the heterogeneity and fairness of AI models across different subgroups[4–8]. Researchers often evaluate model performance based on the general population or prevalent disease subtypes, neglecting rare or specific subgroups[9]. For instance, in lung cancer research, a typical deep learning study might categorize lung cancer into the predominant LUAD and LUSC subtypes and normal lung tissue, neglecting the

identification of rare subtypes[10]. This oversight is also evident in ultrasound image diagnoses of breast cancer, leading to a high rate of false negatives and consequently, missed detections[11]. Specifically, among the 48 papers published on thyroid nodule benign-malignant diagnosis with AI models in the past 5 years, 44 reported only the prediction AUC for the overall population or the most common subtype Papillary Thyroid Carcinoma (PTC, 85–95% of incidence)[12–16], whereas only four papers discussed the diagnostic outcomes for less-prevalent subtypes, including Follicular Thyroid Carcinoma (FTC,

[1]State Key Laboratory of Microbial Metabolism, Joint International Research Laboratory of Metabolic and Developmental Sciences, School of Life Sciences and Biotechnology, Shanghai Jiao Tong University, Shanghai 200240, PR China. [2]SJTU-Yale Joint Center of Biostatistics and Data Science, National Center for Translational Medicine, MoE Key Lab of Artificial Intelligence, AI Institute Shanghai Jiao Tong University, Shanghai 200240, PR China. [3]Hongqiao International Institute of Medicine, Shanghai Tong Ren Hospital and School of Public Health, Shanghai Jiao Tong University School of Medicine, Shanghai 200336, PR China. [4]Shanghai Engineering Research Center for Big Data in Pediatric Precision Medicine, NHC Key Laboratory of Medical Embryogenesis and Developmental Molecular Biology & Shanghai Key Laboratory of Embryo and Reproduction Engineering, Shanghai 200020, PR China. [5]These authors contributed equally: Siqiong Yao, Fang Dai. ✉e-mail: zhangweituo@sjtu.edu.cn; qianbiyun@sjtu.edu.cn; huilu@sjtu.edu.cn

3–5% of incidence), Medullary Thyroid Carcinoma (MTC, 1–3% of incidence), and Anaplastic Thyroid Carcinoma (ATC, <1% of incidence), etc. Given sample size imbalance, AI models might be subject to unfairness by omitting features of the less-prevalent subgroups. This may lead to misdiagnosis that negatively impacts the health outcomes of patients from these groups[17–19]. Current medical AI research, however, doesn't provide ample quantitative evaluations of these biases. While some investigations have noted these biases, causality-based unfairness analyses and exploration of potential mitigation strategies leave much to be desired[20,21].

As illustrated in Fig. 1, existing common approaches in handling less-prevalent subtypes include Mixed training, also known as standard empirical risk minimization (standard ERM), and Divide-and-conquer, also known as stratified ERM[22]. Mixed training approaches combine data from all subtypes to train an AI model, with the potential risk that the model might overlook less-prevalent subgroup features. In contrast, Divide-and-conquer approaches utilize data from different subtypes as distinct training sets and train models separately for each subtype. However, less-prevalent subgroups are typically of insufficient sample sizes, that may not be able to support the training of AI models. Conversely, many algorithms are designed to prioritize prediction accuracy for underperforming subgroups, often without adequately considering the implications for the overall model's performance[23,24]. This oversight can lead to complications in real-world clinical applications[25]. Correspondingly, we empirically highlighted the algorithmic unfairness of AI model on thyroid cancer diagnosis in minorities (Fig. 2). We observed that regardless of whether we used Mixed training or divide-and-conquer strategies on our thyroid nodule ultrasound image dataset, models generally performed worse in less-prevalent subgroups compared to the dominant subgroup. Particularly, when considering histological subtypes, we observed significant model performance disparity in FTC and MTC (less-prevalent subgroup), compared with PTC (dominant subgroup). Furthermore, we demonstrated that the observed disparity is attributed to the sample size imbalance via Pareto criterion, a learning curve-based experiment (Methods section: Measurement of Fairness in Medical AI Diagnosis).

To address the shortcomings of Mixed training and Divide-and-conquer approaches in terms of fairness, we propose a Quasi-Pareto Improvement (QPI) approach (Fig. 1c), aiming to enhance the prediction performance of less-prevalent subgroups to help them achieve equitable benefits while retaining the model's prediction performance on the overall population. We designed a deep learning framework Quasi-Pareto Net (QP-Net) to implement the proposed QPI approach by employing state-of-the-art multi-task learning techniques (e.g., adaptive-weight[26]) and transfer learning techniques including adversarial structures[27], Maximum Mean Discrepancy (MMD)[28], and Batch Spectral Shrinkage (BSS) regularization[29]. The effectiveness of the devised QP-Net was validated on our thyroid dataset and illustrated using Centered Kernel Alignment (CKA) method. From a broader perspective, sample imbalance is prevalent in medical datasets, either due to medical limitations or data collection bias[23,30–32]. Previous research has identified that assessment of model unfairness is critical in any machine learning deployment for medical imaging[33]. Consequently, we evaluated the effectiveness of the devised QP-Net on two large public datasets, including ISIC2019[34] and CheXpert[35], where we observed undergeneralization and prediction unfairness of imbalanced samples in several subgrouping schemes, to support the generalizability of our QPI approach. In both public datasets, the QPI and QP-Net have demonstrated significant effectiveness. Quasi-Pareto Improvement can be applied broadly to address the insufficiency and inequity of prediction generalization in imbalanced medical subgroups, thereby aiding in the understanding of subgroup disease mechanisms and accelerating the application of medical AI.

## Results
### Clinical data of subgroups unfairness
We collected 360,455 thyroid ultrasound images from 123,301 patients over the course of nearly a decade at nine top-tier hospitals and one community hospital in China for model training and validation. All ultrasound data were annotated by specialists with more than 5 years of experience and included comprehensive ultrasound reports and aspiration biopsy pathology reports. Various subgrouping schemes can be applied to our dataset. For histological subtypes, PTC accounted for 95.9%, FTC for 3.3%, and MTC for 0.8%; for age, patient age ranged from 0 to 85 years, with the 35–60 age group being the largest at 48%; for gender, the male-to-female ratio was 0.49; for nodule sizes, they ranged from 0.3 cm to 1.3 cm, with the <1 cm group accounting for 58%. For hospitals, the tertiary-to-community ratio was 4.2.

The results of Mixed training (resp. Divide-and-conquer) are illustrated in Fig. 2a, c (resp. Fig. 2b). Figure 2a showed significant AUC disparity across histological subtypes and hospitals. In particular, the AUC of PTC was over 46% and 30% higher than that of FTC and MTC, and the AUC of tertiary hospitals was over 42% higher than that of community hospitals. Figure 2c shows learning curves of Mixed training approaches. With the increase in sample size for PTC and tertiary hospitals (dominant subgroup), the prediction performance of

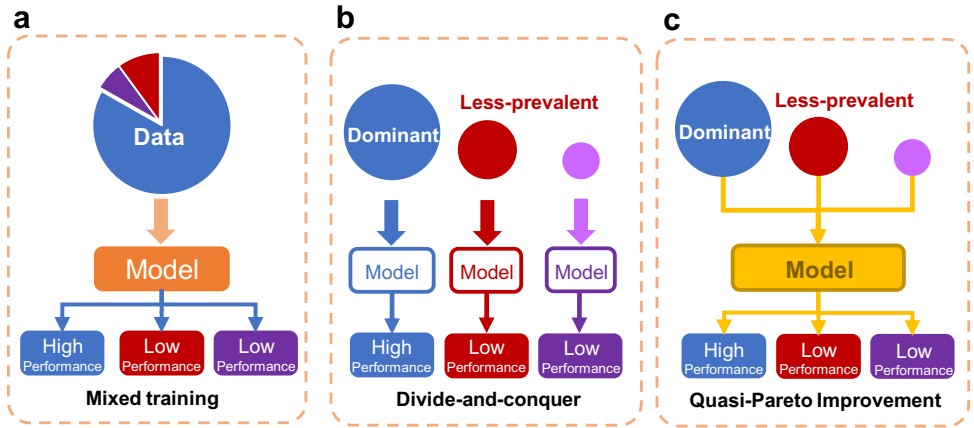

**Fig. 1 | Approaches to handle less-prevalent subgroup. a** Mixed training. Training on the mixed data with multiple subgroups. **b** Divide-and-conquer. Separate training within subgroups. **c** QPI approach. Improving the prediction performance of imbalanced subgroups while maintaining model prediction performance on the overall population.

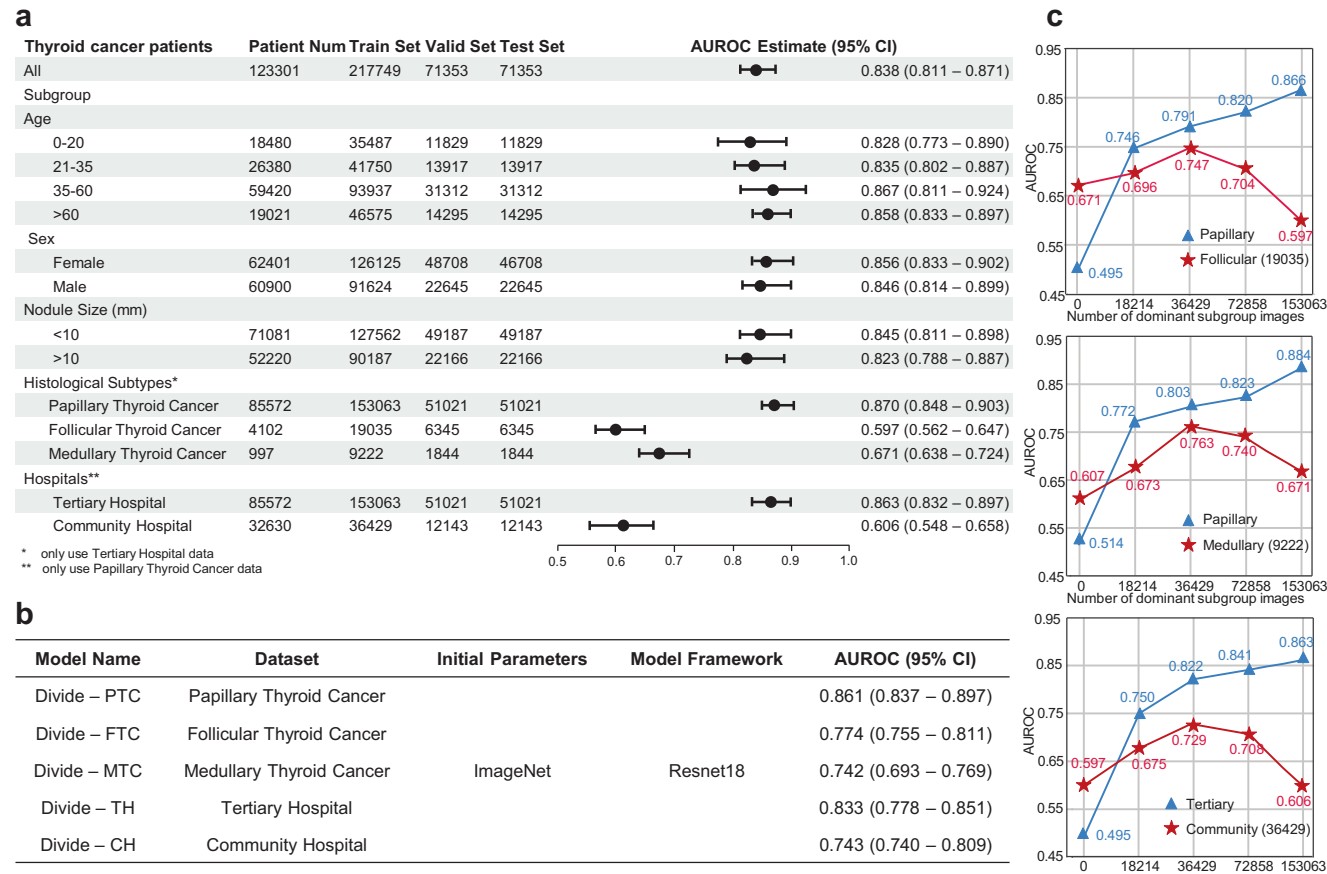

**Fig. 2 | Benign-malignant prediction performances of Mixed training and Divide-and-conquer. a** Prediction performance of Mixed training model on different subgroups. Note that 'Patient Num' column refers to the number of patients while 'Train Set', 'Valid Set' and 'Test Set' columns refer to number of image samples used in experiments. The error bars represent the 95% confidence interval (95% CI) of the AUROC estimate, centered around the mean value. Notably, there were significant differences in prediction performance between histological subtypes and data source subgroups. **b** Performance of Divide-and-conquer models which is performance, with lower prediction performance for the less-prevalent subgroup. **c** Learning curve of Mixed training strategy. For imbalanced subgroups, increasing the sample size of the dominant subgroup resulted in an initial increase followed by a consequential decrease in the prediction performance on the sample-size-fixed less-prevalent subgroup (star), which contravenes the Pareto fairness criterion. Source data are provided as a Source Data file.

FTC, MTC, and community hospitals (less-prevalent subgroup) exhibited an initial improvement followed by a significant decline. This indicated that the dataset of the dominant subgroup has both positive and negative effects on the model prediction of less-prevalent subgroup, but crossing certain threshold severe unfairness manifests and less-prevalent subgroup prediction performance declines. The "Divide-and-conquer" methods, as depicted in Fig. 2b, are notably superior to Mixed training methods, yet disparities persist between dominant subgroup and less-prevalent subgroups. The low predictive AUC for less-prevalent subgroups in Fig. 2b suggests a lack of sufficient data.

**Quasi-Pareto improvement approach**

We propose a QPI approach and designed QP-Net as an implementation of QPI. As shown in Fig. 3, the QP-Net incorporates a two-module structure: a multi-task learning module and a domain adaptation module. The multi-task learning module maintains model performance on general population during training, while the domain adaptation module aligns the distributions of extracted features across two subgroups to improve the network's less-prevalent subgroup feature fitting in which different subgroups are treated as different domains. The training procedure minimized the loss of both class label (benign-malignant) and subgroup (dominant subgroup - less-prevalent subgroup) predictors within themselves, ensuring their own effectiveness while attempting to learn a feature extractor that generates

subgroup-invariant features. The overall loss function is represented by the following equation:

$$\mathscr{L}(\boldsymbol{\omega}_z, \boldsymbol{\omega}_y, \boldsymbol{\omega}_d) = \mathscr{L}_y(\boldsymbol{\omega}_z, \boldsymbol{\omega}_y) + \mathscr{L}_{DA}(\boldsymbol{\omega}_z, \boldsymbol{\omega}_d) \tag{1}$$

$$\mathscr{L}_y(\boldsymbol{\omega}_z, \boldsymbol{\omega}_y) = \delta \mathscr{L}_{domi}(\boldsymbol{\omega}_z, \boldsymbol{\omega}_y) + (1-\delta)\mathscr{L}_{less}(\boldsymbol{\omega}_z, \boldsymbol{\omega}_y) \tag{2}$$

$$\mathscr{L}_{DA}(\boldsymbol{\omega}_z, \boldsymbol{\omega}_d) = -\gamma_d \mathscr{L}_d(\boldsymbol{\omega}_z, \boldsymbol{\omega}_d) + \gamma_{MMD}\mathscr{L}_{MMD}(\boldsymbol{\omega}_z) + \gamma_{BSS}\mathscr{L}_{BSS}(\boldsymbol{\omega}_z) \tag{3}$$

Where, $\mathscr{L}_y$ denotes the multi-task learning loss, including the benign-malignant class prediction loss for dominant subgroup ($\mathscr{L}_{domi}$) and less-prevalent subgroup ($\mathscr{L}_{less}$) respectively, and $\delta$, adaptively determined through the learning session, adjusts the proportion of dominant subgroup loss to less-prevalent subgroup loss in the overall loss function. $\mathscr{L}_{DA}$ represents the domain adaptation module which is used to enhance model performance on the less-prevalent subgroup. $\mathscr{L}_d$[27] and $\mathscr{L}_{MMD}$[28] are two domain adaptation approaches, and $\mathscr{L}_{BSS}$ is a domain adaptation regularization term, which enables the model to focus more on transferable features[29]. Implementation details are provided in Methods section: QP-Net to implement QPI.

Figure 4 provides a comprehensive comparison of the QPI approach and Mixed training approach, verifying the effectiveness of QPI approach on maintaining medical AI fairness. The experiments were performed for 3 (dominant subgroup, less-prevalent subgroup)

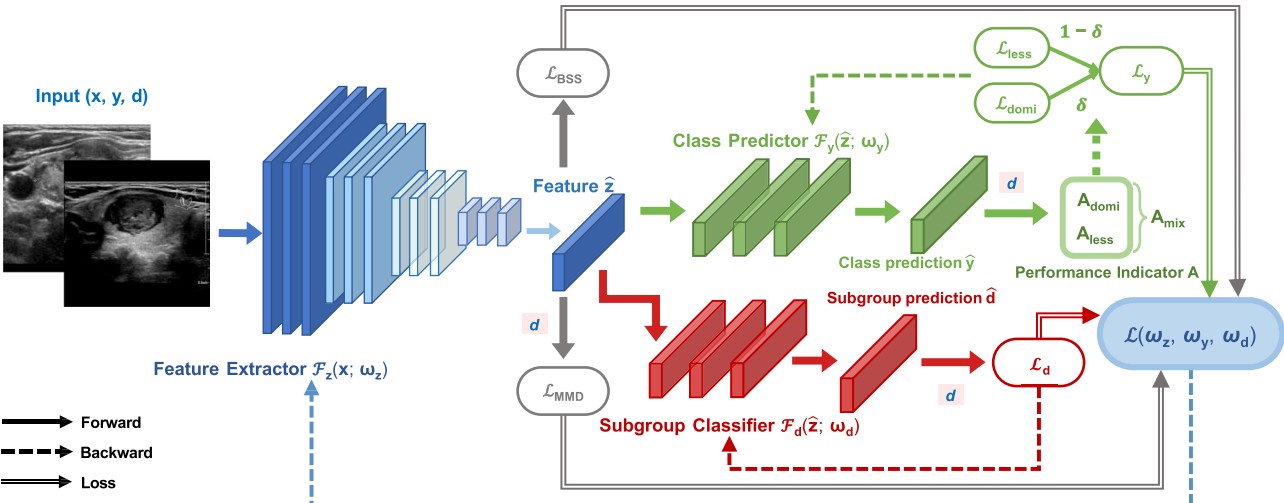

**Fig. 3 | Quasi-Pareto Net diagram.** A typical input includes an image **x**, its corresponding class label $y$ and subgroup $d$. A feature extractor (with parameters $\omega_z$) and a class predictor (with parameters $\omega_y$) form a feedforward network. A multi-task learning structure is incorporated in the class predictor, which retains model performance on the general population. $\mathcal{L}_y$ represents its corresponding loss, including prediction loss for dominant subgroup ($\mathcal{L}_{domi}$) and less-prevalent subgroup ($\mathcal{L}_{less}$), weighted by $\delta$ which is determined by model performance on the two subgroups ($A_{domi}$ and $A_{less}$) through the training session. The domain adaptation module is utilized during the training process to enhance model generalizability, including an adversarial structure (the subgroup classifier with parameters $\omega_d$ whose loss is denoted as $\mathcal{L}_d$), a Maximum Mean Discrepancy loss (denoted as $\mathcal{L}_{MMD}$) which enhances model performance on less-prevalent subgroups, and a regularization term (Batch Spectral Shrinkage loss, denoted as $\mathcal{L}_{BSS}$) which effectively enhances the adaptive ability of the model to address imbalanced sample features while maintaining overall model stability and performance. $\mathcal{L}_y$, $\mathcal{L}_d$, $\mathcal{L}_{MMD}$ and $\mathcal{L}_{BSS}$ form the overall loss $\mathcal{L}$.

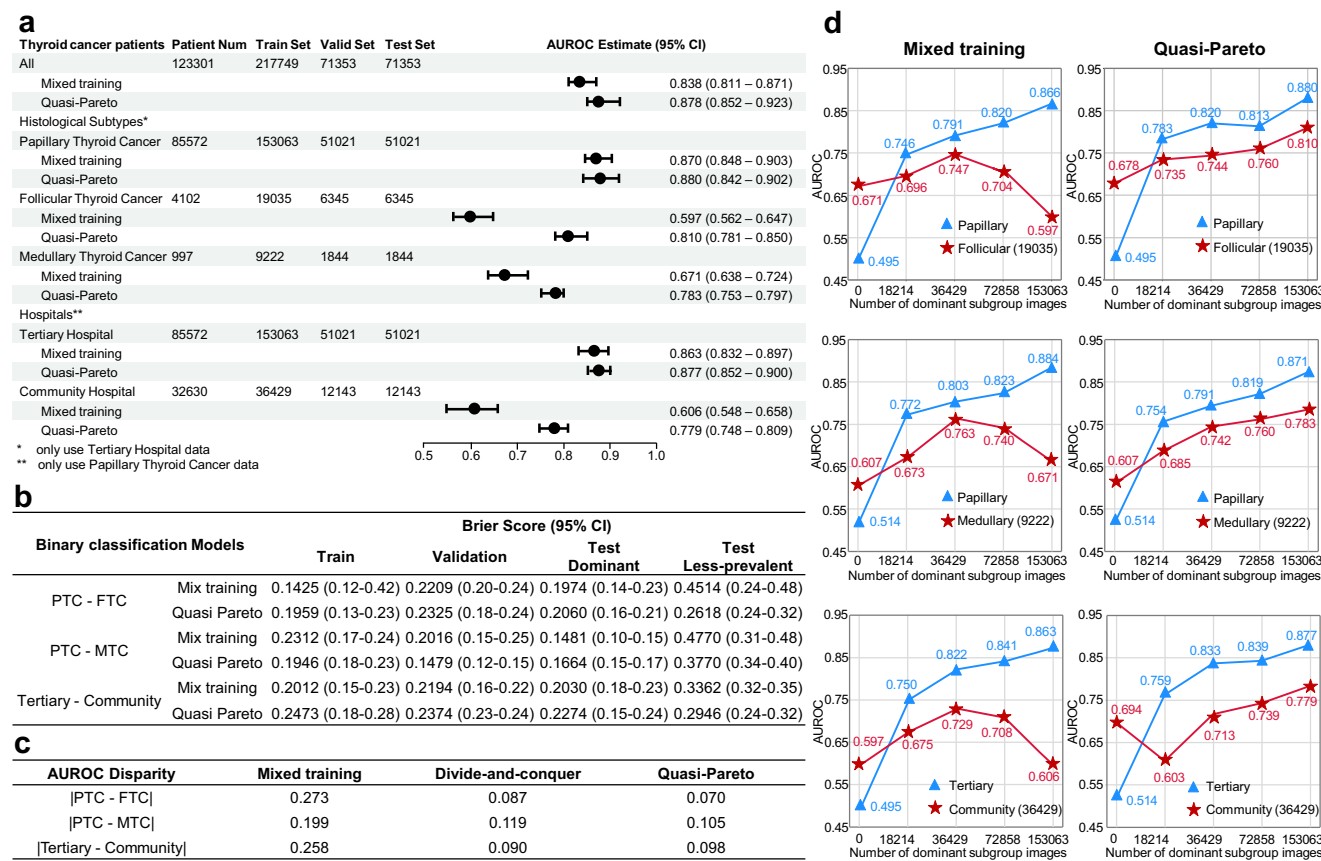

**Fig. 4 | Benign-malignant prediction performances of the QPI approach. a** QPI approach effectively enhances the AUC of model among histological subtypes and source hospital subgroups. Note that the error bars represent the 95% confidence interval (95% CI) of the AUROC estimate, centered around the mean value. **b** QPI approach effectively reduced the model performance disparity in Brier Score among the histological subtypes and source hospital subgroups. **c** QPI approach effectively reduces the model performance disparity in AUC among subgroups. **d** The QPI approach prediction performance for the less-prevalent subgroup continues to rise as the dominant subgroup increases. Source data are provided as a Source Data file.

pairs respectively: PTC vs. FTC, PTC vs. MTC within histological sub-types and Tertiary Hospital vs. Community Hospital within hospitals. Figure 4a showcases the performance of the QPI approach on imbalanced subgroup pairs compared to the Mixed training approach. In particular, the AUC of FTC and MTC were increased by 21.3% and 11.2%, respectively, while that of community hospitals increased by 17.3%. In addition, we observed that the improvement in the performance of the less-prevalent subgroups did not lead to decreases in the prediction performance of the dominant subgroups. Figure 4b displays the Brier Score results for models trained on each subgroup pair. The results indicate that the QPI method notably enhances model calibration for less-prevalent subgroups, with only a marginal increase for dominant subgroups. Figure 4c displays a comparison of the prediction AUC disparity for imbalanced data within histological subtypes and within hospitals, suggesting that the QPI approach mitigates the AUC disparity among imbalanced subgroups. Figure 4d presents the results of learning curve experiments validating whether the prediction performance aligns with the Pareto fairness criterion. It can be readily observed that with QPI approach, the prediction performance of less-prevalent subgroup exhibits consistent improvement as the size of the dominant subgroup increases. In contrast to Mixed training, the performance of QPI approach satisfies the proposed Pareto fairness criterion (Methods section: Measurement of Fairness in Medical AI Diagnosis). It is demonstrated that with QPI approach, the model has the potential to mitigate discrimination against less-prevalent subgroups stemming from conflicting objectives within various subgroups, while simultaneously capitalizing on the transfer of knowledge between these subgroups.

## Layer-wise examination of fairness in neural networks

The learning curve experiment of Mixed training approach revealed both positive and negative effects of the dominant subgroup dataset on less-prevalent subgroup prediction. We employed the CKA method to further locate the dual effect within the deep neural network, especially where the unfairness took place. Following Neyshabur et al. (2020), CKA measures the degree of feature reuse in transfer learning. Comparing two transfer learning strategies (Fig. 5a), ΔCKA can measure the additional contribution of model features learned from the dominant subgroup dataset to less-prevalent subgroup prediction performance. Detailed calculation explanations are located in the Methods section: Centered Kernel Alignment (CKA).

Figure 5b depicts $\Delta CKA_{Mixed}$ and $\Delta CKA_{QPI}$, illustrating the impact of feature reuse on less-prevalent subgroup prediction when employing Mixed training and QPI approach respectively. According to Fig. 5b, $\Delta CKA_{Mixed}$ is positive in middle network layers (roughly layer 7–14) but negative in higher layers (layer 15–17). This illustrates that with Mixed training approach, dominant subgroup dataset has positive effect on less-prevalent subgroup prediction via feature reuse. However, in higher layers of network, the benefit of feature reuse was completely overridden by the negative effect from conflicting features of different populations which significantly discriminated against less-prevalent subgroups under Mixed training. For $\Delta CKA_{QPI}$, The CKA value is non-negative and higher than that of $\Delta CKA_{Mixed}$ in most layers, indicating that QPI approach kept the benefit of knowledge transfer between different populations (feature reuse) till the final layer of neural network.

## External evaluation of QPI approach on public datasets

We observed significant subgroup prediction performance disparity in the following two public datasets and effectively improved model generalization and inequalities using our QPI approach.

CheXpert dataset[35], a large chest radiograph dataset with uncertainty labels and expert comparison. This dataset includes 224,316 chest X-ray images, covering 14 types of chest diseases in 65,240 patients. A DenseNet121 model[36] was trained on CheXpert dataset which was divided into ratios of 60%, 20%, and 20% for training,

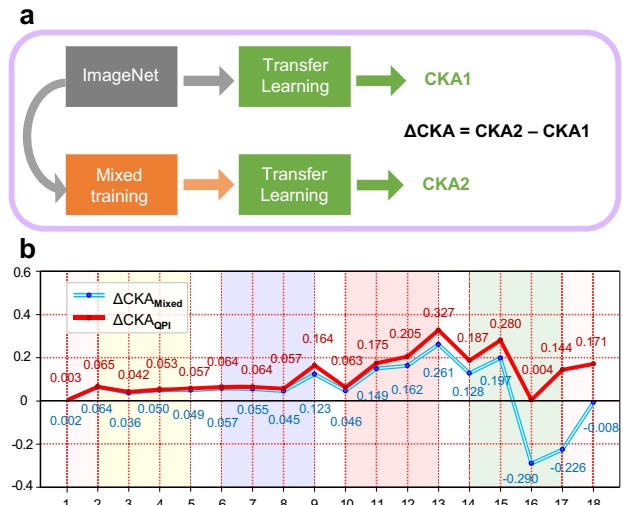

**Fig. 5 | CKA method assesses the effect of dominant subgroup dataset on less-prevalent subgroup prediction. a** CKA1 (resp. CKA2) measures the level of feature reuse in transfer learning from ImageNet pretrained (resp. Mixed training) model to less-prevalent subgroup; ΔCKA measures the additional contribution of dominant subgroup features to less-prevalent subgroup prediction compared with that of ImageNet features. **b** Comparison of ΔCKA_Mixed and ΔCKA_QPI at different neural network layers. For ΔCKA_Mixed (resp. ΔCKA_QPI), the transfer learning step represents training with less-prevalent subgroup samples (resp. via QPI approach). Compared with ΔCKA_Mixed, ΔCKA_QPI is higher and non-negative which demonstrates that the QPI approach reuse more Mixed training features that benefit less-prevalent subgroup prediction task and mitigates the conflicting features in the higher layers. Source data are provided as a Source Data file.

validation and testing. Wherein the samples without training labels were excluded. The experiments were performed for 2 (dominant subgroup, less-prevalent subgroup) pairs (delineated based on the discrepancy in sample sizes) respectively: "18–80" vs. ">80" within age groups, and {White, Asian, Other} vs. {Black} within race groups. Figure 6a shows significant prediction performance differences in both the age and the racial subgroups[37] of female patients. Specifically, the "over 80" age subgroup and the "Female: Black" subgroup contain the fewest observations and theses corresponding prediction AUCs are generally the lowest. After using QPI approach, we can observe varying degrees of AUC development in the prediction performance of these fewest subgroup[38] as shown in Fig. 6b and Fig. 6c.

International Skin Imaging Collaboration 2019 dataset[34]. This dataset includes 25,331 dermoscopic images of nine types of skin diseases (eight diseases and none of the others). An EfficientNets model[39] was trained on this dataset which is divided into the ratio of 60%, 20%, and 20% for training, validation and testing setting. Wherein the samples without training labels were excluded. The experiments were performed for 2 (dominant subgroup, less-prevalent subgroup) pairs respectively: 0–59 vs. 60–85 within age groups, and female vs. male within sex groups. The training set consists of 10,173 male and 5014 female patient samples and Fig. 7a shows significant prediction differences in both the age and the gender subgroups: The prediction AUCs are generally lower in the 60–85 age and the female group. After utilizing QPI approach, we can observe varying degrees of AUC development in the prediction performance of these two fewer subgroup as shown in Fig. 7b and Fig. 7c.

## Discussion

In this study, we address the challenge of fairness in AI models applied to clinical diagnoses. Performance disparity and learning curve experiments highlighted pronounced model bias when using prevalent methods such as Mixed training and Divide-and-conquer

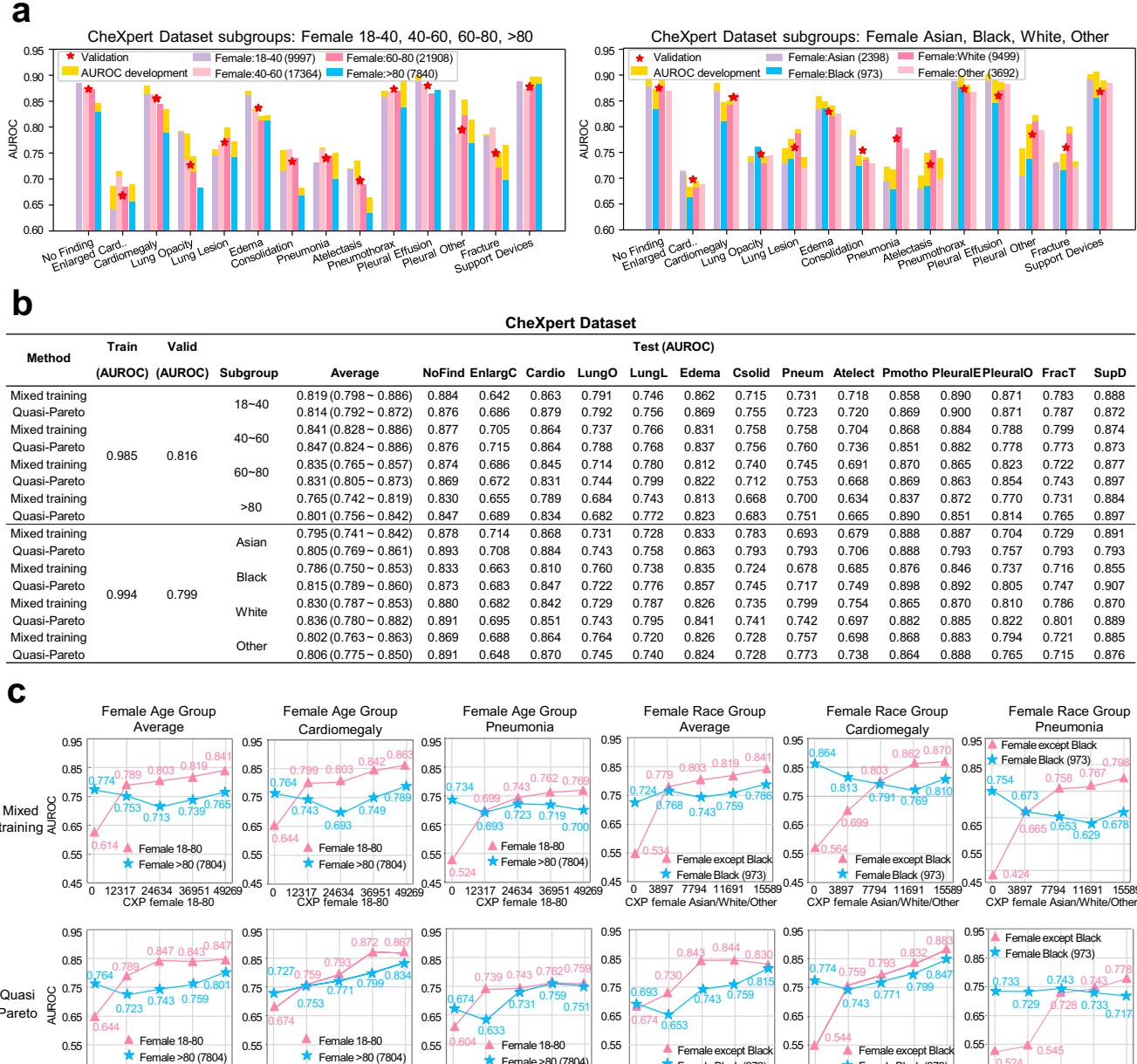

**Fig. 6 | Evaluation performance of QPI approach on CheXpert dataset.**
**a** Significant prediction differences are observed in age and racial subgroups of female patients, respectively. The training dataset size for each group are annotated in the figure legend. Specifically, the ">80" age group and the black female subgroup have the smallest dataset and have shown the lowest AUC performance.

AUC improvement using QPI approach is highlighted on top of each bar. **b** After using QPI approach, we can observe varying degrees of AUC improvement in the prediction performance of the age and racial subgroups of female patients. **c** Learning curves of different subgroups proves effectiveness of QPI approach. Source data are provided as a Source Data file.

approaches, across our thyroid dataset and two comprehensive public datasets spanning multiple subgrouping strategies. In response, we introduced the Quasi-Pareto Improvement approach, designed to enhance prediction accuracy in imbalanced subgroups without compromising overall model performance. Our implementation, termed QP-Net, seamlessly integrates a multi-task learning structure with domain adaptation components, bolstering the network's capability to discern features in imbalanced subgroups.

The QP-Net is adept at minimizing disparities caused by contradictory objectives across subgroups, while simultaneously, leveraging knowledge transfer between subgroups. The effectiveness of the QP-Net has been validated under various subgrouping schemes across multiple datasets. On our thyroid dataset, for example, Fig. 4a shows

that QP-Net reduced the average AUC disparity between PTC and FTC from 0.273 (Mixed training) to 0.070, and Fig. 4c shows that QP-Net can effectively reduce prediction differences introduced by Mixed training. Similar results have also been observed in two public datasets, further validating the effectiveness and generalizability of QP-Net. Moreover, we use CKA to demonstrate that the proposed QP-Net effectively retains more dominant subgroup features that contribute to both dominant subgroup and less-prevalent subgroup prediction during training. This supports that the QPI approach ensures the stability of the predictive performance for the overall population, while also adapting the features of dominant subgroups to a greater extent, thereby enhancing the prediction performance for less-prevalent subgroups.

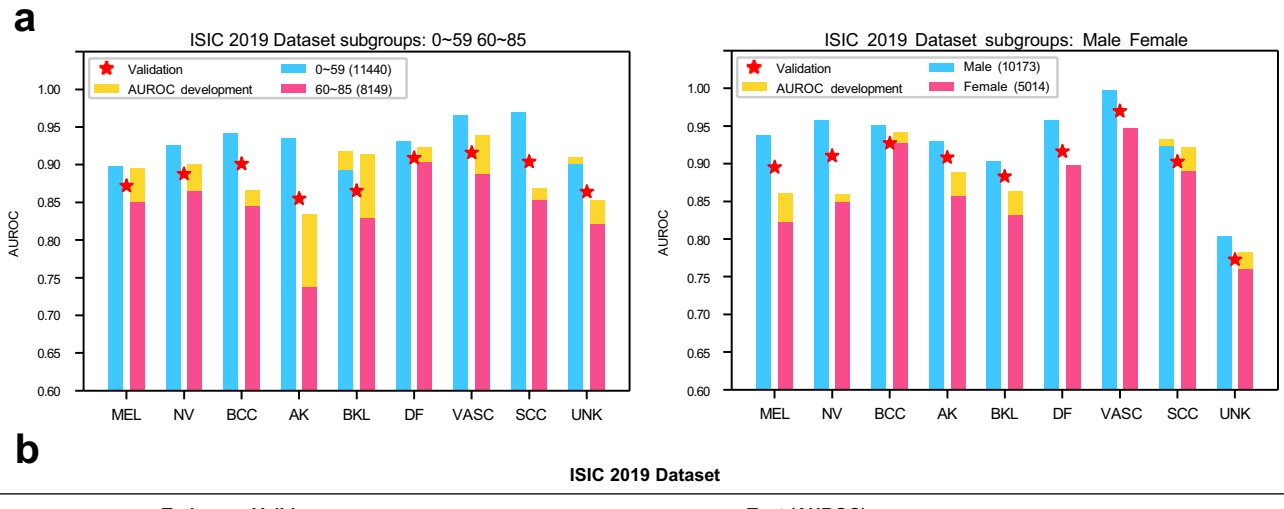

**b**

<table>
<tr><th colspan="12" style="text-align:center">ISIC 2019 Dataset</th></tr>
<tr><th rowspan="2">Method</th><th>Train</th><th>Valid</th><th rowspan="2">Subgroup</th><th colspan="10" style="text-align:center">Test (AUROC)</th></tr>
<tr><th>(AUROC)</th><th>(AUROC)</th><th>Average</th><th>MEL</th><th>NV</th><th>BCC</th><th>AK</th><th>BKL</th><th>DF</th><th>VASC</th><th>SCC</th><th>UNK</th></tr>
<tr><td>Mixed training</td><td rowspan="4">0.962</td><td rowspan="4">0.932</td><td rowspan="2">0-59</td><td>0.897 (0.850 ~ 0.946)</td><td>0.898</td><td>0.925</td><td>0.942</td><td>0.935</td><td>0.893</td><td>0.931</td><td>0.965</td><td>0.969</td><td>0.901</td></tr>
<tr><td>Quasi-Pareto</td><td>0.902 (0.865 ~ 0.949)</td><td>0.891</td><td>0.912</td><td>0.926</td><td>0.936</td><td>0.918</td><td>0.929</td><td>0.961</td><td>0.946</td><td>0.909</td></tr>
<tr><td>Mixed training</td><td rowspan="2">60-85</td><td>0.824 (0.801 ~ 0.862)</td><td>0.851</td><td>0.866</td><td>0.845</td><td>0.737</td><td>0.830</td><td>0.904</td><td>0.888</td><td>0.854</td><td>0.822</td></tr>
<tr><td>Quasi-Pareto</td><td>0.884 (0.820 ~ 0.897)</td><td>0.895</td><td>0.901</td><td>0.866</td><td>0.834</td><td>0.913</td><td>0.923</td><td>0.939</td><td>0.869</td><td>0.853</td></tr>
<tr><td>Mixed training</td><td rowspan="4">0.987</td><td rowspan="4">0.923</td><td rowspan="2">Male</td><td>0.918 (0.872 ~ 0.965)</td><td>0.938</td><td>0.957</td><td>0.951</td><td>0.930</td><td>0.903</td><td>0.957</td><td>0.997</td><td>0.813</td><td>0.858</td></tr>
<tr><td>Quasi-Pareto</td><td>0.922 (0.880 ~ 0.958)</td><td>0.934</td><td>0.956</td><td>0.951</td><td>0.931</td><td>0.905</td><td>0.948</td><td>0.995</td><td>0.857</td><td>0.864</td></tr>
<tr><td>Mixed training</td><td rowspan="2">Female</td><td>0.878 (0.846 ~ 0.931)</td><td>0.823</td><td>0.852</td><td>0.929</td><td>0.858</td><td>0.832</td><td>0.898</td><td>0.948</td><td>0.892</td><td>0.837</td></tr>
<tr><td>Quasi-Pareto</td><td>0.899 (0.842 ~ 0.948)</td><td>0.860</td><td>0.863</td><td>0.942</td><td>0.889</td><td>0.863</td><td>0.886</td><td>0.927</td><td>0.932</td><td>0.824</td></tr>
</table>

**c**

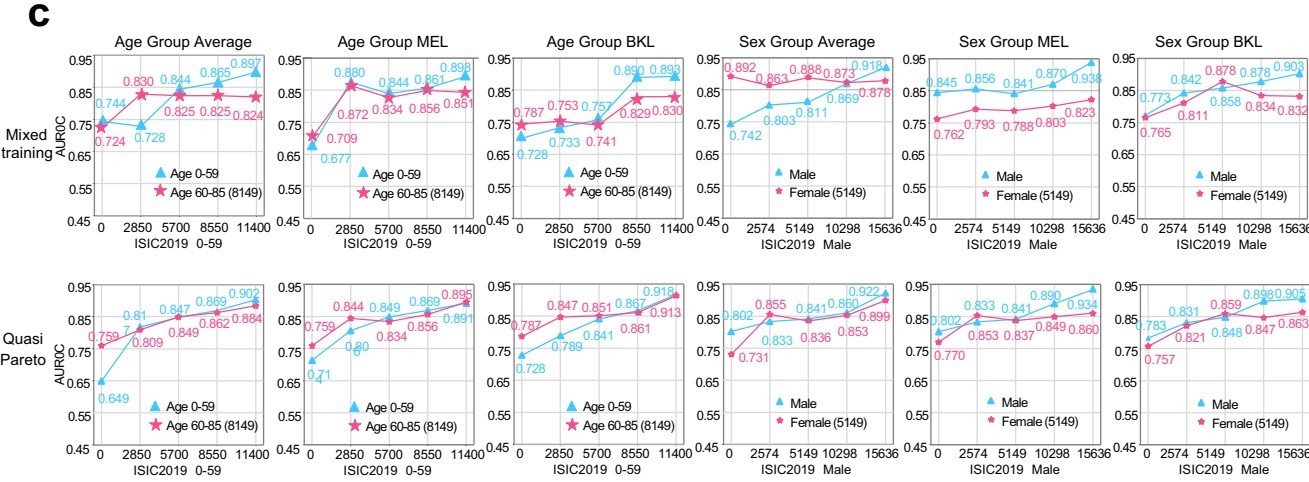

**Fig. 7 | Evaluation performance of QPI approach on the International Skin Imaging Collaboration 2019 dataset. a** Significant prediction differences are observed in the age and the gender subgroups. AUC development using QPI approach is highlighted on top of each bar. The training dataset size for each age group is annotated in the figure legend. Specifically, the dataset of the 60–85 age and the female groups are much smaller than the 0–59 age and the male groups, and the lower AUC performance are shown in these two groups. **b** After using QPI approach, we can observe varying degrees of AUC improvement in the prediction performance of the 60–85 age and the female groups. **c** Learning curves of different subgroups proves effectiveness of QPI approach. Source data are provided as a Source Data file.

This article employs both AUC disparity and Pareto fairness criterion as fairness criteria. Within the academic discourse, fairness in AI models is commonly viewed from two vantage points. One of these perspectives categorizes existing fairness definitions into two subsets: fairness related to model calibration and discrimination capability. Fairness regarding calibration addresses instances where certain subgroups are disproportionately likely to receive positive predictions, deeming such outcomes as unfair[4,23,37,40]. Fairness regarding discrimination, on the other hand, ensures equal model predictability across all subgroups[41–43]. The AUC disparity used in this study falls into the second category. Another perspective bifurcates fairness definitions into causality-based and observational fairness[44]. While the former delves into the roots of unfairness, the latter concentrates on manifest unfair phenomena. In this study, AUC disparity falls under the category of observational fairness, while Pareto fairness criterion belongs to causality-based fairness. Ethically, it's important to note that not all forms of inequity are necessarily considered unfairness. The fairness delineated by Pareto fairness criterion, particularly when stemming from imbalanced samples, holds distinct ethical implications in certain contexts.

In this study, the QPI approach was introduced to mitigate unfairness attributed to imbalanced samples. In previous literature, various methods have also been proposed to tackle similar issues, primarily including sample balancing and domain adaptation

approaches. Based on Stratified batch sampling and Fair meta-learning for segmentation methods, Puyol-Antón et al. has improved the predictive accuracy for Black and Mixed-race subgroups in the segmentation of cardiac MR slices across different ethnicities. Zhang et al. compared several data sampling baseline methods on chest X-rays and concluded that using simple sampling could lead to a decrease in performance for the dominant subgroup population[22]. Idrissi et al. observed that simple group balancing is effective for worst-group-accuracy optimization[45]. Fernando and Tsokos proposed the dynamically weighted Balanced loss to address class imbalance issues in training samples[46]. Compared to the QPI approach proposed in this study, these methods also focus on fairness issues, but they approach fairness from different perspectives to those outlined in Pareto fairness criterion. Supplementary Table 2 showcases the performance of these methods on our thyroid ultrasound dataset. Domain Adaptation approaches are frequently employed to enhance the fairness level of models by transferring them from a source domain to a target domain. This process aims to improve the algorithm's performance on target subgroups. Mukherjee et al. explored the relationship between domain adaptation and individual fairness[47]. Mukhoti et al. used focal loss for model calibration[40]. However, these studies do not explicitly require the models to maintain performance levels within the dominant subgroup population.

Furthermore, this investigation underscores a conundrum: the dominant subgroup dataset exerts both facilitative and detrimental impacts on less-prevalent subgroup model predictions. Our learning curve experiment vividly encapsulates this predicament; as the less-prevalent subgroup's sample size expands, their prediction performance first witnesses an uptick before eventually tapering off. At a fundamental level, the beneficial influence arises from knowledge transfer (i.e., feature reuse) across diverse populations. Conversely, the detrimental effects stem from disparate optimization objectives for varied groups. The CKA experiments further pinpoint this dual influence operative at different tiers of the deep neural network model. Mixed training approach falters due to the unchecked negative influence, while the Divide-and conquer approach underperforms due to the absence of positive effect. The QPI, however, melding both multi-task learning and domain adaptation facets, adeptly curtails the unfavorable influences against minorities while fully harnessing the propitious ones.

There are strengths as well as limitations to this study. We discuss the general issues of medical AI and propose a method in a degree of universality, but future validation is still needed for a broader range of diseases. Also, more complicated scenarios might occur in real-world settings where, given a model trained on a certain dataset, unfairness is observed under multiple different subgroup partitioning schemes that are all applicable to the dataset. While our design of the proposed QPI approach allows for respective utilization under each scheme, further investigation is needed to clarify underlying mechanisms. Furthermore, this study adopts BSS loss as the domain adaptation regularization term within the framework; and its effectiveness compared against other alternatives requires further investigation in future work.

The QPI approach can be broadly applied to address generalization and fairness of AI in clinical practice. To avoid unfairness among subgroups in real-world clinical scenarios, a comprehensive subgroup analysis is always recommended. If unfairness for specific less-prevalent subgroup were to be observed, the QPI approach proposed in this study can be utilized to mitigate unfairness issues. The fairness of medical AI is a highly complex issue, and our understanding of its underlying mechanisms and consequences is still in its infancy. Diverse experts and stakeholders can help gain a deeper understanding of AI applications in medicine and develop appropriate research standards and public policies by engaging in in-depth discussions.

## Methods

### Ethical issues
This study was approved by the institutional review board (IRB) of Shanghai Tong Ren Hospital and undertaken according to the Declaration of Helsinki. Informed consent from patients with thyroid cancer and controls was exempted by the IRB because of the retrospective nature of this study.

### Data source and data pre-processing
We gathered a 10-year dataset of thyroid ultrasound images (from January 2013 to January 2023) and conducted a retrospective study at nine top-tier hospitals and one community hospital in China (see Supplementary methods for the full name list). Each image is evaluated by a physician with more than 5 years of thyroid ultrasound experience. The authors had access to anonymized data only. We initially screened all patients with thyroid nodule in electronic medical record. 148,289 patients who met one of the following criteria were included: (1) Had thyroid cancer or benign diagnosis confirmed by pathology after surgery; (2) had benign diagnosis confirmed by at least 1-year follow-up from experienced radiologists. After inclusion, patients who met one of the following criteria were excluded: (1) comorbidity of other life-threatening condition; (2) lack of or incomplete preoperative ultrasound report; (3) poor ultrasound image quality; (4) controversial pathological diagnosis; (5) history of thyroidectomy or other head-and-neck cancers. After exclusion, 123,301 patients were identified as study participants.

To prevent the interference of textual information at the edges of ultrasound images with the predictive performance of the model, we utilized nnU-net[48] to automatically crop each image, preserving the main image portion while removing the textual information at the edges. While nnU-net achieves a high success rate in image cropping, a common drawback is the frequent occurrence of irregular edges in the cropped images. In our case, we augmented those images with a background with pixel values set to 0. Each image is scaled to 512 × 512 pixels and saved using the PNG file format. Implementation details are provided in Supplementary Materials section: Supplementary details.

It should be noted that although some patients may have more than one corresponding ultrasound image, during the partitioning of the training, validation, and testing datasets, we ensured that there was no patient leakage between the train, valid and test sets.

### Measurement of fairness in medical AI diagnosis
This study is conducted to address a diagnostic problem that aligns with clinical practice, which we have described as follows. Given a dataset $D :: X \times Y$, where $X$ is the instance set and $Y$ is the corresponding class label (diagnosis) space, we denote by $D_{domi}$ and $D_{less}$ the dominant subgroup and less-prevalent subgroup set under a specific subgrouping scheme on $D$, with $|D_{domi}| > |D_{less}|$. $D_{domi}$ and $D_{less}$ share the same $Y$.

For our thyroid ultrasound image dataset, we have $Y = \{Benign, Malignant\}$. In order to detect potential unfairness on our dataset, subgroup analyses were performed for various subgrouping schemes: age, sex, nodule size, histological subtype and hospital, which utilized two measures of fairness: model performance disparity and Pareto fairness criterion. The analyses results were presented in Fig. 2, revealing potential unfairness in each of the two subgrouping schemes: histological subtype and hospital, in which the term 'less-prevalent subgroup' refers to rare subtypes and minority population. Experiments were then performed within each of these two subgrouping schemes.

We perform experiments under these two subgrouping schemes. Under the histological subtype subgrouping scheme, two sets of experiments were performed. Both consider the PTC samples as $D_{domi}$, while the FTC and MTC samples were considered as $D_{less}$. Under the

Hospitals subgrouping scheme, the tertiary hospital (resp. community hospital) samples were considered as $D_{domi}$ (resp. $D_{less}$).

**Test performance disparity.** The performance disparity is often used to indicate model unfairness[49–51]. If a dataset $D$ is divided into $D_1$ and $D_2$ under an underlying subgrouping scheme, the test performance disparity of model $M$ on $D$ is given by:

$$\Delta A = A(M, D_1) - A(M, D_2) \qquad (4)$$

where $A(M, D)$ represents the prediction performance indicator (e.g., AUC) of model $M$ on test set $D$.

Without loss of generality, we also consider a certain subgrouping scheme that divides D into more than 2 subgroup sets. To measure the test performance disparity of $M$ on the entire $D$, $D_1$ (resp. $D_2$) should represent the subgroup on which $M$ has the best (resp. worst) test performance among all the subgroups under the specific subgrouping scheme.

**Pareto fairness criterion.** Performance disparity alone does not encompass the entirety of Pareto fairness. Existing literature presents multiple definitions of Pareto fairness[50]. Most studies on Pareto fairness aim to address fairness concerns within a fixed-resource environment, typically focusing on the 'Pareto frontier'. This represents an optimal state where it's not feasible to benefit some individuals without causing detriment to others[52–54]. Nonetheless, there are phenomena that manifest specifically due to variations in resources, as depicted in Fig. 2c. Such occurrences should also be identified as unfairness. Moreover, the significance of causal-based fairness cannot be overstated when devising fair decision-making algorithms[52,55,56]. From an ethical standpoint, "fairness" recognizes how the unequal outcome is attributable to some systemic disadvantages or discrimination. In the scenarios of medical diagnosis less-prevalent subgroup, a critical concern is whether performance disparities are attributable to sample size imbalances. However, few studies approach this fairness issue from a causal perspective.

To address these limitations, we proposed the "Pareto fairness criterion" measured by experiments based on learning curves. The key idea of "Pareto fairness criterion" is to observe how model performance of less-prevalent subgroup is affected by the sample size of dominant subgroup, which is a causality-based fairness definition differing from existing observational fairness definitions in literature. As depicted in Fig. 2c, the decline in model performance on the less-prevalent subgroup following an initial upward trend can be interpreted as an indication of algorithmic unfairness. This is due to the fact that such inequality arises under the condition of a consistent less-prevalent subgroup sample size, coupled with an increase in the dominant subgroup sample size.

Formally, the Pareto fairness criterion requires a learning curve experiment. Given dataset $D = D_{1,train} \cup D_{1,test} \cup D_{2,train} \cup D_{2,test}$ containing the training and test set of subgroups 1 and 2. Keep all dataset constant except that $|D_{1,train}|$ the training sample size of subgroup 1 increases. An algorithm is said to exhibit Pareto fairness criterion on $D$, if: as $|D_{1,train}|$ increases, its test performance does not decrease for either subgroup 1 ($A(M, D_{1,test})$) or 2 ($A(M, D_{2,test})$).

With Pareto fairness criterion, it can be easily observed that in Fig. 2c, the unfairness phenomenon present in Mixed training is ameliorated in Fig. 4c following the utilization of the QPI approach. Specifically, in Fig. 2c, the reduction in less-prevalent subgroup performance is attributed to the increase of the dominant subgroup training sample size, thereby contravening the second criterion of Pareto fairness criterion and consequently representing a form of unfairness. Conversely, in Fig. 4c, the concurrent augmentation of

dominant subgroup and less-prevalent subgroup performance due to the identical size increase aligns with the Pareto fairness criterion, thus signifying a more equitable phenomenon.

## Approaches to handle unfairness of medical AI diagnosis

As illustrated in Fig. 1, we compared three approaches to handle subgroups for AI model, with two common practices as baseline: Mixed training and divide and conquer. The third approach is proposed by this study, named the Quasi-Pareto Improvement approach. Given a dataset $D = D_{domi} \cup D_{less}$ and denote the loss function as $\mathscr{L}$, we provide detailed explanation for the above-mentioned approaches.

**Mixed training.** Mixed training approaches try to train a model with training set combining $D_{domi}$ and $D_{less}$. The general training objective of Mixed training approaches is to find a common model $\hat{M}_{mix}$ for both dominant subgroup and less-prevalent subgroup, so that:

$$\hat{M}_{mix} = \arg \min_M E_{(x,y)\in D}[\mathscr{L}] \qquad (5)$$

**Divide and conquer.** In the context of divide and conquer approaches, $D_{domi}$ and $D_{less}$ are employed as distinct training sets to separately train models, with the objective of identifying an optimum model for the dominant subgroup and the less-prevalent subgroup, respectively. The general training objectives are as follows:

$$\hat{M}_{domi} = \arg \min_M E_{(x,y)\in D_{domi}}[\mathscr{L}] \qquad (6)$$

$$\hat{M}_{less} = \arg \min_M E_{(x,y)\in D_{less}}[\mathscr{L}] \qquad (7)$$

**Quasi-Pareto Improvement (QPI).** The proposed QPI approach is to perform additional transfer learning on $\hat{M}_{mix}$ to further enhance the model's performance on the less-prevalent subgroup, while endeavoring to maintain its performance on the dominant subgroup. Previous investigations into Pareto improvement mandate that a method shouldn't disadvantage some subgroups while benefiting others[52,54,57,58]. Our QPI approach doesn't rigidly constrain the performance of the dominant subgroup. However, empirical findings from our learning curve experiments, shown in Fig. 4d, indicate that our technique can ameliorate fairness concerns, especially those that emerge with changes in resources, such as increases in sample size.

The QPI approach necessitates the completion of the following two tasks, which inherently form a multi-task learning problem.

Formally, based on $\hat{M}_{mix}$ and $D = D_{domi} \cup D_{less}$, the QPI approach aims to find $\hat{M}_{QPI}$ that:

a. Maximize $A(\hat{M}_{QPI}, D_{less})$;

b. Maintain $\epsilon = A(\hat{M}_{mix}, D_{domi}) - A(\hat{M}_{QPI}, D_{domi})$ at a low level.

## QP-Net to implement Quasi-Pareto improvement

We designed a QP-Net which combined SOTA multi-task learning and domain adaptation techniques to implement Quasi-Pareto Improvement approach, as illustrated in Fig. 3. Denote as $(x, y, d)$ an input image $\mathbf{x}$ with its corresponding class label $y$ and subgroup $d$, $\mathscr{F}_z$ is a feature extractor that extracts features $\hat{z}$ from the image $\mathbf{x}$. $\mathscr{F}_y$ is a class predictor that provides a class prediction $\hat{y}$ for $\mathbf{x}$ based on $\hat{z}$. The combination of $\mathscr{F}_z$ and $\mathscr{F}_y$ is the general architecture of medical image diagnostic models. $\mathscr{F}_d$ is an additional subgroup predictor that we incorporate in our QP-Net following Ganin et al. which generates a subgroup prediction $\hat{d}$ for $\mathbf{x}$ based on $\hat{z}$.

Formally, we denote as $\boldsymbol{\omega}_z$, $\boldsymbol{\omega}_y$ and $\boldsymbol{\omega}_d$ the parameters of $\mathscr{F}_z$, $\mathscr{F}_y$, and $\mathscr{F}_d$ respectively. The forward process of our devised QP-Net is

shown as follows:

$$\hat{z} = \mathscr{F}_z(\mathbf{x}; \boldsymbol{\omega}_z) \quad (8)$$

$$\hat{y} = \mathscr{F}_y(\hat{z}; \boldsymbol{\omega}_y) \quad (9)$$

$$\hat{d} = \mathscr{F}_d(\hat{z}; \boldsymbol{\omega}_d) \quad (10)$$

It should be noted that although $\mathscr{F}_z$, $\mathscr{F}_y$, and $\mathscr{F}_d$ were trained concurrently, only the original model structure ($\mathscr{F}_z$ and $\mathscr{F}_y$) were utilized to predict the data class labels during testing.

As illustrated in Fig. 3, we designed the final loss function as follows:

$$\mathscr{L}(\boldsymbol{\omega}_z, \boldsymbol{\omega}_y, \boldsymbol{\omega}_d) = \mathscr{L}_y(\boldsymbol{\omega}_z, \boldsymbol{\omega}_y) + \mathscr{L}_{DA}(\boldsymbol{\omega}_z, \boldsymbol{\omega}_d) \quad (11)$$

Where $\mathscr{L}_y$ refers to the multi-task learning loss, and $\mathscr{L}_{DA}$ refers to the domain adaptation loss.

**The multi-task learning loss.** For $\mathscr{L}_y$, we form a multi-task learning structure, treating data from different subgroups (dominant subgroup or less-prevalent subgroup) as distinct tasks, and reweight their losses:

$$\mathscr{L}_y = \delta \mathscr{L}_{domi} + (1 - \delta) \mathscr{L}_{less} \quad (12)$$

Where $\mathscr{L}_{domi} = \sum_{(\mathbf{x}_i, y_i) \sim D_{domi}} l_y(\mathbf{x}_i, y_i)$, $\mathscr{L}_{less} = \sum_{(\mathbf{x}_j, y_j) \sim D_{less}} l_y(\mathbf{x}_j, y_j)$. The value of $\delta$ is varied and determined through the training process. $A_{domi}$ (resp. $A_{mix}$) is the test performance of the current model (resp. original Mixed training model) on the dominant subgroup dataset. $\delta$ satisfies the following requirement: when the gap between $A_{domi}$ and $A_{mix}$ is narrow, the value of $\delta$ would be relatively low; as training progresses, if $A_{domi}$ becomes much lower than $A_{mix}$, $\delta$ would be adjusted to a higher value. In experiment, we adaptively determined $\delta$ based on the following equations:

$$\delta(A_{domi}, A_{mix}) = \frac{1}{1 + e^{h(A_{domi}, A_{mix})}} \quad (13)$$

$$h(A_{domi}, A_{mix}) = \frac{A_{domi} - A_{mix}}{\sigma} \quad (14)$$

where $\sigma$ controls the level of $\epsilon$ allowed during training, according to the definition of QPI. In experiments we choose $\sigma$ to 0.05 through comparison experiments (Supplementary Fig. 1). It should be noted that in different real-world settings, the optimal choice of $\sigma$ can vary due to different underlying subgroup feature distributions.

**The domain adaptation loss.**

$$\mathscr{L}_{DA}(\boldsymbol{\omega}_z, \boldsymbol{\omega}_d) = -\gamma_d \mathscr{L}_d(\boldsymbol{\omega}_z, \boldsymbol{\omega}_d) + \gamma_{MMD} \mathscr{L}_{MMD}(\boldsymbol{\omega}_z) + \gamma_{BSS} \mathscr{L}_{BSS}(\boldsymbol{\omega}_z) \quad (15)$$

There are three components within the domain adaptation loss, namely $\mathscr{L}_d$ the adversarial structure loss[27], $\mathscr{L}_{MMD}$ the MMD loss[28], and $\mathscr{L}_{BSS}$ the BSS loss[29].

For $\mathscr{L}_d(\boldsymbol{\omega}_z, \boldsymbol{\omega}_d) = \sum L_d(\mathbf{x}_i, d_i; \boldsymbol{\omega}_z, \boldsymbol{\omega}_d)$, cross-entropy is used to calculate $L_d$, which is defined according to the following expression:

$$L_d(\mathbf{x}_i, d_i; \boldsymbol{\omega}_z, \boldsymbol{\omega}_d) = l_d(\hat{d}_i, d_i) = \begin{cases} -\log \hat{d}_i, & d_i = 1 \\ -\log(1 - \hat{d}_i), & d_i = 0 \end{cases} \quad (16)$$

$\mathscr{L}_{MMD}$ (Maximum Mean Discrepancy)[28] measures the distance of features from domains using the distance of the mean of the two domains after mapping each sample to a Reproducing Kernel Hilbert Space (RKHS). In our case, we defined the MMD loss as:

$$\mathscr{L}_{MMD} = \left\| \sum_{i=1}^{|D_{domi}|} \frac{\phi(\hat{z}_i^{domi})}{|D_{domi}|} - \sum_{j=1}^{|D_{less}|} \frac{\phi(\hat{z}_j^{less})}{|D_{less}|} \right\|^2 \quad (17)$$

where, $\phi$ is the mapping function to RKHS, which is usually unknown in practice. Additionally, we defined function $M_{D_1 \Delta D_2}$ on samples from $D_1$ and $D_2$ as follows:

$$M_{D_1 \Delta D_2} = \sum_{\mathbf{x}_i \sim D_1} \sum_{\mathbf{x}_j \sim D_2} \frac{k(\mathscr{F}_z(\mathbf{x}_i; \boldsymbol{\omega}_z), \mathscr{F}_z(\mathbf{x}_j; \boldsymbol{\omega}_z))}{N_1 N_2} \quad (18)$$

where, $N_i$ is the number of samples drawn from $D_i$, and $k(\cdot, \cdot)$ is a kernel function, so that $\mathscr{L}_{MMD}$ can be rewritten as follows:

$$\mathscr{L}_{MMD} = M_{D_{domi} \Delta D_{domi}} - 2 M_{D_{domi} \Delta D_{less}} + M_{D_{less} \Delta D_{less}} \quad (19)$$

During experiments, we designed $k(\cdot, \cdot)$ as $k(\mathbf{z}_1, \mathbf{z}_2) = e^{-||\mathbf{z}_1 - \mathbf{z}_2||^2}$, and computing $\mathscr{L}_{MMD}$ in this way is proven to be effective.

For $\mathscr{L}_{BSS}$, we apply the BSS loss which suppresses the small singular values of the feature matrices during each training batch to mitigate negative transfer[29]. Within a training batch, we construct a feature matrix $\mathbf{Z}$ from a batch size $b$ of feature vectors $\hat{z}$, and apply SVD on $\mathbf{Z}$ as $\mathbf{Z} = \mathbf{U}\boldsymbol{\Sigma}\mathbf{V}^*$. In this case, $\boldsymbol{\Sigma}$ is the singular value matrix, of which the main diagonal elements $[\sigma_1, \sigma_2, ..., \sigma_b]$ are the singular values of $\mathbf{Z}$. $\mathscr{L}_{BSS}$ in this training batch is defined as the squared sum of the smallest $k$ singular values:

$$\mathscr{L}_{BSS} = \sum_{i=1}^{k} \sigma_{-i}^2 \quad (20)$$

where, $\sigma_{-i}$ is the $i$-th smallest singular value of $\mathbf{Z}$.

Training details are provided in Supplementary materials section: Supplementary details.

## Centered Kernel Alignment (CKA)

To quantitatively compare network features before and after transfer learning, we utilized CKA which can reliably identify similarities between layer representations across networks as per Kornblith et al. (2019). Within architecturally identical networks $M_1$ and $M_2$, each pair of corresponding layers possessing matrix of activations of $j_1$ (from $M_1$) and $j_2$ (from $M_2$) neurons for certain $i$ examples are subsequently denoted as $\mathbf{W}_1 \in R^{i \times j_1}$ and $\mathbf{W}_2 \in R^{i \times j_2}$ respectively[59]. CKA was then calculated using the following formula:

$$CKA(W_1, W_2) = \frac{HSIC(\mathbf{K}, \mathbf{L})}{\sqrt{HSIC(\mathbf{K}, \mathbf{K}) HSIC(\mathbf{L}, \mathbf{L})}} \quad (21)$$

where, $\mathbf{K} = \mathbf{W}_1 \mathbf{W}_1^T$ and $\mathbf{L} = \mathbf{W}_2 \mathbf{W}_2^T$, and HSIC (Hilbert-Schmidt independence criterion) calculates the statistical independence of the two distributions by normalization. We refer to Kornblith et al. (2019) for detailed calculation.

Built upon this foundation, Neyshabur et al. (2020) designed experiments employing CKA to quantify the degree of feature reuse in transfer learning[60]. The CKA experiments we conducted in Fig. 5 are based on similar design. In our CKA experiments, we primarily employed two ResNet-18 models: The ImageNet pretrained network $M_1$, and the network $M_2$ trained on dominant subgroup dataset with $M_1$ as initial state.

In Fig. 5a, $CKA_1$ (resp. $CKA_2$) represents the similarity between network representations before and after transfer learning on the less-prevalent subgroup dataset when using ImageNet pretrained model (resp. Mixed training model) as the initial state, quantifying how many features are retained from the respective model after the transfer learning process.

Subsequently, we calculate $\Delta CKA = CKA_2 - CKA_1$, which measures additional features reused from Mixed training model compared with ImageNet pretrained model. Due to the influence of different transfer learning strategies on the retaining of initial features, $\Delta CKA$ can reflect whether the additional contribution of the Mixed training model to less-prevalent subgroup prediction at various layers of the network is positive or negative under certain transfer learning strategies.

### Experimental variables control

All experiments for Mixed training, divide and conquer, and QPI approach utilize ResNet18 as the primary neural network.

**Data loading section.** The process of data extraction is set to randomly extract without repetition. Data is grouped into batches of the same size (248) and fed into the model. Class balance is implemented prior to the training session, ensuring an equal benign-malignant ratio among the subgroups.

**Training section.** The initial learning rate for all experiments is set to 0.0001. All three models use the same optimizer, with the learning rate reduced by 0.5 every 20 epochs. For binary and multi-class classification problems, Cross-Entropy Loss (CE Loss) is used, and for CheXpert task with multiple binary classifications, Binary Cross-Entropy Loss (BCE Loss) is employed. Early stopping is applied to all models; training is halted on the next epoch if the change in validation AUC over the past ten epochs is less than 0.01.

**Result selection.** Each model's final results are obtained by averaging the outcomes from five repeated experiments. The confidence interval for the AUC is calculated by resampling from the test set five times, each time 1000 images with replacement.

### Reporting summary

Further information on research design is available in the Nature Portfolio Reporting Summary linked to this article.

## Data availability

Data supporting the findings described in this manuscript are have either been made available online or could be obtained from the corresponding author upon request. Anonymized MICCAI 2020 TN-SCUI ultrasound images of thyroid nodules used in this study can be accessed at https://github.com/fangdai-dear/QuasiParetoImprovement. Anonymized partial thyroid ultrasonography data used in this study are subject to privacy restrictions but may be made available upon request to the corresponding authors. Response to requests will be made within 15 work days. Source data are provided with this paper. The CheXpert public dataset is available at https://stanfordmlgroup.github.io/competitions/CheXpert/. The ISIC2019 Skin Image public dataset is available at https://challenge.isic-archive.com/data/#2019. Source data are provided with this paper.

## Code availability

The code necessary to reproduce the results of this manuscript are provided at https://github.com/fangdai-dear/QuasiParetoImprovement and deposited to Zenodo[61].

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

## Acknowledgements

This work was supported in part by funding from the Neil Shen's SJTU Medical Research Fund, SJTU Trans-med Awards Research STAR20210106, the Innovative Research Team of High-Level Local

Universities in Shanghai (SHSMU-ZDCX20212200), NSFC 81903417. The content of this article does not reflect the view of the funding sources. We would like to express our gratitude to Xiaoqiang Yu, Yicheng Zhu, Jingjing Sun, Min Wang, Maofeng Wang, and Yun Wang for their invaluable assistance in data collection and sample annotation. Our sincere thanks to Jianfeng Sang and Fenyong Sun for providing us with clinical guidance.

## Author contributions

S.Y., F.D., P.S, W.Z., B.Q., and H.L. developed the concept for the manuscript. S.Y. and F.D. contributed to drafting of the manuscript. S.Y., F.D. and P.S. designed the model and analysis the data. W.Z. and H.L. contributed to critical revision of the manuscript. B.Q. contributed to provide medical data and clinical advice. S.Y. and F.D. contributed equally to this work. All authors critically revised the manuscript.

## Competing interests

The authors declare no competing interests.
