## [Peer Review File · Nature Communications]

Reviewers' Comments:

Reviewer #1:

Remarks to the Author:

==== Key results =====

The authors address the problem of discrepancies in machine learning model performance between patients groups, with a particular emphasis on performance differences between majority and minority groups. They do so in the context of thyroid nodule classification based on ultrasound images. Towards this end, they describe and analyze a large (123301 patients, 360455 images) private dataset of expert-annotated images from ten Chinese hospitals. The compared patient subgroups are based on age, sex, nodule size, histological subtype, and hospital type. They demonstrate how standard empirical risk minimization strongly favors majority groups, and they propose a new approach, based on the combination of several prior works, to reduce the performance differences while maintaining high performance for the majority groups. The efficacy of the new approach is demonstrated on the private thyroid ultrasound dataset described above, as well as on two public datasets (skin lesions / ISIC2019 and chest x-rays / CheXpert).

==== Significance =====

Methodologically, there is only limited novelty in this manuscript. The newly proposed unfairness mitigation method ("Quasi-Pareto Improvement") consists of the application of a loss function that combines four previously (separately) proposed terms:

- 1) a dynamic weighted loss proposed by Chen et al. (2020),
- 2) an standard adversarial domain classification loss as proposed, for example, by Ajakan et al. (2014),
- 3) a standard MMD loss, also for domain invariance purposes, like proposed, for example, by Tzeng et al. (2014), and
- 4) a Batch spectral shrinkage loss proposed by Chen et al. (2019).

This is not to say that the results presented in this manuscript could not be significant. The analysis of such a large dataset, and the subgroup differences therein, is certainly of interest. Moreover, I commend the authors for working on and raising awareness of the highly important issue of subgroup performance differences.

However, the authors appear to claim significance primarily based on their new unfairness mitigation approach. In my view, that claim would require stronger evidence than is currently provided in the manuscript. The authors provide neither an ablation study concerning the different elements of their method (i.e., the different loss terms) nor a comparison to any of the various previously proposed unfairness mitigation approaches. As such, it is currently not possible to judge a) whether all of the components of the proposed approach are actually important and necessary, and b) how the new approach compares to previous work. Some important baseline methods to compare to should include at least:

- i) simple group balancing (see Zhang et al. and Idrissi et al.),
- ii) a minimax approach such as the one proposed by Martinez et al.,
- iii) simple domain invariance in isolation, and
- iv) the dynamic weighted loss in isolation.

==== Data and methodology =====

The analyzed private dataset is certainly impressive in size. Several details remained unclear to me, however.

- I have concluded from the text that probably the numbers in the first column of Fig. 2 refer to patients, whereas the numbers in the later columns (train/val/test) refer to images? This should be clarified.
- The numbers in the "Hospital" category sum to more than the numbers in the "All" category.
- The authors write that exclusion criteria "such as [...]" were applied. Are there others that were

applied that are not listed?

- The authors list three patient inclusion criteria. I suspect patients had to satisfy just one of these? This should be clarified; the current formulation is ambiguous in this regard. Also, what was the initial selection process? Were **all** patients in participating institutions selected that satisfied the inclusion criteria and not the exclusion criteria? Or was there some prior selection process?
- Did the authors ensure that there was no patient leakage between the train/val/test sets?
- The authors write that they "use nnU-net to automatically crop each image retaining the main portion of it". How precisely do they do this?
- The authors write that "for irregular images, a background with pixel value 0 is added". What does this mean? What is an "irregular image"?

Some important aspects of the fundamental setup of the prediction task also remained unclear to me.

1) I could not figure out the fundamental prediction task that the authors are solving. Is it binary classification (malignant/benign)? Or multi-label (different thyroid nodule subtypes)? In other words, what is "Y" in the Problem formulation section?

2) Similarly, what is "D" / which domains do the authors consider? All possible combinations of the different subgroups, i.e., would "female, 21-35y, small nodule, PTC, Community Hospital" be one of the considered domains, and are the domain classifier and MMD terms then distinguishing between all of these combinations? Or are there different domain classifiers acting on the Male vs. Female, Tertiary vs. Community Hospital etc. axes? Or something else entirely? In Eq. (11), what are D_{min} and D_{maj} ?

3) The authors write that hyper parameters were "determined through experiments". Could they be a bit more specific? How were they determined, and on which dataset (train/val/test) were the corresponding experiments performed?

==== Analytical approach ====

While the unfairness mitigation approach proposed by the authors does not appear implausible to me, I would like to see a more convincing motivation for this particular approach, and empirical evidence of the utility of the individual components.

1) It is known that marginal domain invariance approaches have drawbacks - in particular, they require "equalizing" disease type distributions between domains, which may be detrimental concerning rare subtypes. See, e.g., Tachet des Combes et al., Zhao et al., or Li et al. I would like to see, firstly, an argument for why the authors believe that this will still be useful to do here, secondly, whether they considered using class-conditional invariance instead, and thirdly, empirical evidence that it actually helps. The latter would be provided by an ablation experiment.

2) Both adversarial domain classification and the MMD loss serve the same goal: enforcing (marginal) domain invariance. Why do the authors use both at the same time? Again, empirical evidence that this is actually helpful would be appreciated.

3) Concerning the dynamic weighted loss, the authors write that " g_{Δ} satisfied the following requirements..." but they never actually define this function. Moreover, the corresponding reference to Chen et al. is incomplete; the paper title and any other identifiers (DOI, URL, ...) are missing.

4) What motivated the authors to add the BSS loss (and not any other standard regularization term)? Again, empirical evidence of its utility in the given context would be appreciated.

==== Clarity and context ====

Besides the clarity-related issues already outlined above, I consider the manuscript's engagement with prior literature to be severely lacking. In particular, very little credit is given to the extremely extensive prior literature on domain adaptation and algorithmic fairness (including in medical imaging). In some places, it reads as if the authors claim novelty on the methodological side.

Oakden-Rayner et al. discuss problems resulting from the aggregation of different disease subtypes into a single label, closely related to the authors' consideration of rare disease subtypes.

Adversarial and MMD-based domain invariance approaches have been used extensively for over a decade; see some of the references provided above.

Mukherjee et al. discuss the relationship between domain adaptation and algorithmic fairness approaches. The use of group-invariant representations is a standard approach in the algorithmic fairness literature and has been known since at least Zemel et al. (2013). Thus, I would strongly contest the authors' claim of "innovation" on this point.

The authors claim in various places that their method is unsupervised, but it is not clear if that is true. It seems that both domain labels and class labels are required for all samples in the authors' method?

Finally, there is a fundamental difference between, on the one hand, demographic patient groups and, on the other hand, different disease subtypes (representing output classes). I suggest that the authors clearly distinguish between these, and not summarize them under the single term "groups".

Other clarity issues:

- For the experiments presented in the graphs in fig. 4, what is the corresponding minority dataset size? How exactly were these experiments performed, i.e., how were the extra images to add selected?
- The whole "CKA Explanation" section was very opaque to me. Are feature distributions compared between different *networks* or between different *patient groups* here? What are "CKA lines", what do the arrows in Fig. 5(a) and (b) between the different "lines" mean, and what are all the axis labels in that figure?
- In Figs. 6 and 7, how were the specific subgroups to show here selected? Were trends comparable in other subgroups?
- In the discussion, the authors write that "prediction differences might exist even if the proportions of subgroup data are identical. In other words, that proportional imbalance may not necessarily result in prediction differences." - Both individual statements are correct, but the second statement does not follow from the first one.
- When / where is Eq. (24) used? Also, I assume this is taken from some standard reference book that should be cited?

==== Suggested improvements =====

In conclusion, I would suggest the following major improvements:

1. Addition of and comparison to a much wider range of baseline methods.
2. Addition of an ablation study that assesses the importance and utility of the four different loss function components.
3. Stronger engagement with and clearer referencing of prior work throughout the manuscript, in particular concerning algorithmic fairness and domain adaptation approaches.
4. Improvement of the clarity (and completeness) of presentation.

==== Reviewer expertise =====

My expertise is mostly in machine learning and algorithmic fairness as applied to medicine & medical imaging. I have no expertise concerning ultrasound images or thyroid cancer.

==== References ====

Ajakan et al. (2014), Domain-Adversarial Neural Networks, NeurIPS.
<https://arxiv.org/abs/1412.4446>.

Idrissi et al. (2022), Simple data balancing achieves competitive worst-group-accuracy, Conference on Causal Learning and Reasoning / Proceedings of Machine Learning Research.
<https://proceedings.mlr.press/v177/idrissi22a>.

Li et al. (2020), Rethinking Distributional Matching Based Domain Adaptation,
<https://arxiv.org/abs/2006.13352>.

Martinez et al. (2020), Minimax Pareto Fairness: A Multi Objective Perspective, ICML.
<http://proceedings.mlr.press/v119/martinez20a>.

Mukherjee et al. (2022), Domain Adaptation meets Individual Fairness. And they get along. NeurIPS. <https://arxiv.org/abs/2205.00504>.

Oakden-Rayner et al. (2020), Hidden stratification causes clinically meaningful failures in machine learning for medical imaging, ACM Conference on Health, Inference, and Learning.
<https://doi.org/10.1145/3368555.3384468>.

Tachet des Combes et al. (2020), Domain Adaptation with Conditional Distribution Matching and Generalized Label Shift, NeurIPS.
<https://papers.nips.cc/paper/2020/hash/dfbfa7ddcfff581f50edcf9a0204bb-Abstract.html>.

Tzeng et al. (2014), Deep Domain Confusion: Maximizing for Domain Invariance,
<https://arxiv.org/abs/1412.3474>.

Zemel et al. (2013), Learning Fair Representations, ICML.
<http://proceedings.mlr.press/v28/zemel13.html>.

Zhang et al. (2022), Improving the fairness of chest x-ray classifiers, Conference on Health, Inference, and Learning (CHIL). <https://proceedings.mlr.press/v174/zhang22a>.

Zhao et al. (2022), Fundamental Limits and Tradeoffs in Invariant Representation Learning, JMLR.
<https://jmlr.org/papers/v23/21-1078.html>.

Reviewer #2:

Remarks to the Author:

Summary

The authors propose a novel framework to reduce bias in a classification problem where performance drops for minority subgroups, addressing a central problem related to the area of fairness in AI. This is formulated in the context of thyroid ultrasound nodules, although the methodology is presented in a generic way, and the application in hand is a proof of concept of the framework.

The method performs very well compared to a naive approach (basically, not doing anything about the minorities) and to a simple approach (producing a different model for each subgroup), improving the performance in the minority groups while maintaining overall performance. The results hold for two other datasets that are tested.

The main issue is that there is very limited analysis of related methods in AI in healthcare or AI in medical imaging to put the proposed Quasi Pareto Improvement method in the context of the related literature; this should be clearly investigated in a "related work" section, and also the state

of the art methods for supervised bias reduction should be included in the quantitative analysis. Without this, it is difficult to understand the potential impact of the method and its relevance beyond thyroid nodule ultrasound imaging, so I believe this additional set of experiments, with the corresponding statistical analysis is needed; else, authors need to argue why there is no suitable methodology related to fairness in AI to compare the proposed method to.

Other minor comments are as follows:

1. I would suggest some reorganization of the text to inform the reader early on that this study is focused on thyroid nodules, and then broaden the scope to explain what the impact of the work is beyond this application, in the context of the general literature of fairness in AI. Currently the opposite is done.

2. The initial "summary" section seems too large, getting rather deep into the methods and results, for example including a "CKA explanation" subsection which seems out of place. This part of the paper should be more succinct, leaving details for the paper body. Please check other articles in the journal for reference.

3. The domain classification sub net seems a multi-task learning approach. How is this different otherwise?

4. L31: "Using thyroid cancer as an example, we conducted a literature search using keywords such as "thyroid nodule", "malignant diagnosis", and "rare subtype"...." This search is very targeted at a clinical application, rather than at a more general AI methodology to address fairness and biases, where the literature is broader. This is actually fine if the paper is scoped specifically to the application of interest (ultrasound thyroid nodule imaging). The way the beginning of the introduction is written suggests that there is less work on the topic (generally) than there actually is. I suggest then to introduce the application of interest *before* describing the problem of bias and fairness and *before* describing the literature search.

5. L329 "where, (...) are hyper parameters determined through experiments." Please include (perhaps in an appendix or supplementary material) all these experiments to enable reproducibility of this research. Also one needs to understand how the different approaches (Quasi-pareto, divide and conquer, single model) were optimized to ensure fair comparison.

Response to reviewers' letter

Manuscript ID: NCOMMS-23-21729

Title: Enhancing the Generalizability and Fairness of Ultrasonographical AI Model among Heterogeneous Thyroid Nodule Population by a Novel Quasi-Pareto Improvement

We sincerely thank the reviewers for their positive comments and constructive suggestions to improve our manuscript. The following is a copy of the comments (marked in blue) and our detailed point-by-point response. The revised version of the manuscript, marked in red in the text, can be located in the re-submitted files. We have extensively revised the manuscript in accordance with the feedback and suggestions provided by the reviewers. In particular, we present comprehensive experimental results, including an ablation study, comparisons with state-of-the-art domain adaptation methods, and experiments comparing hyperparameters. Should the article's length become restricted, please guide us in making necessary edits or revisions.

Reply to Reviewer # 1:

Comments:

The authors address the problem of discrepancies in machine learning model performance between patients groups, with a particular emphasis on performance differences between majority and minority groups. They do so in the context of thyroid nodule classification based on ultrasound images. Towards this end, they describe and analyze a large (123301 patients, 360455 images) private dataset of expert-annotated images from ten Chinese hospitals. The compared patient subgroups are based on age, sex, nodule size, histological subtype, and hospital type. They demonstrate how standard empirical risk minimization strongly favors majority groups, and they propose a new approach, based on the combination of several prior works, to reduce the performance differences while maintaining high performance for the majority groups. The efficacy of the new approach is demonstrated on the private thyroid ultrasound dataset described above, as well as on two public datasets (skin lesions / ISIC2019 and chest x-rays / CheXpert).

Response:

We would like to express our sincere gratitude to the reviewer for their positive feedback and valuable suggestions on our research. The main contributions of this study are twofold: First, we clarified the unfairness issues attributed to sample imbalance by introducing a causal criterion "Pareto fairness" based on learning curve. Second, we propose the Quasi-Pareto Improvement (QPI) approach to mitigate unfairness that can be attributed to sample imbalance. We have designed the QP-Net as an implementation of the QPI approach, which incorporated a multi-task learning module and a domain adaptation module, and we have been validated on our thyroid ultrasound dataset as well as on two publicly available datasets. QP-Net can reduce discrimination against minorities due to conflicting objectives in different subgroups, meanwhile, fully benefit from the transfer of knowledge between subgroups.

Comments:

Methodologically, there is only limited novelty in this manuscript. The newly proposed unfairness mitigation method ("Quasi-Pareto Improvement") consists of the application of a

loss function that combines four previously (separately) proposed terms:

- 1) a dynamic weighted loss proposed by Chen et al. (2020),
- 2) a standard adversarial domain classification loss as proposed, for example, by Ajakan et al. (2014),
- 3) a standard MMD loss, also for domain invariance purposes, like proposed, for example, by Tzeng et al. (2014), and
- 4) a Batch spectral shrinkage loss proposed by Chen et al. (2019).

This is not to say that the results presented in this manuscript could not be significant. The analysis of such a large dataset, and the subgroup differences therein, is certainly of interest. Moreover, I commend the authors for working on and raising awareness of the highly important issue of subgroup performance differences.

Response:

We appreciate the reviewer's support of the goal of our work. There are indeed four components in the loss function of our devised QP-Net, including L_y , L_d (adversarial structure), L_{MMD} (MMD loss), and L_{BSS} (BSS loss).

We sincerely apologize for our previous mistakes in naming L_y as 'dynamic-weighted imbalanced subgroups, while the 'dynamic-weighted loss', as proposed by Fernando et al. (2021), was specifically tailored to tackle imbalanced class problems. In the revised manuscript, we have uniformly revised the nomenclature for L_y , referring to it as the 'multi-task learning loss'.

Our proposed QP-Net approach combines two key components: the multi-task learning module and the domain adaptation module, which was required by the special objective of the QPI approach. It simultaneously asks to fulfill two objectives: maximizing the performance of the minority subgroup while preserving the performance of the majority subgroup as much as possible. The employed components, namely L_d , L_{MMD} , and L_{BSS} , in QP-Net are methods that have been used in many recent studies to achieve state-of-the-art results and have demonstrated the best performance on our dataset among existing methods. In the subsequent responses, we provide a more detailed comparative validation of the roles and necessity of these components (via an ablation study).

In the revised manuscript, we have also made modifications to the presentation and explanatory text of the loss function in the Results section to provide a more intuitive depiction of its 'two-part composition' feature. The updated content is as follows:

Line 110 - 125:

'As shown in Fig. 3, the QP-Net ... in Methods section: QP-Net to implement QPI.'

Comments:

However, the authors appear to claim significance primarily based on their new unfairness mitigation approach. In my view, that claim would require stronger evidence than is currently provided in the manuscript. The authors provide neither an ablation study concerning the different elements of their method (i.e., the different loss terms) nor a comparison to any of the various previously proposed unfairness mitigation approaches. As such, it is currently not possible to judge a) whether all of the components of the proposed approach are actually important and necessary, and b) how the new approach compares to previous work. Some important baseline methods to compare to should include at least:

- i) simple group balancing (see Zhang et al. and Idrissi et al.),
- ii) a minimax approach such as the one proposed by Martinez et al.,
- iii) simple domain invariance in isolation, and
- iv) the dynamic weighted loss in isolation.

Response:

We sincerely appreciate the reviewer’s two important suggestions and have provided results for both the ablation study and comparison experiments with previously proposed unfairness mitigation approaches. Based on the ablation study, it is evident that each loss term significantly contributes to the overall performance of the entire framework. Through comparative experiments, our approach has proven to be optimal in optimizing performance concerning the minority subgroup while maintaining performance concerning the majority subgroup.

a) We’ve provided results of the ablation study (results shown in Supplementary Table 1) as presented below. Our proposed QPI approach incorporates dual optimization objectives, namely multi-task learning and domain adaptation. The results showed that all the components utilized within our QP-Net contribute to the overall performance. We assert the necessity of optimizing components for each distinct objective. An effective implementation of the QPI approach should contain both multi-task learning and domain adaptation components.

Adaptive weight in L_y	Loss			Subgroup (Maj & Min)	AUC				
	$\gamma_d L_d(\omega_x, \omega_d)$	$\gamma_{MMD} L_{MMD}(\omega_x)$	$\gamma_{BSS} L_{BSS}(\omega_x)$		Train	Valid	Test	Test (Maj)	Test (Min)
	δ	$\gamma_d = 0.2$	$\gamma_{MMD} = 1$						
(1)				Papillary & Follicular	0.9788	0.8486	0.8572	0.8804	0.6741
				Papillary & Medullary	0.9327	0.8363	0.8471	0.8847	0.6502
				Tertiary & Community	0.9411	0.8415	0.8489	0.8852	0.6970
(2)	✓			Papillary & Follicular	0.9676	0.8344	0.8525	0.8659	0.7022
				Papillary & Medullary	0.9483	0.8470	0.8244	0.8764	0.6736
				Tertiary & Community	0.9276	0.8478	0.8510	0.8845	0.6459
(3)		✓		Papillary & Follicular	0.9542	0.8279	0.8110	0.8300	0.6736
				Papillary & Medullary	0.9459	0.8324	0.8345	0.8456	0.7145
				Tertiary & Community	0.9483	0.8235	0.8214	0.8378	0.6591
(4)			✓	Papillary & Follicular	0.9532	0.8245	0.8246	0.8323	0.6523
				Papillary & Medullary	0.9422	0.8334	0.8169	0.8223	0.7088
				Tertiary & Community	0.9428	0.8402	0.8449	0.8523	0.6827
(5)	✓	✓		Papillary & Follicular	0.8980	0.8252	0.8412	0.8475	0.7329
				Papillary & Medullary	0.9116	0.8230	0.8381	0.8491	0.7440
				Tertiary & Community	0.9324	0.8308	0.8389	0.8523	0.7157
(6)	✓		✓	Papillary & Follicular	0.9377	0.8414	0.8426	0.8620	0.7003
				Papillary & Medullary	0.9712	0.8091	0.8220	0.8432	0.7145
				Tertiary & Community	0.9739	0.8130	0.8249	0.8414	0.7289
(7)	✓	✓	✓	Papillary & Follicular	0.9382	0.8208	0.8422	0.8621	0.7575
				Papillary & Medullary	0.9005	0.8048	0.8112	0.8374	0.7329
				Tertiary & Community	0.8892	0.8331	0.8213	0.8375	0.7451
(8)	✓	✓	✓	Papillary & Follicular	0.9239	0.8312	0.8423	0.8528	0.7555
				Papillary & Medullary	0.9291	0.8002	0.8199	0.8256	0.7679
				Tertiary & Community	0.9210	0.8293	0.8377	0.8420	0.7761
(9)	✓	✓	✓	Papillary & Follicular	0.9768	0.8784	0.8672	0.8893	0.8098
				Papillary & Medullary	0.9382	0.8621	0.8572	0.8688	0.7832
				Tertiary & Community	0.9091	0.8621	0.8693	0.8772	0.7794

‘Supplementary Table 1: Ablation study for components of the proposed QP-Net. a. (1) ~ (9): The empirical results support that each component contributes to the overall model performance. b. (2) vs. (5): the adversarial structure contributes to model performance on minority subgroups. c. (2) vs. (7): empirical evidence on the effectiveness of marginal domain invariance on our thyroid ultrasound dataset. d. (5) vs. (6) vs. (7) and (8) vs. (9): combining adversarial structure and MMD loss yields better performance. e. (3) vs. (5), and (4) vs. (6): multi-task learning module contributes to preserving model performance on majority subgroup.

f. (5) vs. (8), and (7) vs. (9): BSS loss contributes to the improvement of model performance on minority subgroup.’

b) In response to the various comparable methods including those suggested by the reviewers, we have supplemented the relevant comparative results in Supplementary Table 2.

Type	Methods			Subgroup (Maj & Min)	AUC					
	Name	Data load	Training Process		Train	Valid	Test	Test (Maj)	Test (Min)	
Simple Group Balancing	I. Balanced – ERM ¹	Upsampling minority groups	ERM	Papillary & Follicular	0.9232	0.7412	0.8672	0.8710	0.7423	
				Papillary & Medullary	0.9471	0.8212	0.8512	0.8527	0.7622	
				Tertiary & Community	0.9277	0.7923	0.8591	0.8680	0.7283	
		Ours	Upsampling minority groups	Quasi Pareto	Papillary & Follicular	0.9819	0.8511	0.8631	0.8721	0.7988
					Papillary & Medullary	0.9125	0.8518	0.8512	0.8621	0.7842
					Tertiary & Community	0.9013	0.8710	0.8628	0.8784	0.7512
	II. SUBG ²	Subsampling majority groups	ERM	Papillary & Follicular	0.9459	0.8343	0.8213	0.8452	0.6709	
				Papillary & Medullary	0.9672	0.8500	0.8490	0.8544	0.6862	
				Tertiary & Community	0.9362	0.8322	0.8352	0.8465	0.7060	
		Ours	Subsampling majority groups	Quasi Pareto	Papillary & Follicular	0.9102	0.8523	0.8484	0.8546	0.7772
					Papillary & Medullary	0.9491	0.8600	0.8520	0.8527	0.7592
					Tertiary & Community	0.9441	0.8662	0.8677	0.8755	0.7101
Minimax Approach	III. Minimax Pareto fairness ³	Common	Minimax Pareto Fair Optimization	Papillary & Follicular	0.9667	0.8392	0.8012	0.8256	0.7931	
Domain Invariance	IV. Domain-Adversarial Neural Network ⁴	Common	ERM + Maximize domain classifier loss	Papillary & Medullary	0.9392	0.8101	0.8173	0.8378	0.7693	
				Tertiary & Community	0.9102	0.7972	0.8087	0.8182	0.7771	
				Papillary & Follicular	0.8906	0.8407	0.8381	0.8493	0.7438	
Dynamic Weighted Loss	V. Dynamically Weighted Balanced Loss ⁵	Common	ERM + DWS Loss	Papillary & Medullary	0.9321	0.8205	0.8304	0.8349	0.6779	
				Tertiary & Community	0.8964	0.8417	0.8333	0.8436	0.7417	
				Papillary & Follicular	0.9432	0.8326	0.8438	0.8624	0.7533	
Optimized loss functions	VI. Focal Loss ⁶	Common	ERM + Focal Loss	Papillary & Medullary	0.9121	0.8534	0.8223	0.8552	0.7563	
				Tertiary & Community	0.9372	0.8171	0.8241	0.8439	0.7474	
				Papillary & Follicular	0.9485	0.8578	0.8532	0.8669	0.7200	
Ours	Quasi Pareto	Common	$\mathcal{L}_y(\omega_x, \omega_y)$ + $\mathcal{L}_{\text{class conditional}}$ + $\gamma_{\text{MMD}} \mathcal{L}_{\text{MMD}}(\omega_x)$ + $\gamma_{\text{BSS}} \mathcal{L}_{\text{BSS}}(\omega_x)$	Papillary & Follicular	0.9504	0.8231	0.8425	0.8531	0.7021	
				Papillary & Medullary	0.9231	0.8191	0.8302	0.8462	0.7482	
				Tertiary & Community	0.9011	0.8245	0.8110	0.8366	0.7531	
		Common	$\mathcal{L}_y(\omega_x, \omega_y)$ + $\gamma_{\text{MMD}} \mathcal{L}_{\text{MMD}}(\omega_x)$ + $\gamma_{\text{BSS}} \mathcal{L}_{\text{BSS}}(\omega_x)$	Papillary & Follicular	0.9768	0.8784	0.8672	0.8893	0.8098	
				Papillary & Medullary	0.9382	0.8621	0.8572	0.8688	0.7832	
				Tertiary & Community	0.9091	0.8621	0.8693	0.8772	0.7794	

‘Supplementary Table 2: Method comparison with state-of-art fairness and domain adaptation approaches. The results support that without both components (multi-task learning and domain adaptation) within the proposed QP-Net, existing methods can’t reach fairness in both majority and minority subgroups.’

From the results, it can be observed that methods employed in simple group balancing, as well as the Adversarial method and optimized loss functions method, all exhibit inferior performance on the minority group compared to our QPI approach, indicating that their domain adaptation effects are suboptimal. While the Minimax Pareto method achieves performance close to ours on the minority group, it leads to a significant deterioration in performance on the majority group. This aligns with our argument, as this method incurs a higher payoff on the majority group compared to our QPI approach.

As for Martinez et al., we also provide an argument on the difference between their method and ours. Our method and minimax pareto are both intended to handle the subgroup unfairness problem of AI, and the methods are both somehow related to the concept of “Pareto”, but in different senses. Minimax pareto method aims to effectively move along the pareto front, which is based on the fairness concept regarding model calibration. While improving the model performance of the minority group using the minimax pareto method, the model performance of the majority group has to decrease as a necessary trade-off (by the definition of pareto front). On the contrary, our Quasi-Pareto Improvement framework aims to push the pareto front

forward by optimizing the between-subgroup transfer learning. While improving performance for minority using our QPI approach, there should also be an aim to maintain the performance of the majority group. Since the two methods have distinct aims, we use different weighting strategies in the multi-task objective function. In real-world applications, our QPI approach and the minimax pareto method can be used simultaneously as a combination. For policy decision-makers, our QPI approach is almost always desirable since no one is harmed by our model modification (this is why we call it “Pareto improvement” as conventional concept in sociology and economics). But the application of minimax pareto method would have more complicated ethical considerations on its trade-off between different populations.

Comments:

The analyzed private dataset is certainly impressive in size. Several details remained unclear to me, however.

- I have concluded from the text that probably the numbers in the first column of Fig. 2 refer to patients, whereas the numbers in the later columns (train/val/test) refer to images? This should be clarified.

Response:

Thank you for your careful reading. We have modified the caption for clarification:

Line 79 - 81:

‘Note that numbers in ‘Patient Num’ column refer to patients, while numbers in columns including ‘Train Set’, ‘Valid Set’ and ‘Test Set’ refer to images.’

- The numbers in the "Hospital" category sum to more than the numbers in the "All" category.

Response:

Thank you very much for pointing out the issue, and we’ve rectified it.

In Fig. 2(a), we inadvertently presented the numbers in the "Community Hospital" category incorrectly. In fact, the data within the "Hospital" category all correspond to the PTC subtype. Therefore, the numbers in the "Hospital" category should be summed to match those in the "Papillary Thyroid Cancer" category. We have rectified the numbers in the "Community Hospital" category to ensure their alignment with the actual circumstances In Fig. 2(a):

Line 77-86:

'Fig. 2: Benign-malignant prediction performances ...'

- The authors write that exclusion criteria "such as [...]" were applied. Are there others that were applied that are not listed?

Response:

No, there aren't any other criteria that were applied but not listed. What we listed were the complete exclusion criteria. In the revised manuscript, we have made minor adjustments to enhance clarity.

Line 320 - 324:

'After inclusion, ... as study participants.'

- The authors list three patient inclusion criteria. I suspect patients had to satisfy just one of these? This should be clarified; the current formulation is ambiguous in this regard. Also, what was the initial selection process? Were *all* patients in participating institutions selected that satisfied the inclusion criteria and not the exclusion criteria? Or was there some prior selection process?

Response:

Thank you for pointing this out, and we have modified the text for clarification.

In the revised manuscript, we have refined the formulation of the inclusion criteria and the exclusion criteria. Patients had to satisfy just one of the two inclusion criteria to be included, and patients satisfying any one of the five exclusion criteria were excluded.

There was no prior selection process, and we initially screened all patients with thyroid nodules in electronic medical records at multiple hospitals.

The modified passage explaining the inclusion criteria is in line 315 - 320:

'We gathered a 10-year dataset ... from experienced radiologists.'

- Did the authors ensure that there was no patient leakage between the train/val/test sets?

Response:

During the partitioning of the training, validation, and testing datasets, we ensured that there was no patient overlap among these sets. Specifically, the data of a particular patient would only exist in one of the three subsets: train, valid, or test. We have also incorporated a corresponding description in the revised manuscript to emphasize this aspect:

Line 332 - 334:

'It should be noted that although some patients may have more than one corresponding ultrasound image, during the partitioning of the training, validation, and testing datasets, we ensured that there was no patient leakage between the train, valid and test sets.'

- The authors write that they "use nnU-net to automatically crop each image retaining the main portion of it". How precisely do they do this?

Response:

The term "Main portion" refers to the section displaying only the thyroid ultrasound image. We have added the corresponding experimental details and results in Supplementary materials. Supplementary Materials Line 47 - 60:

'We employed the Attention ... a correction ratio of 4.27%.'

In the main text, within the "Methods" section, we have also revised the description of this step. The added content is in:

Line 325 - 327:

'To prevent the interference of textual ... information at the edges.'

Line 329 - 331:

'Each image is scaled to 512×512 pixels ... Supplementary details.'

- The authors write that "for irregular images, a background with pixel value 0 is added". What does this mean? What is an "irregular image"?

Response:

Irregular image refers to an image with irregular edges. After cropping by nnU-net, images often exhibit irregular edges. Before utilizing these images with irregular edges as model inputs, we augmented them with a background with pixel values set to 0.

In the revised manuscript, we have revised the phrasing. The modified content is as follows:

Line 327 - 329:

'While nnU-net achieves a high success rate ... pixel values set to 0.'

Comments:

Some important aspects of the fundamental setup of the prediction task also remained unclear to me.

1) I could not figure out the fundamental prediction task that the authors are solving. Is it binary classification (malignant/benign)? Or multi-label (different thyroid nodule subtypes)? In other words, what is "Y" in the Problem formulation section?

Response:

For thyroid data, the primary prediction task is binary classification between benign and malignant cases. The various thyroid subtypes were considered as covariants that can influence the prediction performance.

Therefore, in the formulation section, "Y" refers to the malignant / benign class label. In the revised manuscript, we have incorporated a clarification regarding this information. The revised passage is presented below:

Line 341:

'For our thyroid ultrasound image dataset, we have $Y = \{Benign, Malignant\}$.'

Furthermore, we have supplemented this information in the captions of Figure 2 and Figure 4 to enhance comprehensibility. The revised captions are provided below:

Line 77:

'Fig. 2: Benign-malignant prediction performances for imbalanced subgroups. ...'

Line 152:

'Fig. 4: Benign-malignant prediction performances of the QPI approach. ...'

2) Similarly, what is "D" / which domains do the authors consider? All possible combinations of the different subgroups, i.e., would "female, 21-35y, small nodule, PTC, Community Hospital" be one of the considered domains, and are the domain classifier and MMD terms then distinguishing between all of these combinations? Or are there different domain classifiers

acting on the Male vs. Female, Tertiary vs. Community Hospital etc. axes? Or something else entirely? In Eq. (11), what are D_{min} and D_{maj} ?

Response:

Our QPI approach focuses on a single subgrouping scheme at a time. Under a specific subgrouping scheme, for example, histological subtypes in the context of thyroid ultrasound data, if we observe unfairness between PTC and FTC, then $\mathcal{D} = \{PTC, FTC\}$. The majority and minority subgroups are determined based on their relative sizes.

In the revised manuscript, we have made revisions within our problem formulation to enhance comprehensibility. The modified description is presented below:

line 337 - 340:

'This study is conducted to address a diagnostic problem that aligns with clinical practice, which we have described as follows. Given a dataset $D :: X \times Y$, where X is the instance set and Y is the corresponding class label (diagnosis) space, we denote by D_{maj} and D_{min} the majority and minority set under a specific subgrouping scheme on D , with $|D_{maj}| > |D_{min}|$. D_{maj} and D_{min} share the same Y .'

In real-world settings, there would be multiple different subgrouping schemes that apply to a certain dataset, and we would observe the existence of model unfairness under several subgrouping schemes. This allows the QPI approach to be applied sequentially under each of these schemes to mitigate the unfairness problem. In the Discussion, we have incorporated an explanation that addresses unfairness scenarios involving multiple subgrouping schemes.

Line 299 - 302:

'Also, more complicated scenarios ... clarify underlying mechanisms.'

3) The authors write that hyper parameters were "determined through experiments". Could they be a bit more specific? How were they determined, and on which dataset (train/val/test) were the corresponding experiments performed?

Response:

We have supplemented the details regarding hyperparameters in the Supplementary Materials, where we have presented a comparative analysis of selecting different hyperparameters on the thyroid subtype data. In the revised manuscript, we refer to the details in supplementary materials:

Line 474:

'Training details are provided ... Supplementary details.'

Supplementary Materials Line 20 - 30:

Hyperparameter comparison (γ)			Subgroup (Maj & Min)	AUC				
γ_d	γ_{NMD}			Train	Valid	Test	Test (Maj)	Test (Min)
(1)	0	0	Papillary & Follicular	0.8980	0.8252	0.8412	0.8475	0.7329
			Papillary & Medullary	0.9116	0.8430	0.8381	0.8491	0.7440
			Tertiary & Community	0.9324	0.8508	0.8389	0.8523	0.7157
(2)	0.1	0	Papillary & Follicular	0.9790	0.8466	0.8460	0.8617	0.7189
			Papillary & Medullary	0.9871	0.8505	0.8524	0.8688	0.7236
			Tertiary & Community	0.9909	0.8413	0.8495	0.8665	0.6950
(3)	0.2	0	Papillary & Follicular	0.9887	0.8278	0.8416	0.8597	0.6967
			Papillary & Medullary	0.9732	0.8305	0.8381	0.8502	0.7472
			Tertiary & Community	0.9391	0.8179	0.8080	0.8243	0.6813
(4)	0.3	0	Papillary & Follicular	0.9732	0.8305	0.8381	0.8502	0.7472
			Papillary & Medullary	0.9472	0.8360	0.8288	0.8378	0.7569
			Tertiary & Community	0.9759	0.8354	0.8329	0.8453	0.7419
(5)	0.4	0	Papillary & Follicular	0.9194	0.8247	0.8334	0.8512	0.7086
			Papillary & Medullary	0.9607	0.8436	0.8416	0.8604	0.6823
			Tertiary & Community	0.9825	0.8454	0.8496	0.8713	0.6756
(6)	0	1	Papillary & Follicular	0.9792	0.8403	0.8429	0.8586	0.7111
			Papillary & Medullary	0.9806	0.7755	0.8423	0.8618	0.7017
			Tertiary & Community	0.9812	0.8482	0.8466	0.8624	0.6827
(7)	0.1	1	Papillary & Follicular	0.9079	0.8641	0.8519	0.8800	0.7722
			Papillary & Medullary	0.9069	0.8403	0.8552	0.8762	0.7557
			Tertiary & Community	0.9479	0.8598	0.8527	0.8686	0.7296
(8)	0.2	1	Papillary & Follicular	0.9768	0.8784	0.8672	0.8893	0.8098
			Papillary & Medullary	0.9382	0.8621	0.8572	0.8688	0.7832
			Tertiary & Community	0.9091	0.8621	0.8693	0.8772	0.7687
(9)	0.3	1	Papillary & Follicular	0.9631	0.8377	0.8509	0.8641	0.7641
			Papillary & Medullary	0.9079	0.8341	0.8319	0.8435	0.7559
			Tertiary & Community	0.9069	0.8303	0.8325	0.8464	0.7157
(10)	0.4	1	Papillary & Follicular	0.9938	0.8060	0.8139	0.8514	0.7506
			Papillary & Medullary	0.9634	0.8341	0.8382	0.8562	0.7490
			Tertiary & Community	0.9321	0.8352	0.8499	0.8623	0.7578
(11)	0	2	Papillary & Follicular	0.9948	0.8162	0.8023	0.8512	0.6453
			Papillary & Medullary	0.9859	0.8023	0.8376	0.8500	0.6479
			Tertiary & Community	0.9050	0.8299	0.8150	0.8516	0.6500
(12)	0.1	2	Papillary & Follicular	0.8907	0.8247	0.8252	0.8467	0.6373
			Papillary & Medullary	0.8907	0.8241	0.8247	0.8463	0.6369
			Tertiary & Community	0.8901	0.8217	0.8241	0.8450	0.6433
(13)	0.2	2	Papillary & Follicular	0.8817	0.8313	0.8340	0.8475	0.7101
			Papillary & Medullary	0.8866	0.8308	0.8335	0.8471	0.6442
			Tertiary & Community	0.8850	0.8309	0.8338	0.8473	0.7109
(14)	0.3	2	Papillary & Follicular	0.9414	0.8299	0.8319	0.8487	0.6803
			Papillary & Medullary	0.9407	0.8324	0.8315	0.8499	0.6716
			Tertiary & Community	0.9186	0.8265	0.8264	0.8467	0.6463
(15)	0.4	2	Papillary & Follicular	0.9382	0.8353	0.8328	0.8393	0.5377
			Papillary & Medullary	0.9732	0.8055	0.8032	0.8300	0.6792
			Tertiary & Community	0.9412	0.8325	0.8380	0.8485	0.7485

‘Supplementary Table 3: Hyperparameter comparison results of all three pairs of subgroups on our thyroid ultrasound dataset. Hyperparameters used in the experiment are determined based on empirical results, and the optimal hyperparameter combination yielded the best model performance on all three pairs of subgroups.’

Hyperparameter comparison (γ)				Subgroup (Maj & Min)	AUC				
σ	γ_d	γ_{NMD}	γ_{BBS}		Train	Valid	Test	Test (Maj)	Test (Min)
(1)	0.01			Papillary & Follicular	0.9423	0.8553	0.8225	0.8374	0.7623
				Papillary & Medullary	0.9422	0.8534	0.8373	0.8423	0.7540
				Tertiary & Community	0.9534	0.8463	0.8255	0.8423	0.7657
(2)	0.05			Papillary & Follicular	0.9768	0.8784	0.8672	0.8893	0.8098
				Papillary & Medullary	0.9382	0.8621	0.8572	0.8688	0.7832
				Tertiary & Community	0.9091	0.8621	0.8693	0.8772	0.7794
(3)	0.1			Papillary & Follicular	0.9342	0.8653	0.8553	0.8797	0.6367
				Papillary & Medullary	0.9530	0.8564	0.8581	0.8602	0.6772
				Tertiary & Community	0.9535	0.8554	0.8503	0.8643	0.6834
(4)	0.01			Papillary & Follicular	0.9727	0.8432	0.8400	0.8450	0.7232
				Papillary & Medullary	0.9827	0.8676	0.8423	0.8634	0.7540
				Tertiary & Community	0.9762	0.8611	0.8354	0.8644	0.7435
(5)	0.5			Papillary & Follicular	0.9748	0.8625	0.8643	0.8743	0.7020
				Papillary & Medullary	0.9727	0.8332	0.8145	0.8442	0.7321
				Tertiary & Community	0.9861	0.8655	0.7934	0.8223	0.6981
(6)	0.1			Papillary & Follicular	0.9846	0.7632	0.7533	0.7932	0.6428
				Papillary & Medullary	0.9625	0.7653	0.7523	0.7943	0.6782
				Tertiary & Community	0.9542	0.7540	0.7356	0.7534	0.6550

‘Supplementary Fig. 1: Hyperparameter comparison experiments of σ . (a) σ governs the extent of minor weight adjustments, and larger σ values yield a smoother $\delta(A_{maj}, A_{mix})$ curve. (b) Empirical results demonstrate that $\sigma=0.05$ emerges as a preferable parameter selection.’

Comments:

While the unfairness mitigation approach proposed by the authors does not appear implausible to me, I would like to see a more convincing motivation for this particular approach, and empirical evidence of the utility of the individual components.

1) It is known that marginal domain invariance approaches have drawbacks - in particular, they require "equalizing" disease type distributions between domains, which may be detrimental concerning rare subtypes. See, e.g., Tachet des Combes et al., Zhao et al., or Li et al. I would like to see, firstly, an argument for why the authors believe that this will still be useful to do here, secondly, whether they considered using class-conditional invariance instead, and thirdly, empirical evidence that it actually helps. The latter would be provided by an ablation experiment.

Response:

We appreciate the suggestions from the reviewer. Our response would follow the three steps within the suggestions.

(1) In the literature, the comparison between Marginal Domain Invariance and Class-Conditional Invariance domain adaptation techniques primarily applies to unsupervised scenarios, as seen in Tachet des Combes et al. and Li et al. Since our approach pertains to a supervised setting, we did not initially consider class-conditional invariance.

(2) We sincerely appreciate the reviewer's advice about trying class-conditional invariance, we agree it might potentially improve our algorithm, and we've tested a representative class-conditional invariance approach on our thyroid ultrasound dataset, as shown in Supplementary Table 2. However, in our specific context, class-conditional invariance did not yield better performance compared to Marginal domain invariance. One plausible explanation for this result could be that when optimizing a supervised classifier, the feature spaces of different domains with the same class label may naturally attempt to align themselves as closely as possible.

(3) We've also provided empirical evidence in Supplementary Table 1 ablation study. On our thyroid dataset, the model performance noticeably diminishes when any domain invariance component is excluded. However, Zhao et al. illustrated that using domain-invariant representations could result in a potential loss of accuracy if the domain index contains discriminative information about the target attribute. Indeed, it can be noticed that the effectiveness of marginal invariance and class-conditional invariance approaches differs in different real-world scenarios. Hence, we recommend that future implementation of the QPI approach should be realized according to the specific prediction problems.

2) Both adversarial domain classification and the MMD loss serve the same goal: enforcing (marginal) domain invariance. Why do the authors use both at the same time? Again, empirical evidence that this is actually helpful would be appreciated.

Response:

We sincerely thank the reviewer for making the point. In our case, we did not delve into such considerations extensively; rather, we simply chose the best loss based on our experiments. In previous research, combining two losses with similar objectives has shown the potential for achieving improved results.

Our main point is that to achieve the objective of QPI, it is essential to employ a multi-task learning component to balance the losses associated with both the majority and minority

classes and simultaneously a domain adaptation component to enhance feature transfer for both the majority and minority classes. The specific choice of loss functions may need to be tailored to the specific case at hand.

The variability of the two components is evident through the comparison of ablation study results provided in Supplementary Table 1, specifically (2) vs. (5) vs. (6). Additionally, by comparing the results of (5), (6), and (7), it becomes apparent that combining both components would yield an enhanced overall performance of the framework across various scenarios, exhibiting complementary effects.

3) Concerning the dynamic weighted loss, the authors write that " g_δ satisfied the following requirements..." but they never actually define this function. Moreover, the corresponding reference to Chen et al. is incomplete; the paper title and any other identifiers (DOI, URL, ...) are missing.

Response:

Regarding g_δ , in our experiments, we adaptively adjust its value based on the testing AUC of the model on different subgroups after each training epoch.

In the revised manuscript, we have removed ' g_δ ', simply using the notation δ , and we have included details of the practical adjustment of δ :

Line 435 - 446:

'The value of δ is varied and ... underlying subgroup feature distributions.'

Regarding the reference, we have removed this erroneous reference from the manuscript. As explained in our response to previous questions, we've made a terminological revision in the article, replacing the designation of the component corresponding to L_y with 'multi-task learning loss' instead of 'dynamic weighted loss', as there are fundamental differences between our design and the previously proposed method. The revised text is as follows:

Line 11-12:

'Our tailored deep learning ... multi-task learning with domain-adaptation.'

Line 129:

'A multi-task learning structure is incorporated in the label predictor ...'

Line 242 - 244:

'Our implementation, termed QP-Net, ... in imbalanced subgroups.'

Line 432 - 433:

'For \mathcal{L}_y , we form a multi-task learning structure, ... reweight their losses.'

4) What motivated the authors to add the BSS loss (and not any other standard regularization term)? Again, empirical evidence of its utility in the given context would be appreciated.

Response:

The addition of the BSS loss significantly improves the performance of the framework across various scenarios, according to ablation study results in Supplementary Table 1. By comparing (5) vs. (8), and (7) vs. (9), performance of the QP-Net is improved by BSS loss on both majority and minority. Therefore, the effectiveness of BSS loss in our framework can be verified empirically.

We employ the BSS loss to restrain detrimental knowledge from the source domain during training, aiming to enhance the model's learning efficiency for transferable features. The

functionality of the proposed QPI approach is not particularly reliant on specific regularization terms, so other regularization terms are alternative if effective as well. In the discussion section, we have added relevant content as follows:

Line 302 - 314:

'Furthermore, this study adopts BSS loss ...further investigation in future work.'

Comments:

Besides the clarity-related issues already outlined above, I consider the manuscript's engagement with prior literature to be severely lacking. In particular, very little credit is given to the extremely extensive prior literature on domain adaptation and algorithmic fairness (including in medical imaging). In some places, it reads as if the authors claim novelty on the methodological side.

Oakden-Rayner et al. discuss problems resulting from the aggregation of different disease subtypes into a single label, closely related to the authors' consideration of rare disease subtypes. Adversarial and MMD-based domain invariance approaches have been used extensively for over a decade; see some of the references provided above.

Mukherjee et al. discuss the relationship between domain adaptation and algorithmic fairness approaches. The use of group-invariant representations is a standard approach in the algorithmic fairness literature and has been known since at least Zemel et al. (2013). Thus, I would strongly contest the authors' claim of "innovation" on this point.

Response:

We sincerely appreciate the reviewer's comment. In the revised manuscript, we have included discussions regarding related work in fairness and fairness mitigation within the medical domain:

Line 269 – 285:

'In this study, the QPI approach was ... within the majority population.'

Meanwhile, in our previous responses, we also elucidated the primary innovation related to the loss function in this study. Essentially, our QPI approach design integrates both a multi-task learning module and a domain adaptation module, aiming to mitigate unfairness according to the proposed Pareto Fairness concept. While the implementations of the QPI approach may vary in future research due to the diversity of problems in reality, this study asserts that both modules are essential for addressing unfairness issues within the framework of Pareto Fairness criteria.

We appreciate the reviewer for providing relevant literature. In this regard, we have cited Oakden-Rayner et al. (2020) in the Introduction (line 60), and have included Mukherjee et al. (2022) in the Discussion section (line xx). Please note that Mukherjee et al. (2022) focused on individual fairness defined by L-Lipschitz continuity, which differs from the Pareto fairness addressed in this study.

On our thyroid data set, we've also provided empirical evidence in Supplementary Table 2 that the devised QP-Net have optimal performance comparing with previous fairness mitigation and domain adaptation methods. For future research, we recommend selecting the most suitable method based on the specific problem and data at hand.

The authors claim in various places that their method is unsupervised, but it is not clear if that

is true. It seems that both domain labels and class labels are required for all samples in the authors' method?

Response:

We appreciate the reviewer's comments and have modified the text to avoid confusion. In the revised manuscript, we removed the word 'unsupervised' when introducing our domain adaptation approaches, as all our experiments were conducted in a fully supervised setting.

In the previous manuscript, we twice mentioned when introducing the domain classifier that this component can achieve unsupervised domain adaptation, which was actually true but not relevant to this study.

Finally, there is a fundamental difference between, on the one hand, demographic patient groups and, on the other hand, different disease subtypes (representing output classes). I suggest that the authors clearly distinguish between these, and not summarize them under the single term "groups".

Response:

We sincerely appreciate the reviewer's suggestions. We have clarified the term accordingly. In our revised manuscript, we have transitioned away from using the generic term 'groups' and have opted for the term 'different subgrouping schemes' to precisely describe them, and use 'histological subtypes' and 'hospitals' to refer to the two subgrouping schemes respectively.

In this study, separate sets of experiments for each subgrouping scheme were conducted, as unfairness in terms of AUC disparity and learning curves were observed for both subgrouping schemes. It can be noticed that our QPI approach demonstrates efficacy across these two distinct subgrouping schemes with differing medical implications. This effectively underscores the adaptability of our framework to various subgrouping schemes.

Other clarity issues:

- For the experiments presented in the graphs in fig. 4, what is the corresponding minority dataset size? How exactly were these experiments performed, i.e., how were the extra images to add selected?

Response:

The corresponding minority dataset sizes are added in the revised legend in Fig. 4 (d).

Experimental details: All the training samples of the minority subgroup were used. For example, in the PTC (majority) vs. FTC (minority) experiment, both PTC and FTC datasets were partitioned into training, validation and test subsets, with the validation and test set sizes maintained uniformly throughout the experiment. Within the experiment, all the training FTC samples are retained in the training set, while varying quantities of training PTC samples are iteratively added to the training set, leading to the training of separate models. The learning curve graph assessed how different majority sample sizes impact the model's performance on both majority and minority data.

- The whole "CKA Explanation" section was very opaque to me. Are feature distributions compared between different *networks* or between different *patient groups* here? What are "CKA lines", what do the arrows in Fig. 5(a) and (b) between the different "lines" mean, and what are all the axis labels in that figure?

Response:

We appreciate the reviewer's comments. We've realized that the original Fig. 5 and the corresponding explanation in the manuscript is opaque, so we replaced them with a more concise and comprehensible rendition.

The CKA section is used to illustrate the underlying mechanism of the effectiveness of our QPI approach. In the Discussion we've added content to better explain this issue.

Line 286 - 296:

'This investigation underscores a conundrum ... the propitious ones.'

The utilization of CKA in this study is grounded in prior research, notably the work of Kornblith et al. (2019), which elucidates the concept and utility of CKA, and Neyshabur et al. (2020), who applied CKA to quantify feature reuse in transfer learning. In the revised manuscript we've made clear reference to these studies.

We have added detailed explanations of CKA in the "Methods" section:

Line 476 - 498:

'Centered Kernel Alignment (CKA)

To quantitatively compare ... under certain transfer learning strategies.'

We have also modified our presentation of the CKA results with enhanced comprehensibility. The revised content is provided in the Results section: Locate the unfairness layer within the neural network.

Line 157 - 163:

'The learning curve experiment revealed ... Centered Kernel Alignment (CKA).'

Line 166 - 173:

'Fig. 5: CKA method assess ... in the higher layers.'

Line 175- 183:

'Fig. 5(b) depicts ΔCKA_{mix} and ΔCKA_{QPI} ... final layer of neural network.'

Comments:

-- In Figs. 6 and 7, how were the specific subgroups to show here selected? Were trends comparable in other subgroups?

Response:

According to existing literature, significant differences were reported in the experimental results between the male and female populations. To enhance the predictive performance for the female population, this study solely validated the effectiveness of Quasi Pareto Improvement on the female subgroup.

In line with our analysis of thyroid cancer, we conducted a proportion change analysis to address the issue of imbalanced data for both public datasets. The empirical results illustrate that in most circumstances, with an increase in the majority subgroup size, the predictive fairness for the minority subgroup remains improving, signifying a substantial enhancement in fairness. The corresponding results have been incorporated into Figure 6 and Figure 7, with their caption revised as well:

Method	Train (AUC)		Subgroup	Average	Test (AUC)														
	AUC	AUC			NoFind	EnlargC	Carlini	LungO	LungL	Edema	ConsId	Pneum	Atelect	PneumoB	PneumoE	PneumoF	Fracture	SepDi	
Baseline	0.919	(0.796 - 0.884)	18-40	0.814	0.842	0.863	0.791	0.746	0.862	0.725	0.731	0.718	0.838	0.399	0.871	0.763	0.888		
Quasi-Parceto	0.985	0.816	40-60	0.814	(0.792 - 0.872)	0.876	0.686	0.879	0.792	0.756	0.869	0.755	0.723	0.720	0.869	0.900	0.871	0.787	0.872
QP	0.994	0.799	60-80	0.877	0.705	0.864	0.737	0.766	0.831	0.758	0.758	0.704	0.868	0.884	0.788	0.799	0.874		
Baseline			-80	0.847	(0.824 - 0.886)	0.876	0.715	0.864	0.798	0.768	0.837	0.726	0.760	0.736	0.851	0.882	0.776	0.773	0.873
Quasi-Parceto			Asian	0.853	(0.765 - 0.877)	0.874	0.666	0.845	0.714	0.780	0.812	0.760	0.745	0.691	0.870	0.868	0.823	0.722	0.877
QP			Black	0.765	(0.742 - 0.819)	0.830	0.655	0.799	0.684	0.743	0.813	0.668	0.700	0.634	0.837	0.872	0.770	0.731	0.884
Baseline			White	0.801	(0.756 - 0.842)	0.847	0.689	0.834	0.682	0.772	0.823	0.683	0.751	0.665	0.898	0.851	0.814	0.768	0.897
Quasi-Parceto			Other	0.795	(0.741 - 0.842)	0.878	0.714	0.868	0.731	0.728	0.833	0.783	0.693	0.679	0.888	0.887	0.704	0.729	0.891
QP				0.805	(0.769 - 0.861)	0.893	0.708	0.884	0.743	0.758	0.863	0.793	0.793	0.706	0.888	0.793	0.757	0.793	0.793
Baseline				0.746	(0.720 - 0.813)	0.813	0.663	0.810	0.760	0.758	0.835	0.724	0.678	0.685	0.870	0.848	0.737	0.716	0.815
Quasi-Parceto				0.803	(0.789 - 0.866)	0.873	0.683	0.847	0.722	0.776	0.857	0.745	0.717	0.749	0.898	0.892	0.805	0.747	0.907
QP				0.830	(0.787 - 0.853)	0.880	0.682	0.842	0.729	0.787	0.826	0.735	0.799	0.734	0.865	0.870	0.810	0.786	0.870
Baseline				0.836	(0.780 - 0.842)	0.891	0.695	0.831	0.743	0.795	0.841	0.741	0.742	0.697	0.882	0.885	0.822	0.801	0.889
Quasi-Parceto				0.802	(0.761 - 0.863)	0.869	0.688	0.864	0.764	0.720	0.826	0.728	0.727	0.698	0.868	0.883	0.794	0.721	0.883
QP				0.856	(0.775 - 0.850)	0.891	0.648	0.870	0.745	0.740	0.824	0.728	0.733	0.738	0.864	0.888	0.765	0.715	0.876

Line 222 - 223:

‘(d) Learning curves ... effectiveness of QPI approach.’

Method	Train (AUC)		Subgroup	Average	Test (AUC)									
	AUC	AUC			MEL	NV	BCC	AK	BKL	DF	VASC	SCC	UNK	
Baseline	0.918	(0.872 - 0.965)	Male	0.918	0.938	0.957	0.951	0.930	0.903	0.957	0.997	0.813	0.858	
Quasi-Parceto	0.987	0.923	Female	0.922	(0.880 - 0.958)	0.934	0.956	0.951	0.931	0.905	0.948	0.995	0.857	0.864
QP	0.962	0.932	0-59	0.878	(0.846 - 0.931)	0.823	0.852	0.929	0.858	0.832	0.898	0.948	0.892	0.837
Baseline			60-85	0.899	(0.842 - 0.948)	0.869	0.863	0.942	0.889	0.863	0.886	0.927	0.932	0.824
Quasi-Parceto				0.897	(0.850 - 0.946)	0.898	0.925	0.942	0.935	0.893	0.911	0.963	0.969	0.901
QP				0.902	(0.865 - 0.949)	0.891	0.912	0.926	0.936	0.918	0.929	0.961	0.946	0.909
Baseline				0.826	(0.801 - 0.862)	0.851	0.866	0.845	0.737	0.830	0.904	0.888	0.854	0.822
Quasi-Parceto				0.884	(0.820 - 0.897)	0.895	0.901	0.866	0.834	0.913	0.923	0.939	0.869	0.853

Line 234 - 235:

‘(d) Learning curves ... effectiveness of QPI approach.’

It is worth noting that when there is an imbalance in sample sizes within subgroups, it is common practice to initially subject the model to performance fairness evaluation experiments.

Upon identifying pronounced predictive performance disparities or discernible reductions in expressive power for the minority samples with the rise in the larger sample group, the Quasi Pareto Improvement method is then applied to enhance fairness. We have included relevant descriptions in the discussion section as follows:

Line 305- 311:

'The QPI approach can be broadly ... engaging in in-depth discussions.'

- In the discussion, the authors write that "prediction differences might exist even if the proportions of subgroup data are identical. In other words, that proportional imbalance may not necessarily result in prediction differences." - Both individual statements are correct, but the second statement does not follow from the first one.

Response:

We appreciate the reviewer's comment. In the revised manuscript we've removed this statement as it is less relevant to the topic of this study.

Our original intention was to convey that prediction differences might not solely be attributed to proportional imbalance but could also arise from other underlying factors: 'Moreover, we discovered through experiments that prediction differences might exist even if the proportions of subgroup data were identical. In other words, prediction differences may not necessarily result from proportional imbalance.'

Comments

- When / where is Eq. (24) used? Also, I assume this is taken from some standard reference book that should be cited?

Response:

In the revised manuscript, the original equation has been moved to Supplementary Materials Eq. 28. It represents a standard confidence interval formula. In Figure 2 (a) and Figure 4 (a), the column "Estimate (95% CI)" provides the confidence intervals for the model AUC under the respective conditions.

Within the revised manuscript, we've included the source reference for this equation, as detailed below:

Supplementary materials Line 91 - 94:

'Then, ... with $n - 1$ degrees of freedom⁸.'

Supplementary materials Line 134:

'8. Ci, B., & Rule, R. O. (1987). Confidence intervals. Lancet, 1(8531), 494-7.'

===== Suggested improvements =====

In conclusion, I would suggest the following major improvements:

1. Addition of and comparison to a much wider range of baseline methods.
2. Addition of an ablation study that assesses the importance and utility of the four different loss function components.
3. Stronger engagement with and clearer referencing of prior work throughout the manuscript, in particular concerning algorithmic fairness and domain adaptation approaches.
4. Improvement of the clarity (and completeness) of presentation.

Response:

We sincerely appreciate the reviewer's comments and suggestions. We have performed all of these in addressing your comments.

Reviewer #2 (Remarks to the Author):

Summary

The authors propose a novel framework to reduce bias in a classification problem where performance drops for minority subgroups, addressing a central problem related to the area of fairness in AI. This is formulated in the context of thyroid ultrasound nodules, although the methodology is presented in a generic way, and the application in hand is a proof of concept of the framework.

The method performs very well compared to a naive approach (basically, not doing anything about the minorities) and to a simple approach (producing a different model for each subgroup), improving the performance in the minority groups while maintaining overall performance. The results hold for two other datasets that are tested.

The main issue is that there is very limited analysis of related methods in AI in healthcare or AI in medical imaging to put the proposed Quasi Pareto Improvement method in the context of the related literature; this should be clearly investigated in a "related work" section, and also the state-of-the-art methods for supervised bias reduction should be included in the quantitative analysis. Without this, it is difficult to understand the potential impact of the method and its relevance beyond thyroid nodule ultrasound imaging, so I believe this additional set of experiments, with the corresponding statistical analysis is needed; else, authors need to argue why there is no suitable methodology related to fairness in AI to compare the proposed method to.

Response:

In revised manuscript, we've added related work in the Discussion section, introducing previously proposed unfairness mitigation methods and domain adaptation methods:

Line 269 - 285:

'In this study, the QPI approach ... within the majority population.'

Among related work, several existing methods with comparability include simple group balancing (balanced-ERM, SUBG), Minimax approach (Minimax Pareto fairness), Adversarial approach (Adversarial discriminator network), and Optimized loss functions (Focal Loss). We have included comparative experiments on these comparable methods in Supplementary Table 2. The results indicate that our method outperforms the others in terms of performance on both majority and minority groups.

Type	Methods			Subgroup (Maj & Min)	AUC				
	Name	Data load	Training Process		Train	Valid	Test	Test (Maj)	Test (Min)
Simple Group Balancing	I. Balanced – ERM ¹	Upsampling minority groups	ERM	Papillary & Follicular	0.9232	0.7412	0.8672	0.8710	0.7423
				Papillary & Medullary	0.9471	0.8212	0.8512	0.8527	0.7622
				Tertiary & Community	0.9277	0.7923	0.8591	0.8680	0.7283
	Ours	Upsampling minority groups	Quasi Pareto	Papillary & Follicular	0.9819	0.8511	0.8631	0.8721	0.7988
				Papillary & Medullary	0.9125	0.8518	0.8512	0.8621	0.7842
				Tertiary & Community	0.9013	0.8710	0.8628	0.8784	0.7512
	II. SUBG ²	Subsampling majority groups	ERM	Papillary & Follicular	0.9459	0.8343	0.8213	0.8452	0.6709
				Papillary & Medullary	0.9672	0.8500	0.8490	0.8544	0.6862
				Tertiary & Community	0.9362	0.8322	0.8352	0.8465	0.7060
	Ours	Subsampling majority groups	Quasi Pareto	Papillary & Follicular	0.9102	0.8523	0.8484	0.8546	0.7772
				Papillary & Medullary	0.9491	0.8600	0.8520	0.8527	0.7592
				Tertiary & Community	0.9441	0.8662	0.8677	0.8755	0.7101
Minimax Approach	III. Minimax Pareto fairness ³	Common	Minimax Pareto Fair Optimization	Papillary & Follicular	0.9667	0.8392	0.8012	0.8256	0.7931
Domain Invariance	IV. Domain-Adversarial Neural Network ⁴	Common	ERM + Maximize domain classifier loss	Papillary & Medullary	0.9392	0.8101	0.8173	0.8378	0.7693
				Tertiary & Community	0.9102	0.7972	0.8087	0.8182	0.7771
				Papillary & Follicular	0.8906	0.8407	0.8381	0.8493	0.7438
Dynamic Weighted Loss	V. Dynamically Weighted Balanced Loss ⁵	Common	ERM + DWS Loss	Papillary & Medullary	0.9321	0.8205	0.8304	0.8349	0.6779
				Tertiary & Community	0.8964	0.8417	0.8333	0.8436	0.7417
				Papillary & Follicular	0.9432	0.8326	0.8438	0.8624	0.7533
Optimized loss functions	VI. Focal Loss ⁶	Common	ERM + Focal Loss	Papillary & Medullary	0.9121	0.8534	0.8223	0.8552	0.7563
				Tertiary & Community	0.9372	0.8171	0.8241	0.8439	0.7474
				Papillary & Follicular	0.9485	0.8578	0.8532	0.8669	0.7200
Ours	Quasi Pareto	Common	$\mathcal{L}_y(\omega_x, \omega_y)$ $-\mathcal{L}_{\text{Class conditional}}$ $+ Y_{MMD} \mathcal{L}_{MMD}(\omega_x)$ $+ Y_{BSS} \mathcal{L}_{BSS}(\omega_x)$	Papillary & Follicular	0.9504	0.8231	0.8425	0.8531	0.7021
				Papillary & Medullary	0.9231	0.8191	0.8302	0.8462	0.7482
				Tertiary & Community	0.9011	0.8245	0.8110	0.8366	0.7531
		Common	$\mathcal{L}_y(\omega_x, \omega_y)$ $- \gamma_d \mathcal{L}_d(\omega_x, \omega_d)$ $+ Y_{MMD} \mathcal{L}_{MMD}(\omega_x)$ $+ Y_{BSS} \mathcal{L}_{BSS}(\omega_x)$	Papillary & Follicular	0.9768	0.8784	0.8672	0.8893	0.8098
				Papillary & Medullary	0.9382	0.8621	0.8572	0.8688	0.7832
				Tertiary & Community	0.9091	0.8621	0.8693	0.8772	0.7794

‘Supplementary Table 2: Method comparison with state-of-art fairness and domain adaptation approaches. The results support that without both components (multi-task learning and domain adaptation) within the proposed QP-Net, existing methods can’t reach fairness in both majority and minority subgroups.’

As for Martinez et al., we also provide an argument on the difference between their method and ours. Our method and minimax pareto are both intended to handle the subgroup unfairness problem of AI, and the methods are both somehow related to the concept of “Pareto”, but in different senses. Minimax pareto method aims to effectively move along the pareto front, which is based on the fairness concept regarding model calibration. While improving the model performance of the minority group using the minimax pareto method, the model performance of the majority group has to decrease as a necessary trade-off (by the definition of pareto front). On the contrary, our Quasi-Pareto Improvement framework aims to push the pareto front forward by optimizing the between-subgroup transfer learning. While improving minority using our QPI approach, the performance of majority group is not required to decrease. Since the two methods have distinct aims, we use different weighting strategies in the multi-task objective function. In real-world applications, our QPI approach and the minimax pareto method can be used simultaneously as a combination. For policy decision-makers, our QPI approach is almost always desirable since no one is harmed by our model modification (this is why we call it “Pareto improvement” drawing from conventional concept in sociology and economics). However the application of minimax pareto method would have more complicated ethical considerations on its trade-off between different populations.

Other minor comments are as follows:

1. I would suggest some reorganization of the text to inform the reader early on that this is study

is focused on thyroid nodules, and then broaden the scope to explain what the impact of the work is beyond this application, in the context of the general literature of fairness in AI. Currently the opposite is done.

Response:

Thank you very much for your feedback. We've reorganized the introduction section accordingly.

Line 26 - 68:

'AI has made substantial progress ... accelerating the application of medical AI.'

In revised manuscript, we first explain that we design the QPI approach based on our observation on our thyroid ultrasound dataset, and then broaden the scope to explain that we validate the approach on two public datasets based on the fact that sample imbalance is prevalent in medical subgroup analysis.

Actually, during the design of the framework, we did not specifically tailor our methods to the particular medical characteristics of thyroid nodules. Consequently, our QPI approach is not limited to the use of ultrasound thyroid nodule datasets and holds relatively broad applicability.

2. The initial "summary" section seems too large, getting rather deep into the methods and results, for example including a "CKA explanation" subsection which seems out of place. This part of the paper should be more succinct, leaving details for the paper body. Please check other articles in the journal for reference.

Response:

We have made adjustments to the Result section, moving certain details to the Method section, and presenting our conclusions more clearly. The specific modifications are as follows:

(1) While displaying the loss function, we have removed implementation details to ensure that this aspect is thoroughly expounded upon in the method section. The revised content for this section is provided below:

Line 110 - 125:

'As shown in Fig. 3, the QP-Net ... in Method section: QP-Net to implement QPI.'

(2) Additionally, we have revised the CKA explanation content.

The CKA section is used to illustrate the underlying mechanism of the effectiveness of our QPI approach. In the Discussion we've added content to better explain this issue.

Line 286 - 296:

'This investigation underscores a conundrum ... the propitious ones.'

The utilization of CKA in this study is grounded in prior research, notably the work of Kornblith et al. (2019), which elucidates the concept and utility of CKA, and Neyshabur et al. (2020), who applied CKA to quantify feature reuse in transfer learning.

To enhance comprehensibility, we've modified the results of CKA in the Results section: Locate the unfairness layer within the neural network:

Line 157 - 163:

'The learning curve experiment revealed ... Centered Kernel Alignment (CKA).'

Line 166 - 173:

'Fig. 5: CKA method assesses the effect ... in the higher layers.'

Line 175 - 183:

'Fig. 5(b) depicts ΔCKA_{mix} and ΔCKA_{QPI} ... final layer of neural network.'

The details of CKA experiments are added in the Methods section: CKA, to enhance comprehensibility of our CKA experiments results. The revised content is provided below:

Line 476 - 498:

'Centered Kernel Alignment (CKA)

To quantitatively compare network features ... under certain transfer learning strategies.'

3. The domain classification sub net seems a multi-task learning approach. How is this different otherwise?

Response:

Indeed, the domain classification subnet is a component of the domain adaptation module in our QP-Net, which constitutes an adversarial learning structure. It aims to facilitate the learning of more domain-invariant features by the feature extractor. Empirical evidence of the effectiveness of this component in our QP-Net is provided in Supplementary Table 1.

We can observe from the ablation study that removing this subnet leads to a decline in the overall framework's performance on the minority group. We speculate that the removal of this subnet might result in a diminished ability of the feature extractor to extract domain-invariant features.

Furthermore, in our QP-Net, we introduced a multi-task learning structure, but different tasks are defined with the benign-malignant prediction task on different subgroups. In the Introduction section, we also empirically demonstrate with data that across different subgroups, this prediction task can indeed be considered 'multi-task' on our thyroid dataset.

Adaptive weight in L_y	Loss				Subgroup (Maj & Min)	AUC				
	$\gamma_d L_d(\omega_d, \omega_d)$	$\gamma_{MMD} L_{MMD}(\omega_d)$	$\gamma_{BSS} L_{BSS}(\omega_d)$	Train		Valid	Test	Test (Maj)	Test (Min)	
	δ	$\gamma_d = 0.2$	$\gamma_{MMD} = 1$							$\gamma_{BSS} = 0.5$
(1)					Papillary & Follicular	0.9788	0.8486	0.8572	0.8804	0.6741
					Papillary & Medullary	0.9327	0.8363	0.8471	0.8847	0.6502
					Tertiary & Community	0.9411	0.8415	0.8489	0.8852	0.6970
(2)	✓				Papillary & Follicular	0.9676	0.8344	0.8525	0.8659	0.7022
					Papillary & Medullary	0.9483	0.8470	0.8244	0.8764	0.6736
					Tertiary & Community	0.9276	0.8478	0.8510	0.8845	0.6459
(3)		✓			Papillary & Follicular	0.9542	0.8279	0.8110	0.8300	0.6736
					Papillary & Medullary	0.9459	0.8324	0.8345	0.8456	0.7145
					Tertiary & Community	0.9483	0.8235	0.8214	0.8378	0.6591
(4)			✓		Papillary & Follicular	0.9532	0.8245	0.8246	0.8323	0.6523
					Papillary & Medullary	0.9422	0.8334	0.8169	0.8223	0.7088
					Tertiary & Community	0.9428	0.8402	0.8449	0.8523	0.6827
(5)	✓	✓			Papillary & Follicular	0.8980	0.8252	0.8412	0.8475	0.7329
					Papillary & Medullary	0.9116	0.8230	0.8381	0.8491	0.7440
					Tertiary & Community	0.9324	0.8308	0.8389	0.8523	0.7157
(6)	✓		✓		Papillary & Follicular	0.9377	0.8414	0.8426	0.8620	0.7003
					Papillary & Medullary	0.9712	0.8091	0.8220	0.8432	0.7145
					Tertiary & Community	0.9739	0.8130	0.8249	0.8414	0.7289
(7)	✓	✓	✓		Papillary & Follicular	0.9382	0.8208	0.8422	0.8621	0.7575
					Papillary & Medullary	0.9005	0.8048	0.8112	0.8374	0.7329
					Tertiary & Community	0.8892	0.8331	0.8213	0.8375	0.7451
(8)	✓	✓		✓	Papillary & Follicular	0.9239	0.8312	0.8423	0.8528	0.7555
					Papillary & Medullary	0.9291	0.8002	0.8199	0.8256	0.7679
					Tertiary & Community	0.9210	0.8293	0.8377	0.8420	0.7761
(9)	✓	✓	✓	✓	Papillary & Follicular	0.9768	0.8784	0.8672	0.8893	0.8098
					Papillary & Medullary	0.9382	0.8621	0.8572	0.8688	0.7832
					Tertiary & Community	0.9091	0.8621	0.8693	0.8772	0.7794

‘Supplementary Table 1: Ablation study for components of the proposed QP-Net. a. (1) ~ (9): The empirical results support that each component contributes to the overall model performance. b. (2) vs. (5): the adversarial structure contributes to model performance on minority subgroups. c. (2) vs. (7): empirical evidence on the effectiveness of marginal domain invariance on our thyroid ultrasound dataset. d. (5) vs. (6) vs. (7) and (8) vs. (9): combining adversarial structure and MMD loss yields better performance. e. (3) vs. (5), and (4) vs. (6): multi-task learning module contributes to preserving model performance on majority subgroup. f. (5) vs. (8), and (7) vs. (9): BSS loss contributes to the improvement of model performance on minority subgroup.’

4. L31: "Using thyroid cancer as an example, we conducted a literature search using keywords such as "thyroid nodule", "malignant diagnosis", and "rare subtype"...." This search is very targeted at a clinical application, rather than at a more general AI methodology to address fairness and biases, where the literature is broader. This is actually fine if the paper is scoped specifically to the application of interest (ultrasound thyroid nodule imaging). The way the beginning of the introduction is written suggests that there is less work on the topic (generally) than there actually is. I suggest then to introduce the application of interest *before* describing the problem of bias and fairness and *before* describing the literature search.

Response:

We sincerely appreciate your guidance. We’ve modified the Introduction section totally, as explained in our response to minor comment 1 of the reviewer.

Specifically, we have revised the first two paragraphs of the Introduction section in the revised manuscript to provide readers with a clear initial understanding of the focal points

addressed in this study.

Line 26 - 37:

‘AI has made substantial progress ... mitigate the existing unfairness.’

In the Discussion section of the revised manuscript, we have also incorporated an extensive literature review in two key areas: the definition of fairness and approaches to address fairness issues.

Line 257 - 285:

‘This article employs both AUC disparity ... within the majority population.’

5. L329 "where, (...) are hyper parameters determined through experiments." Please include (perhaps in an appendix or supplementary material) all these experiments to enable reproducibility of this research. Also one needs to understand how the different approaches (Quasi-pareto, divide and conquer, single model) were optimized to ensure fair comparison.

Response:

We appreciate the reviewer’s comments and suggestions, and we’ve added our details and results of hyperparameter comparison experiments to Supplementary Materials.

(1) Supplementary Fig. 1 provides experimental results of different values of σ .

Hyperparameter comparison (γ)					AUC					
	σ	γ_d	γ_{MMD}	γ_{BBS}	Subgroup (Maj & Min)	Train	Valid	Test	Test (Maj)	Test (Min)
(1)	0.01	0.2	1	0.5	Papillary & Follicular	0.9423	0.8553	0.8225	0.8374	0.7623
					Papillary & Medullary	0.9422	0.8534	0.8373	0.8423	0.7540
					Tertiary & Community	0.9534	0.8463	0.8255	0.8423	0.7657
Papillary & Follicular	0.9768				0.8784	0.8672	0.8893	0.8098		
Papillary & Medullary	0.9382				0.8621	0.8572	0.8688	0.7832		
Tertiary & Community	0.9091				0.8621	0.8693	0.8772	0.7794		
(2)	0.05	0.2	1	0.5	Papillary & Follicular	0.9342	0.8653	0.8553	0.8797	0.6367
					Papillary & Medullary	0.9530	0.8564	0.8581	0.8602	0.6772
					Tertiary & Community	0.9535	0.8554	0.8503	0.8643	0.6834
Papillary & Follicular	0.9727				0.8432	0.8400	0.8450	0.7232		
Papillary & Medullary	0.9827				0.8676	0.8423	0.8634	0.7540		
Tertiary & Community	0.9762				0.8611	0.8354	0.8644	0.7435		
(3)	0.1	0.2	1	0.5	Papillary & Follicular	0.9748	0.8625	0.8643	0.8743	0.7020
					Papillary & Medullary	0.9727	0.8332	0.8145	0.8442	0.7321
					Tertiary & Community	0.9861	0.8655	0.7934	0.8223	0.6981
Papillary & Follicular	0.9846				0.7632	0.7533	0.7932	0.6428		
Papillary & Medullary	0.9625				0.7653	0.7523	0.7943	0.6782		
Tertiary & Community	0.9542				0.7540	0.7356	0.7534	0.6550		
(4)	0.5	0.2	1	1	Papillary & Follicular	0.9423	0.8553	0.8225	0.8374	0.7623
					Papillary & Medullary	0.9422	0.8534	0.8373	0.8423	0.7540
					Tertiary & Community	0.9534	0.8463	0.8255	0.8423	0.7657
Papillary & Follicular	0.9768				0.8784	0.8672	0.8893	0.8098		
Papillary & Medullary	0.9382				0.8621	0.8572	0.8688	0.7832		
Tertiary & Community	0.9091				0.8621	0.8693	0.8772	0.7794		
(5)	0.1	0.2	1	1	Papillary & Follicular	0.9342	0.8653	0.8553	0.8797	0.6367
					Papillary & Medullary	0.9530	0.8564	0.8581	0.8602	0.6772
					Tertiary & Community	0.9535	0.8554	0.8503	0.8643	0.6834
Papillary & Follicular	0.9727				0.8432	0.8400	0.8450	0.7232		
Papillary & Medullary	0.9827				0.8676	0.8423	0.8634	0.7540		
Tertiary & Community	0.9762				0.8611	0.8354	0.8644	0.7435		
(6)	0.1	0.2	1	1	Papillary & Follicular	0.9748	0.8625	0.8643	0.8743	0.7020
					Papillary & Medullary	0.9727	0.8332	0.8145	0.8442	0.7321
					Tertiary & Community	0.9861	0.8655	0.7934	0.8223	0.6981
Papillary & Follicular	0.9846				0.7632	0.7533	0.7932	0.6428		
Papillary & Medullary	0.9625				0.7653	0.7523	0.7943	0.6782		
Tertiary & Community	0.9542				0.7540	0.7356	0.7534	0.6550		

(b)

‘Supplementary Fig. 1: Hyperparameter comparison experiments of σ . (a) σ governs the extent of minor weight adjustments, and larger σ values yield a smoother $\delta(A_{maj}, A_{mix})$ curve. (b) Empirical results demonstrate that $\sigma=0.05$ emerges as a preferable parameter selection.’

Detailed analysis is provided in Supplementary materials line 32 - 43:

‘Explanation for Supplementary ... preferable parameter selection.’

(2) Supplementary Table 3 provides a comparative analysis of the hyperparameter selection for two additional parameters, namely γ_d and γ_{MMD} , which serves to underscore that, in contrast to scenarios where $\{\gamma_d \neq 0, \gamma_{MMD} = 0\}$ (indicating exclusive utilization of domain adaptation) and $\{\gamma_d = 0, \gamma_{MMD} \neq 0\}$ (indicating sole reliance on MMD), the hyperparameter combination $\{\gamma_d = 0.2, \gamma_{MMD} = 1\}$ emerges as the more favorable choice.

Hyperparameter comparison (γ)			Subgroup (Maj & Min)	AUC				
γ_d	γ_{MMD}			Train	Valid	Test	Test (Maj)	Test (Min)
(1)	0	0	Papillary & Follicular	0.8980	0.8252	0.8412	0.8475	0.7329
			Papillary & Medullary	0.9116	0.8430	0.8381	0.8491	0.7440
			Tertiary & Community	0.9324	0.8508	0.8389	0.8523	0.7157
(2)	0.1	0	Papillary & Follicular	0.9790	0.8466	0.8460	0.8617	0.7189
			Papillary & Medullary	0.9871	0.8505	0.8524	0.8688	0.7236
			Tertiary & Community	0.9909	0.8413	0.8495	0.8665	0.6950
(3)	0.2	0	Papillary & Follicular	0.9887	0.8278	0.8416	0.8597	0.6967
			Papillary & Medullary	0.9732	0.8305	0.8381	0.8502	0.7472
			Tertiary & Community	0.9391	0.8179	0.8080	0.8243	0.6813
(4)	0.3	0	Papillary & Follicular	0.9732	0.8305	0.8381	0.8502	0.7472
			Papillary & Medullary	0.9472	0.8360	0.8288	0.8378	0.7569
			Tertiary & Community	0.9759	0.8354	0.8329	0.8453	0.7419
(5)	0.4	0	Papillary & Follicular	0.9194	0.8247	0.8334	0.8512	0.7086
			Papillary & Medullary	0.9607	0.8436	0.8416	0.8604	0.6823
			Tertiary & Community	0.9825	0.8454	0.8496	0.8713	0.6756
(6)	0	1	Papillary & Follicular	0.9792	0.8403	0.8429	0.8586	0.7111
			Papillary & Medullary	0.9806	0.7755	0.8423	0.8618	0.7017
			Tertiary & Community	0.9812	0.8482	0.8466	0.8624	0.6827
(7)	0.1	1	Papillary & Follicular	0.9079	0.8641	0.8519	0.8800	0.7772
			Papillary & Medullary	0.9069	0.8403	0.8552	0.8762	0.7557
			Tertiary & Community	0.9479	0.8598	0.8527	0.8686	0.7296
(8)	0.2	1	Papillary & Follicular	0.9768	0.8784	0.8672	0.8893	0.8098
			Papillary & Medullary	0.9382	0.8621	0.8572	0.8688	0.7832
			Tertiary & Community	0.9091	0.8621	0.8693	0.8772	0.7687
(9)	0.3	1	Papillary & Follicular	0.9631	0.8377	0.8509	0.8641	0.7641
			Papillary & Medullary	0.9079	0.8341	0.8319	0.8435	0.7559
			Tertiary & Community	0.9069	0.8303	0.8325	0.8464	0.7157
(10)	0.4	1	Papillary & Follicular	0.9938	0.8060	0.8139	0.8514	0.7506
			Papillary & Medullary	0.9634	0.8341	0.8382	0.8562	0.7490
			Tertiary & Community	0.9321	0.8352	0.8499	0.8623	0.7578
(11)	0	2	Papillary & Follicular	0.9948	0.8162	0.8023	0.8512	0.6453
			Papillary & Medullary	0.9859	0.8023	0.8376	0.8500	0.6479
			Tertiary & Community	0.9050	0.8299	0.8150	0.8516	0.6500
(12)	0.1	2	Papillary & Follicular	0.8907	0.8247	0.8252	0.8467	0.6373
			Papillary & Medullary	0.8907	0.8241	0.8247	0.8463	0.6369
			Tertiary & Community	0.8901	0.8217	0.8241	0.8450	0.6433
(13)	0.2	2	Papillary & Follicular	0.8817	0.8313	0.8340	0.8475	0.7101
			Papillary & Medullary	0.8866	0.8308	0.8335	0.8471	0.6442
			Tertiary & Community	0.8850	0.8309	0.8338	0.8473	0.7109
(14)	0.3	2	Papillary & Follicular	0.9414	0.8299	0.8319	0.8487	0.6803
			Papillary & Medullary	0.9407	0.8324	0.8315	0.8499	0.6716
			Tertiary & Community	0.9186	0.8265	0.8264	0.8467	0.6463
(15)	0.4	2	Papillary & Follicular	0.9382	0.8353	0.8328	0.8393	0.5377
			Papillary & Medullary	0.9732	0.8055	0.8032	0.8300	0.6792
			Tertiary & Community	0.9412	0.8325	0.8380	0.8485	0.7485

‘Supplementary Table 3: Hyperparameter comparison results of all three pairs of subgroups on our thyroid ultrasound dataset. Hyperparameters used in the experiment are determined based on empirical results, and the optimal hyperparameter combination yielded the best model performance on all three pairs of subgroups.’

Regarding the optimization of the three experiments (mix training, divide and conquer, and QPI), we’ve added corresponding explanations in the Methods section: Experimental Variables Control for Mix Training, Divide and Conquer, and QPI approach within the revised manuscript.

Line 501 - 514:

‘All three experiments utilize ... 1000 images with replacement.’

Finally, we would like to express our gratitude once again for your valuable feedback on the manuscript. Your input is instrumental in making our research more robust and insightful.

Reviewers' Comments:

Reviewer #1:

Remarks to the Author:

Firstly, I would like to thank the authors for their very substantial revision, which surely must have been a tremendous amount of work, and for taking my concerns so seriously! Many of my key concerns have been addressed, and the claims made by the authors are now supported by extensive evidence. Overall, the manuscript is in much better shape, and I really appreciate all the details provided by the authors. The new experiments and baselines now convincingly demonstrate the practical superiority of the new proposed method.

Some important issues remain, however, and I will outline them below. I am confident that the authors will be able to resolve these remaining issues in another revision.

****Major issues****

1. For Figure 4, it is currently not clear whether these are the results obtained by a **single** model, optimized to reduce disparities with respect to one specific (which?) majority-minority pairing, or whether these are results obtained from **multiple** models, optimized to reduce disparities with respect to different majority-minority pairings. From the text, I suspect that the latter is the case: one model was trained to minimize PTC vs (FTC, MTC) disparities, and another model was trained to minimize Tertiary vs. Community hospital disparities. If this is the case, it would be essential to i) clearly state this, and ii) also assess whether minimizing disparities with respect to one criterion leads to improvement or degeneration in disparities with respect to the other criterion, as is often observed in practice.

1b. Relatedly, for the CheXpert case study / Fig. 6, which subgroups were used for the QPI training? Gender, age, race? Again, was a single model trained or multiple different ones?

2. The authors choose AUROC as their key metric for comparing performance between subgroups. AUROC is a purely discriminative performance measure (equal to the likelihood of positive examples being ranked above negative examples) and fully ignores calibration. It could be the case that a group has good (within-group) AUROC but will in practice always be classified as negative because of poor model calibration in that group. This would not be detected by the analyses currently presented in the paper. To resolve this doubt, group performance should also be compared according to a secondary metric that accounts for potential calibration issues, such as for example the (balanced) Brier score.

3. The notion of Pareto fairness should be defined and used more precisely, and clearly related to the extensive existing literature on Pareto fairness. (And maybe, the use of the term should be reconsidered.) In the caption of Fig. 2, for instance, the authors write that "For imbalanced subgroups, increasing the sample size resulted in [...] a consequential decrease in the prediction performance on the sample-size-fixed minority group, which contravenes the Pareto fairness criterion." Decreasing performance on a minority group is not in conflict with the standard definition of Pareto fairness, cf., e.g., Zietlow et al., Martinez et al., Wang et al., Wei and Niethammer. In lines 376/377, the authors write that "an algorithm is said to exhibit Pareto fairness on D , if as [majority sample size] increases, its test performance does not decrease for either [the majority or minority subgroups]." This sounds like a useful fairness criterion; I am just not convinced that it has any relationship to a Pareto frontier of any sort? In their response to my previous review, the authors also write that their "Quasi-Pareto Improvement framework aims to push the Pareto front forward" - but a Pareto front is, in virtually all instances I know of, defined as an immovable theoretically achievable optimum, not as something that can be further optimized.

4. While the authors now cite significantly more prior work, citations are still very sparse despite an extensive body of prior work. E.g., in the section "Measurement of Fairness in Medical AI Diagnosis" (lines 336 - 385), a total of one (1) paper is cited (reference 52), which, in fact, is a rather obscure and specific reference that has no connection to medical AI, general AI fairness, or the specific methods used here. This is despite very fundamental concepts being introduced and discussed in this section, such as the fact that performance disparities are often used to indicate model unfairness, and the whole Pareto fairness topic (no work is cited on this topic).

5. The authors interpret the different disease subtypes as different "domains", and then proceed to use domain-invariant learning methods. However, disease subtypes would not usually be considered different "domains" but rather (often unobserved) label subclasses, c.f., e.g., Li et al. and Mahajan et al. Typical domains include things like hospital type, scanner type, gender/sex, age, etc., which are unrelated from the label y . See, e.g., Guan and Liu. (I tried to hint at this issue in my previous review where I remarked on the fundamental difference between demographic patient groups and disease subtypes, but I did not spell this out clearly enough.) The empirical results seem to indicate that what the authors do is useful, but this very unusual choice of "domains" should be spelled out and discussed clearly.

****Minor issues****

1. There are various references to Anaplastic Thyroid Carcinoma (ATC) throughout the introduction, but this never appears again in the rest of the manuscript.
2. In line 59, the "CKA" abbreviation appears for the first time. It should be spelled out and an appropriate reference provided.
3. In Figure 2a, I assume that the little points+errorbars and the "Estimate" column both refer to AUROC. This should be clearly labeled.
4. In lines 105-106, the authors write that "The results of Divide-and-conquer approaches are also plagued by similar issues concerning minority subgroups. The results of low prediction AUC for minority subgroups..." I think it should be noted that the results are already **much** better with this approach compared to the "mixed" approach? Also, I do not think that these results indicate "inadequate training", but rather insufficient data?
5. I would suggest noting directly after Eqs. (1) and (2) that δ is chosen automatically and adaptively. (I initially assumed it would be a constant scaling factor.)
6. In Fig. 3, I assume that "domain" = "subgroup"? If yes, I would strongly suggest unifying the nomenclature. Also, I would suggest indicating visually which terms receive the subgroup/domain as an input, and which ones do not.
7. In line 156, I suggest rephrasing the heading "Locate the unfairness layer within the neural network"
8. The sentence in line 157-158 probably refers to the "mixed" baseline training method?
9. In lines 159-160, the authors write that "CKA measures the degree of feature reuse in transfer learning", but I suspect that it also has something to do with model performance? E.g. in the caption of Fig. 5, the authors seem to indicate that positive values of δ CKA would indicate beneficial performance gains?
10. In Fig. 6a, does the yellow "AUC development" refer to the gains from using the QPI approach? If so, this should be clearly stated. Also, in Fig. 6b, the three different yellow/orange colors are very hard to distinguish.
11. In lines 260-261, the authors write that "Calibration addresses instances where certain subgroups are disproportionately likely to receive positive predictions." This is not a correct characterization of calibration.
12. In lines 301-302, the authors write that "the proposed QPI approach allows for sequential utilization under each scheme" - could the authors elaborate on this? What do they mean by "sequential utilization"? Why sequential, and not in parallel / for all groupings simultaneously?
13. In lines 366-367, the authors write that "In the scenarios of medical diagnosis minority subgroup, [fairness] means whether the performance disparity is caused by the imbalance of sample size." This is a highly simplistic notion of fairness in medical AI. There are many other reasons why a medical AI system might be unfair, and a performance disparity can also be unfair if it is not caused by sample size imbalance. See e.g. Petersen et al., which may also be of interest to the authors in other regards since it is very related to the topic of the manuscript.
14. What the authors call "mixed training" corresponds to standard empirical risk minimization (ERM). This should be noted somewhere. Similarly, what the authors call "Divide and conquer" is also known as "stratified training", see, e.g., Ref 33 (Zhang et al.).
15. Lines 478+. "kornblith et al." should probably be upper case; i , j_1 and j_2 are not specified, and what is a "characteristic representation"?
16. In lines 507-508, the binary / multi-class cases are mixed up. Also, is early stopping really done w.r.t. the **training** AUC? It would be much more usual to do so w.r.t. the **validation** AUC.
17. I am confused concerning our previous exchange concerning the sample numbers in Fig 2a -

why would the authors want to only list the PTC cases in the "Hospitals" category, and not the other cases? This is very unusual & confusing. If the authors really want to keep it like this, they should clearly note this in the table and/or its caption.

18. It is true that most of the domain adaptation literature considers the unsupervised case. However, the fundamental theorems and incompatibilities also fully apply to the supervised case; see, e.g., <https://arxiv.org/pdf/2305.01397.pdf>

19. Concerning marginal vs. class-conditional invariance: marginal invariance is especially problematic if prevalences differ between domains. Could the authors remark on the prevalence / class balance in their different domains? If this is uniform across domains, that would explain why marginal invariance is not problematic here.

20. In their response to my previous review, the authors write that "in previous research, combining two [domain invariance] losses with similar objectives has shown the potential for achieving improved results." - could they provide a reference for this statement?

21. In supplementary table 2, what does "Training process = Quasi Pareto" (rows 2 and 4) mean? (How) is it different from using the loss in the lowest row of the table? And finally, which of the three "Ours" methods in that table correspond to the proposed QIP method? I suspect it's the final row? This should all be clearly spelled out in the table caption.

References

Guan and Liu <https://ieeexplore.ieee.org/stamp/stamp.jsp?arnumber=9557808>

Li et al. <https://arxiv.org/pdf/2006.13352.pdf>

Mahajan et al. <http://proceedings.mlr.press/v139/mahajan21b/mahajan21b.pdf>

Petersen et al. <https://doi.org/10.1016/j.patter.2023.100790>

Wang et al. <https://dl.acm.org/doi/pdf/10.1145/3447548.3467326>

Wei and Niethammer <https://onlinelibrary.wiley.com/doi/pdf/10.1002/sam.11560>

Zietlow et al.

https://openaccess.thecvf.com/content/CVPR2022/papers/Zietlow_Leveling_Down_in_Computer_Vision_Pareto_Inefficiencies_in_Fair_Deep_CVPR_2022_paper.pdf

Reviewer #2:

Remarks to the Author:

I thank the authors for the efforts improving the paper. I can appreciate that they have made an effort to address my comments, which they have addressed mostly satisfactorily. I still think some could have been addressed better. Specifics follow below, which I think can be addressed in a revision.

1. To my main concern, about the contextualization of the proposed method beyond Thyroid nodules, authors have replied by adding content in the discussion. This indeed addresses my point however I find it strange to place the text in the discussion. One would expect that, when stating the motivation for the work, at the beginning of the paper, authors describe what the current problem is, why is it not solved, and how current approaches fall short -that is the place to put this review of similar literature.

2. Stating early that authors propose a generic framework then exemplified by thyroid nodules: Related to the main point, in comment 1.1. I suggested to state early on that the study is about thyroid nodules. however, in the revised version, they convey that "AI has made substantial progress as a clinical diagnostic tool for a variety of diseases", to then follow on with examples of thyroid nodules work (without describing that this is the focus, or arguing whether thyroid

nodules would be representative of a wider range of disease), and following from those examples they infer that " Given sample size imbalance, AI models might be subject to unfairness by omitting features of the minority subgroups" which is a broad statement, generic, and not thyroid nodule specific. The same trend follows in the rest of the motivation section. In brief, they try to motivate and defend their work as a generic framework but all evidence is pulled from thyroid nodule works, and that does not hold/generalize. Please, either broaden the focus and build your case form a wider range of applications, or really formulate your proposal as a thyroid nodule work.

In general lines, authors have addressed all my other comments.

REVIEWER COMMENTS

Reviewer #1 (Remarks to the Author):

Firstly, I would like to thank the authors for their very substantial revision, which surely must have been a tremendous amount of work, and for taking my concerns so seriously! Many of my key concerns have been addressed, and the claims made by the authors are now supported by extensive evidence. Overall, the manuscript is in much better shape, and I really appreciate all the details provided by the authors. The new experiments and baselines now convincingly demonstrate the practical superiority of the new proposed method.

Some important issues remain, however, and I will outline them below. I am confident that the authors will be able to resolve these remaining issues in another revision.

****Major issues****

For Figure 4, it is currently not clear whether these are the results obtained by a **single** model, optimized to reduce disparities with respect to one specific (which?) majority-minority pairing, or whether these are results obtained from **multiple** models, optimized to reduce disparities with respect to different majority-minority pairings. From the text, I suspect that the latter is the case: one model was trained to minimize PTC vs (FTC, MTC) disparities, and another model was trained to minimize Tertiary vs. Community hospital disparities. If this is the case, it would be essential to i) clearly state this, and ii) also assess whether minimizing disparities with respect to one criterion leads to improvement or degeneration in disparities with respect to the other criterion, as is often observed in practice.

1b. Relatedly, for the CheXpert case study / Fig. 6, which subgroups were used for the QPI training? Gender, age, race? Again, was a single model trained or multiple different ones?

Response:

We sincerely thank the reviewers for the recognition of our work and the valuable suggestions.

i) Yes, the latter is the case. The results in Fig. 4 are obtained from multiple models, each of which was optimized with respect to one specific majority-minority pairing. However, we did not train a model for PTC vs (FTC, MTC); instead, we addressed the fairness issues for FTC and MTC separately using distinct models. Specifically, as shown in Fig. 4 (b) (c) (d), we optimized a total of three models on our thyroid dataset: one was optimized for PTC vs. FTC, one for PTC vs. MTC, and one for Tertiary Hospital (TH) vs. Community Hospital (CH).

We've made corresponding modifications in the revised manuscript to clearly state that the experiments were performed for each of the three majority-minority pairs:

Line 148-150:

The experiments were performed ... within hospitals.

ii) There is a potential for improvement in model performance on other subgroups when implementing the QPI approach on one subgroup, although this may depend on the specific circumstances. Our approach primarily consists of a multi-task learning component and

domain adaptation components. The multi-task learning component is designed to balance the model between the given majority subgroup and minority subgroup, and its impact on other subgroups may vary depending on the specific condition, for example, the underlying connections and differences between subgroups. However, the domain adaptation components are designed to improve the model's capability to learn domain-invariant features. When focusing on enhancing the performance of a specific minority subgroup, it is possible that this trend could result in improved model performance on other subgroups.

1b. Multiple different models were trained, each of which was to address unfairness issues w.r.t. one specific majority-minority pairing.

For CheXpert case study in Fig. 6, the majority-minority pairings are:

- a. Female: majority {18-40, 40-60, 60-80} vs. minority {>80};
- b. Female: majority {White, Asian, Other} vs. minority {Black};

For Skin Imaging case study in Fig. 7, the majority-minority pairings are:

- a. Majority {0-59} vs. minority {60~85}
- b. Majority {Female} vs. minority {Male}

We've made corresponding modifications in the caption of Fig. 6 and Fig. 7 to clearly state that the experiments on the two public datasets were performed for each of the above majority-minority pairs:

Line 211-213:

The experiments were performed ... within race groups.

Line 226-228:

The experiments were performed ... within sex groups.

2. The authors choose AUROC as their key metric for comparing performance between subgroups. AUROC is a purely discriminative performance measure (equal to the likelihood of positive examples being ranked above negative examples) and fully ignores calibration. It could be the case that a group has good (within-group) AUROC but will in practice always be classified as negative because of poor model calibration in that group. This would not be detected by the analyses currently presented in the paper. To resolve this doubt, group performance should also be compared according to a secondary metric that accounts for potential calibration issues, such as for example the (balanced) Brier score.

Response:

Thanks very much for the reviewers' advice, we chose the Brier score [1] to assess and compare the accuracy of binary predictions or prediction models in our study.

We've added the Brier score results in Fig. 4 and revised the corresponding caption as follows. From Figure 4(b), it is evident that the Brier Score results of the QPI model for minority subgroups decreased significantly when compared to those of Mix training, while the results for majority subgroups showed a slight increase.

Line 168-175:

Fig. 4: Benign-malignant ... (b) OPI approach effectively ... minority subgroup. ...

We've added a corresponding analysis regarding Brier score results in the revised manuscript. Line 154-157:

Fig. 4 (b) displays the Brier Score results ... increase for majority subgroups.

We've added the explanation of the Brier score in Supplementary Materials as follows.

Supplementary line 119-124:

(3) **Brier Score**

The Brier Score is a measure ... the better.

$$BS = \frac{1}{N} \sum_{t=1}^N (y_{\text{pred}} - y_{\text{label}})^2$$

Where, y_{pred} is the predicted probability, y_{label} is the true label.

The corresponding reference have also been added to Supplementary Materials:

Supplementary line 145:

9. Brier, G. W. (1950). Verification of forecasts expressed in terms of probability. Monthly weather review, 78(1), 1-3.

3. The notion of Pareto fairness should be defined and used more precisely, and clearly related to the extensive existing literature on Pareto fairness. (And maybe, the use of the term should be reconsidered.) In the caption of Fig. 2, for instance, the authors write that "For imbalanced subgroups, increasing the sample size resulted in [...] a consequential decrease in the prediction performance on the sample-size-fixed minority group, which contravenes the Pareto fairness criterion." Decreasing performance on a minority group is not in conflict with the standard

definition of Pareto fairness, cf., e.g., Zietlow et al., Martinez et al., Wang et al., Wei and Niethammer. In lines 376/377, the authors write that "an algorithm is said to exhibit Pareto fairness on D, if as [majority sample size] increases, its test performance does not decrease for either [the majority or minority subgroups]." This sounds like a useful fairness criterion; I am just not convinced that it has any relationship to a Pareto frontier of any sort? In their response to my previous review, the authors also write that their "Quasi-Pareto Improvement framework aims to push the pareto front forward" - but a Pareto front is, in virtually all instances I know of, defined as an immovable theoretically achievable optimum, not as something that can be further optimized.

Response:

We sincerely appreciate the reviewer's comments, and provide detailed explanations to address the issues. In summary, current discussions on 'Pareto frontier' in extant literature are usually observation-based and within a resource-fixed environment (where a Pareto frontier is immovable), while the fairness criterion proposed in this study is causality-based and within a resource-changing environment (where a Pareto frontier might be moved). Indeed, we acknowledge the existence of certain issues in our previous response, as a Pareto frontier cannot be influenced by our method.

To avoid any overlap with existing literature, this revision has involved the uniform renaming of our defined fairness criterion as the 'Pareto fairness criterion', rather than the widely used notation of 'Pareto fairness', and has also provided a clearer contextualization of our notion of fairness within the revised manuscript (modification list is provided in our response to major issue 4).

In the existing literature on Pareto fairness, researchers primarily discuss 'Pareto improvement' and 'Pareto frontier'. 'Pareto improvement' refers to actions that result in certain individuals gaining in a given situation without causing any other individuals to lose. 'Pareto frontier' represents a theoretically achievable optimum under some resource-fixed conditions, where no further Pareto improvement is possible. However, the unfairness phenomena considered in our study manifest as changes in model performance with an increase in (majority) sample size. In such cases the Pareto frontier may be 'put forward' due to changes in total resource availability in the environment.

Our defined 'Pareto fairness criterion' is causality-based, which also differs from the discussions on fairness in existing literature which are usually observational-based. Existing literature often observes tradeoff in model performance between the majority and minority performance, but such discussions do not delve into the causes of performance disparity. Our 'Pareto fairness criterion' is causality-based, taking a more in-depth analysis by investigating and discussing the impact of sample size imbalance on fairness.

4. While the authors now cite significantly more prior work, citations are still very sparse despite an extensive body of prior work. E.g., in the section "Measurement of Fairness in Medical AI Diagnosis" (lines 336 - 385), a total of one (1) paper is cited (reference 52), which, in fact, is a rather obscure and specific reference that has no connection to medical AI, general AI fairness, or the specific methods used here. This is despite very fundamental concepts being introduced and discussed in this section, such as the fact that performance disparities are often

used to indicate model unfairness, and the whole Pareto fairness topic (no work is cited on this topic).

Response:

Thanks for the reviewer's advice on writing and citations. When addressing major issue 3, we've added citations regarding the concept of fairness regarding disparity and Pareto fairness, and prior work regarding Pareto improvement, accompanied by the contextualization of our notation of fairness. Our modifications are listed as follows.

Line 378:

The performance disparity is often used to indicate model unfairness⁴⁹⁻⁵¹.

Line 389-390:

Performance disparity ... definitions of Pareto fairness⁵⁰.

Line 390-392:

Most studies on Pareto fairness ... detriment to others⁵²⁻⁵⁴.

Line 392-395:

Nonetheless, there are phenomena ... decision-making algorithms^{52,55,56}.

Line 398-399:

However, few studies approach this fairness issue from a causal perspective.

Line 402-403:

which is a causality-based fairness ... in literature.

Line 440-441:

Previous investigations into Pareto improvement ... while benefiting others^{52,54,57,58}.

Line 441-444:

Our QPI approach doesn't rigidly constrain ... increases in sample size.

5. The authors interpret the different disease subtypes as different "domains", and then proceed to use domain-invariant learning methods. However, disease subtypes would not usually be considered different "domains" but rather (often unobserved) label subclasses, c.f., e.g., Li et al. and Mahajan et al. Typical domains include things like hospital type, scanner type, gender/sex, age, etc., which are unrelated from the label y . See, e.g., Guan and Liu. (I tried to hint at this issue in my previous review where I remarked on the fundamental difference between demographic patient groups and disease subtypes, but I did not spell this out clearly enough.) The empirical results seem to indicate that what the authors do is useful, but this very unusual choice of "domains" should be spelled out and discussed clearly.

Response:

We appreciate the reviewer's suggestions. In the context of our problem, we posit that it is potentially reasonable that different subtypes of thyroid cancer are treated as different domains.

There are prior works that motivated us to explore domain-invariant methods in the context of our problem. Previous studies utilizing medical AI for benign-malignant diagnosis of lung cancer chose different subtypes as different domains [2]. Also in extant literature, there are prior works that treat different types of cancer as different domains [3]. Thus, for thyroid cancer, it is potentially reasonable that different subtypes are also regarded as different types of cancer based on their biological mechanisms.

At the same time, we appreciate the reviewer for pointing out minor issues 18 and 19, to which class balance may be an underlying reason why our domain-invariant approaches would work.

****Minor issues****

1. There are various references to Anaplastic Thyroid Carcinoma (ATC) throughout the introduction, but this never appears again in the rest of the manuscript.

Response:

In the revised manuscript line 57, we apologize for the mistake of using ‘MTC and ATC’ previously and have replaced it with ‘FTC and MTC’.

Line 57:

..., we observed significant model performance disparity in FTC and MTC (minority), ...

Concerning the reference to 'ATC' in line 38 of the Introduction, its inclusion was intended to provide a more comprehensive overview of our literature search results.

2. In line 59, the "CKA" abbreviation appears for the first time. It should be spelled out and an appropriate reference provided.

Response:

We sincerely thank the reviewer for pointing out this mistake and have made modifications regarding this issue in the revised manuscript line 68.

Line 68-69:

... on our thyroid dataset and illustrated using Centered Kernel Alignment (CKA) method.

3. In Figure 2a, I assume that the little points+errorbars and the "Estimate" column both refer to AUROC. This should be clearly labeled.

Response:

Yes indeed. We've labeled it clearly in the revised Fig. 2(a) as follows.

Line 86-96:

Fig. 2: Benign-malignant prediction ... Pareto fairness criterion.

4. in lines 105-106, the authors write that "The results of Divide-and-conquer approaches are also plagued by similar issues concerning minority subgroups. The results of low prediction AUC for minority subgroups..." I think it should be noted that the results are already *much* better with this approach compared to the "mixed" approach? Also, I do not think that these results indicate "inadequate training", but rather insufficient data?

Response:

We appreciate the reviewer's comment and have added corresponding content in the revised manuscript.

Line 113-114:

The "Divide-and-conquer" methods ... majority and minority subgroups.

As for divide and conquer approaches, 'insufficient data' is indeed more accurate as the indication of the corresponding results. We've modified this sentence in the revised manuscript to clearly state this point.

Line 114-115:

The low predictive AUC for minority subgroups in Fig. 2 (b) suggests a lack of sufficient data.

5. I would suggest noting directly after Eqs. (1) and (2) that delta is chosen automatically and adaptively. (I initially assumed it would be a constant scaling factor.)

Response:

We thank the reviewer for pointing this out and have added corresponding contents accordingly in the revised manuscript.

Line 131-132:

..., and δ , adaptively determined through ... overall loss function.

6. In Fig. 3, I assume that "domain" = "subgroup"? If yes, I would strongly suggest unifying the nomenclature. Also, I would suggest indicating visually which terms receive the subgroup/domain as an input, and which ones do not.

Response:

We appreciate the reviewer's comment. Indeed, 'domain' = 'subgroup'. We've made modifications throughout the revised manuscript regarding this problem, unifying the nomenclature by using 'subgroup' instead of 'domain', except for the notion of 'domain adaptation' which refers to the existing domain adaptation methods in the literature.

Line 121-122:

the domain-adaptation module ... across two subgroups to improve ...

Line 122-123:

... subgroup feature fitting in which different subgroups are treated as different domains.

Line 123-124:

The training procedure minimized ... and subgroup (majority-minority) ...

Line 125:

... learn a feature extractor that generates subgroup-invariant features.

Line 142:

... including an adversarial structure (the red subgroup classifier) and ...

Regarding the input, in the revised manuscript, we have indicated visually the components that uses subgroup as an input in Fig. 3, which is shown as follows.

Line 137-145:

Fig. 3: Quasi-Pareto Net diagram. A typical input includes ... A feature ... and performance.

7. In line 156, I suggest rephrasing the heading "Locate the unfairness layer within the neural network".

Response:

Indeed, the previous heading may not have been sufficiently apt. We have refined it to 'Layer-wise Examination of Fairness in Neural Networks' in revised manuscript line 176.

Line 176:

Layer-wise Examination of Fairness in Neural Networks

8. The sentence in line 157-158 probably refers to the "mixed" baseline training method?

Response:

Yes indeed, and we thank the reviewer for pointing this out. We've modified the sentence to form a clearer expression.

Line 177-178:

The learning curve experiment of the mixed training approach revealed ... minority prediction.

9. In lines 159-160, the authors write that "CKA measures the degree of feature reuse in transfer learning", but I suspect that it also has something to do with model performance? E.g. in the caption of Fig. 5, the authors seem to indicate that positive values of delta CKA would indicate beneficial performance gains?

Response:

The CKA results themselves directly measure the extent of feature reuse, rather than performance. However, this study posited that the degree of feature reuse in the majority of features is implicitly linked to the model's performance in minority predictions.

The discussion of the results in Fig. 5 is directly built upon the findings in Fig. 4 (which demonstrate significant performance improvements with the QPI approach). We aimed to utilize CKA results to further explore the underlying reasons for the observed performance gains. Positive Δ CKA values indicate that the method reuses more features for minority prediction. In this context, the QPI approach exhibits a higher level of feature reuse compared to mixed training. As Fig. 4 has already indicated that the QPI method indeed leads to performance gains, we inferred that the results in Fig. 5 offer an explanation for the findings in Fig. 4, specifically that the QPI method retains more features conducive to both majority and minority prediction.

10. In Fig.6a, does the yellow "AUC development" refer to the gains from using the QPI approach? If so, this should be clearly stated. Also, in Fig. 6b, the three different yellow/orange colors are very hard to distinguish.

Response:

Thanks for the reviewer's suggestions. Indeed, the 'AUC development' refers to the gains from using the QPI approach. We've stated it clearly in the caption of Fig. 6 in the revised manuscript.

Also, we apologize for the trouble caused by the color selection of the picture and have modified the color of the picture, unifying the color of AUC development as yellow in Fig.6 (a) and Fig.6 (b) and avoiding using colors similar to yellow for other cylinders.

Line 237-248:

CheXpert Dataset

Method	Train (AUROC)	Valid (AUROC)	Subgroup	Test (AUROC)															
				Average	NoFind	EnlargC	Cardio	LungO	LungL	Edema	Csolid	Pneum	Atelect	Pmotho	PleuralE	PleuralO	Fracture	SupD	
Baseline			18-40	0.819 (0.798 - 0.886)	0.884	0.642	0.863	0.791	0.746	0.862	0.715	0.731	0.718	0.858	0.890	0.871	0.783	0.888	
Quasi-Pareto			18-40	0.814 (0.792 - 0.872)	0.876	0.686	0.879	0.792	0.756	0.869	0.755	0.723	0.720	0.869	0.900	0.871	0.787	0.872	
Baseline			40-60	0.841 (0.828 - 0.886)	0.877	0.705	0.864	0.737	0.766	0.831	0.758	0.758	0.704	0.868	0.884	0.788	0.779	0.874	
Quasi-Pareto			40-60	0.847 (0.824 - 0.886)	0.876	0.715	0.864	0.788	0.768	0.837	0.756	0.760	0.736	0.851	0.882	0.778	0.773	0.873	
Baseline	0.985	0.816	60-80	0.835 (0.765 - 0.857)	0.874	0.686	0.845	0.714	0.780	0.812	0.740	0.745	0.691	0.870	0.865	0.823	0.722	0.877	
Quasi-Pareto			60-80	0.831 (0.805 - 0.873)	0.869	0.672	0.831	0.744	0.799	0.822	0.712	0.753	0.668	0.869	0.863	0.854	0.743	0.897	
Baseline			>80	0.765 (0.742 - 0.819)	0.830	0.655	0.789	0.684	0.743	0.813	0.668	0.700	0.634	0.837	0.872	0.770	0.731	0.884	
Quasi-Pareto			>80	0.801 (0.756 - 0.842)	0.847	0.689	0.834	0.682	0.772	0.823	0.683	0.751	0.665	0.890	0.851	0.814	0.765	0.897	
Baseline			Asian	0.795 (0.741 - 0.842)	0.878	0.714	0.868	0.731	0.728	0.833	0.783	0.693	0.679	0.888	0.887	0.704	0.729	0.891	
Quasi-Pareto			Asian	0.805 (0.769 - 0.861)	0.893	0.708	0.884	0.743	0.758	0.863	0.793	0.793	0.706	0.888	0.793	0.757	0.793	0.793	
Baseline			Black	0.786 (0.750 - 0.853)	0.833	0.663	0.810	0.760	0.738	0.835	0.724	0.678	0.685	0.876	0.846	0.737	0.716	0.855	
Quasi-Pareto	0.994	0.799	Black	0.803 (0.789 - 0.860)	0.873	0.683	0.847	0.722	0.776	0.857	0.745	0.717	0.749	0.898	0.892	0.805	0.747	0.907	
Baseline			White	0.830 (0.787 - 0.853)	0.880	0.682	0.842	0.729	0.787	0.826	0.735	0.799	0.754	0.865	0.870	0.810	0.786	0.870	
Quasi-Pareto			White	0.836 (0.780 - 0.882)	0.891	0.695	0.851	0.743	0.795	0.841	0.741	0.742	0.697	0.882	0.885	0.822	0.801	0.889	
Baseline			Other	0.802 (0.763 - 0.863)	0.869	0.688	0.864	0.764	0.720	0.826	0.728	0.757	0.698	0.868	0.883	0.794	0.721	0.885	
Quasi-Pareto			Other	0.806 (0.775 - 0.850)	0.891	0.648	0.870	0.745	0.740	0.824	0.728	0.773	0.738	0.864	0.888	0.765	0.715	0.876	

Fig. 6: ...CheXpert dataset. **AUC development ... highlighted in yellow.** ... QPI approach.

11. In lines 260-261, the authors write that "Calibration addresses instances where 18 subgroup are disproportionately likely to receive positive predictions." This is not a correct characterization of calibration.

Response:

Indeed, we appreciate the suggestion and have modified the sentence to address the issue.

Line 285-286:

Fairness regarding calibration addresses instances ... such outcomes as unfair^{4,23,39,40}.

12. In lines 301-302, the authors write that "the proposed QPI approach allows for sequential utilization under each scheme" - could the authors elaborate on this? What do they mean by "sequential utilization"? Why sequential, and not in parallel / for all groupings simultaneously?

Response:

We apologize for any misunderstanding regarding this sentence. In the previous manuscript, we aimed to convey that our method can effectively tackle the unfairness issues within each scheme, but this was not expressed correctly. Within each majority-minority pair,

we trained individual models to address unfairness issues. In the revised manuscript, we have modified the sentence as follows.

Line 327-328:

While our design of ... allows for respective utilization under each scheme, ...

13. In lines 366-367, the authors write that "In the scenarios of medical diagnosis minority subgroup, [fairness] means whether the performance disparity is caused by the imbalance of sample size." This is a highly simplistic notion of fairness in medical AI. There are many other reasons why a medical AI system might be unfair, and a performance disparity can also be unfair if it is not caused by sample size imbalance. See e.g. Petersen et al., which may also be of interest to the authors in other regards since it is very related to the topic of the manuscript.

Response:

We apologize for the mistake. Indeed, many other reasons could lead to unfairness, and this study focuses on the unfairness issue caused by sample size imbalance. We've modified the sentence to address the issue.

Line 397-398:

In the scenarios of ..., a critical concern is whether performance ... size imbalances.

14. What the authors call "mixed training" corresponds to standard empirical risk minimization (ERM). This should be noted somewhere. Similarly, what the authors call "Divide and conquer" is also known as "stratified training", see, e.g., Ref 33 (Zhang et al.).

Response:

We appreciate the reviewer for pointing this out. We've noted them in the revised manuscript accordingly.

Line 45-46:

... mixed training, also known as standard empirical risk minimization (standard ERM), ...

Line 46-47:

..., and divide-and-conquer, also known as stratified ERM²².

15. Lines 478+. "kornblith et al." should probably be upper case; i, j1 and j2 are not specified, and what is a "characteristic representation"?

Response:

We appreciate the reviewer for pointing this out. We've modified corresponding sentences in the revised manuscript accordingly.

Line 516:

... across networks as per Kornblith et al. (2019).

Line 522-523:

We refer to Kornblith et al. (2019) for detailed calculation.

And as for the specification of the notation of i, j1, j2, and 'characteristic representation', we followed the definition of Kornblith et al. (2019), and have made corresponding modifications in the revised manuscript as presented below. Specifically, we acknowledge that the term 'characteristic representation' is not accurate, and have replaced it with proper definitions in the revised manuscript.

Line 517-521:

..., for each pair of corresponding layers whose matrix of activations of j_1 (from M_1) and j_2 (from M_2) neurons for certain i examples are denoted as $X \in \mathbb{R}^{i \times j_1}$ and $Y \in \mathbb{R}^{i \times j_2}$ respectively, we calculated CKA using the following formula:

$$\text{CKA}(M, N) = \frac{\text{HSIC}(K, L)}{\sqrt{\text{HSIC}(K, K) \text{HSIC}(L, L)}} \quad (21)$$

where, $K = XX^T$ and $L = YY^T$,

16. In lines 507-508, the binary / multi-class cases are mixed up. Also, is early stopping really done w.r.t. the *training* AUC? It would be much more usual to do so w.r.t. the *validation* AUC.

Response:

Thanks for the reviewers' questions. We apologize for the lack of clarity and misrepresentation. We used Cross-Entropy Loss (CE Loss) as the loss function for binary classification and multi-class classification to address thyroid benign-malignant tasks and Skin image tasks.

CheXpert task is a multi-label classification task, so we chose Binary Cross-Entropy as multiple binary classification tasks superimposed. Therefore, BCE can also be applied to multi-label classification tasks after a simple modification. Before using BCE, we quantified the output variable between [0,1] (we used the Sigmoid activation function).

In order to correct the vague description, we've made corresponding modifications in the revised manuscript.

Line 546-548:

For binary and multi-class classification problems, Cross-Entropy Loss (CE Loss) is used, and for multi-label classification, Binary Cross-Entropy Loss (BCE Loss) is employed.

As for early stopping, we sincerely appreciate the reviewer for pointing out this mistake. Indeed, early stopping is done with reference to the validation AUC. We've modified the corresponding content in the revised manuscript.

Line 549:

... if the change in validation AUC over the past 10 epochs is less than 0.01.

17. I am confused concerning our previous exchange concerning the sample numbers in Fig 2a - why would the authors want to only list the PTC cases in the "Hospitals" category, and not the other cases? This is very unusual & confusing. If the authors really want to keep it like this, they should clearly note this in the table and/or its caption.

Response:

We sincerely appreciate the reviewer's comments. In the "hospitals" category, patients who seek care at community hospitals typically due to proximity, often for routine check-ups or early-stage screenings. As a consequence, the diseases observed in this setting are often less severe in nature, leading to a scarcity of rare subtypes which were not incorporated into our experimental design in the 'Hospitals' category.

We've made modifications by adding footnotes in Fig. 2 (a) and Fig. 4 (a) to clearly note this information. For revised Fig. 2 (resp. Fig. 4), we refer to our response to minor issue 3 (resp. major issue 2).

18. It is true that most of the domain adaptation literature considers the unsupervised case. However, the fundamental theorems and incompatibilities also fully apply to the supervised case; see, e.g., <https://arxiv.org/pdf/2305.01397.pdf>

Response:

We appreciate the reviewer's comment and have combined our response for minor issues 18 and 19.

19. Concerning marginal vs. class-conditional invariance: marginal invariance is especially problematic if prevalences differ between domains. Could the authors remark on the prevalence / class balance in their different domains? If this is uniform across domains, that would explain why marginal invariance is not problematic here.

Response:

We sincerely appreciate the reviewer for providing the literature on points raised for minor issue 18 and for offering valuable suggestions in minor issue 19.

In our dataset, we did indeed balance the training set, ensuring that different subtypes had the same proportion of benign and malignant cases. As per your reminder, we also believe that this balancing may have contributed to the effectiveness of marginal invariance.

In the previous manuscript, we unintentionally omitted information about class balance. We have incorporated details about the implementation of class balance in the revised manuscript.

Line 542-543:

Class balance is implemented prior to training session ... ratio among the subgroups.

20. In their response to my previous review, the authors write that "in previous research, combining two [domain invariance] losses with similar objectives has shown the potential for achieving improved results." - could they provide a reference for this statement?

Response:

We apologize for the misunderstanding caused by this sentence. In our previous response, our intention was to convey that in prior research, combining two components with similar objectives has demonstrated the potential for achieving improved results, but the components combined don't necessarily have to be two domain invariance losses. Examples of this practice can be found in statistical techniques such as Elastic Net, as well as deep learning studies like the work by Taghanaki et al. [4]. Previous work motivated us to experiment with combining two domain invariance losses, and our final choice is primarily according to empirical evidence.

21. In supplementary table 2, what does "Training process = Quasi Pareto" (rows 2 and 4) mean? (How) is it different from using the loss in the lowest row of the table? And finally, which of the three "Ours" methods in that table correspond to the proposed QPI method? I suspect it's the final row? This should all be clearly spelled out in the table caption.

Response:

Rows 2 and 4 represent first applying specific sampling strategies on the dataset and then using the proposed QPI approach. These two rows intend to form a comparison to rows 1 and 3.

The only difference lies in the data load strategy. The lowest row corresponds to the standard settings of the proposed QPI method, which did not use special sampling strategies, and maintained the original sample size proportion of majority to minority subgroup in the dataset.

The results reveal that, with the same sampling strategies, the QPI approach outperforms the standard ERM, and employing data sampling strategies does not yield additional performance improvements for the QPI approach. We have explicitly noted this information in the caption of Supplementary Table 2.

Supplementary line 14-22:

Type	Methods			Subgroup (Maj & Min)	AUROC				
	Name	Data load	Training Process		Train	Valid	Test	Test (Maj)	Test (Min)
Simple Group Balancing	I. Balanced – ERM ¹	Upsampling minority groups	ERM	Papillary & Follicular	0.9232	0.7412	0.8672	0.8710	0.7423
				Papillary & Medullary	0.9471	0.8212	0.8512	0.8527	0.7622
				Tertiary & Community	0.9277	0.7923	0.8591	0.8680	0.7283
	Ours	Upsampling minority groups	Quasi Pareto	Papillary & Follicular	0.9819	0.8511	0.8631	0.8721	0.7988
				Papillary & Medullary	0.9125	0.8518	0.8512	0.8621	0.7842
				Tertiary & Community	0.9013	0.8710	0.8628	0.8784	0.7512
	II. SUBG ²	Subsampling majority groups	ERM	Papillary & Follicular	0.9459	0.8343	0.8213	0.8452	0.6709
				Papillary & Medullary	0.9672	0.8500	0.8490	0.8544	0.6862
				Tertiary & Community	0.9362	0.8322	0.8352	0.8465	0.7060
	Ours	Subsampling majority groups	Quasi Pareto	Papillary & Follicular	0.9102	0.8523	0.8484	0.8546	0.7772
				Papillary & Medullary	0.9491	0.8600	0.8520	0.8527	0.7592
				Tertiary & Community	0.9441	0.8662	0.8677	0.8755	0.7101
Minimax Approach	III. Minimax Pareto fairness ³	Common	Minimax Pareto Fair Optimization	Papillary & Follicular	0.9667	0.8392	0.8012	0.8256	0.7931
				Papillary & Medullary	0.9392	0.8101	0.8173	0.8378	0.7693
				Tertiary & Community	0.9102	0.7972	0.8087	0.8182	0.7771
Domain Invariance	IV. Domain-Adversarial Neural Network ⁴	Common	ERM + Maximize domain classifier loss	Papillary & Follicular	0.8906	0.8407	0.8381	0.8493	0.7438
				Papillary & Medullary	0.9321	0.8205	0.8304	0.8349	0.6779
				Tertiary & Community	0.8964	0.8417	0.8333	0.8436	0.7417
Dynamic Weighted Loss	V. Dynamically Weighted Balanced Loss ⁵	Common	ERM + DWS Loss	Papillary & Follicular	0.9432	0.8326	0.8438	0.8624	0.7533
				Papillary & Medullary	0.9121	0.8534	0.8223	0.8552	0.7563
				Tertiary & Community	0.9372	0.8171	0.8241	0.8439	0.7474
Optimized loss functions	VI. Focal Loss ⁶	Common	ERM + Focal Loss	Papillary & Follicular	0.9485	0.8578	0.8532	0.8669	0.7200
				Papillary & Medullary	0.9651	0.8313	0.8309	0.8474	0.6659
				Tertiary & Community	0.9712	0.8434	0.8455	0.8619	0.7111
Ours	Quasi Pareto	Common	$\mathcal{L}_y(\omega_x, \omega_y)$ $-\mathcal{L}_{\text{class conditional}}$ $+ \gamma_{\text{MMD}} \mathcal{L}_{\text{MMD}}(\omega_x)$ $+ \gamma_{\text{BSS}} \mathcal{L}_{\text{BSS}}(\omega_x)$	Papillary & Follicular	0.9504	0.8231	0.8425	0.8531	0.7021
				Papillary & Medullary	0.9231	0.8191	0.8302	0.8462	0.7482
				Tertiary & Community	0.9011	0.8245	0.8110	0.8366	0.7531
		Common	$\mathcal{L}_y(\omega_x, \omega_y)$ $-\gamma_d \mathcal{L}_d(\omega_x, \omega_d)$ $+ \gamma_{\text{MMD}} \mathcal{L}_{\text{MMD}}(\omega_x)$ $+ \gamma_{\text{BSS}} \mathcal{L}_{\text{BSS}}(\omega_x)$	Papillary & Follicular	0.9768	0.8784	0.8672	0.8893	0.8098
				Papillary & Medullary	0.9382	0.8621	0.8572	0.8688	0.7832
				Tertiary & Community	0.9091	0.8621	0.8693	0.8772	0.7794

Supplementary Table 2: Method comparison with state-of-art fairness and domain adaptation approaches. The lowest row (highlighted in gray) represents the standard settings of the proposed QP-Net. Rows 2, 4 (with the Name ‘Ours’) also utilize QP-Net for the training process but incorporate additional data load strategy compared to the standard settings. The results support that existing methods cannot achieve fairness in both majority and minority subgroups without both components (multi-task learning and domain adaptation) of the proposed QPI approach. Specifically, the results in rows 2 and 4 (with the Name ‘Ours’) suggest that subgroup sampling strategies do not yield additional performance improvements for the QP-Net.

Reference

1. Brier, G.W. Verification of forecasts expressed in terms of probability. Monthly weather review 78, 1-3 (1950).
2. Angelini, E.D., et al. Pulmonary emphysema subtypes defined by unsupervised machine learning on CT scans. Thorax (2023).

3. Fan, L., Gong, X. & Guo, Y. General Multiscenario Ultrasound Image Tumor Diagnosis Method Based on Unsupervised Domain Adaptation. *Ultrasound in Medicine & Biology* 49, 2291-2301 (2023).
4. Taghanaki, S.A., et al. Combo loss: Handling input and output imbalance in multi-organ segmentation. *Computerized Medical Imaging and Graphics* 75, 24-33 (2019).

Reviewer #2 (Remarks to the Author):

I thank the authors for the efforts improving the paper. I can appreciate that they have made an effort to address my comments, which they have addressed mostly satisfactorily. I still think some could have been addressed better. Specifics follow below, which I think can be addressed in a revision.

1. To my main concern, about the contextualization of the proposed method beyond Thyroid nodules, authors have replied by adding content in the discussion. This indeed addresses my point however I find it strange to place the text in the discussion. One would expect that, when stating the motivation for the work, at the beginning of the paper, authors describe what the current problem is, why is it not solved, and how current approaches fall short -that is the place to put this review of similar literature.

Response:

We appreciate the reviewer's suggestions and have accordingly restructured our motivation content, introducing it in the order recommended by the reviewer. Additionally, we have incorporated a substantial review of the existing literature within the motivation section. The current introduction is divided into three paragraphs, with the core content as follows:

a. The first paragraph:

The primary challenge in current medical AI revolves around the lack of quantitative evaluation and improvement methods for addressing subgroup unfairness. This issue stems from the relatively low incidence rates in patient subgroups, resulting in insufficient attention, effectively concealing the inadequacies in predicting minority groups beneath overall accuracy.

b. The second paragraph:

Common existing approaches for handling minority issues, including mix training (standard ERM) and divide-and-conquer (stratified ERM), still face unfairness issues. Moreover, most existing methods for mitigating imbalanced samples primarily focus on enhancing the prediction accuracy of minority subgroups, often neglecting the impact on the overall model's predictions. This limitation poses challenges in clinical applications.

c. The third paragraph:

In response to the limitations of existing methods, we have developed the QPI approach. We have tested our method on thyroid data, confirming its effectiveness. Furthermore, we have validated it on two public datasets, and in both cases, our method demonstrates effectiveness, highlighting its generalizability.

The corresponding content and references we modified in the Introduction are listed below:

Line 28-30:

Researchers often evaluate model performance ... or specific subgroups⁹.

Line 30-32:

For instance, in lung cancer research, a typical ... of rare subtypes¹⁰.

Line 32-33:

This oversight is also evident in ultrasound ... consequently, missed detections¹¹.

Line 41-43:

Current medical AI research, however, doesn't ... leave much to be desired^{20,21}.

Line 50-52:

Conversely, many algorithms are designed ... for the overall model's performance^{23,24}.

Line 52-53:

This oversight can lead to complications in real-world clinical applications²⁵.

2. Stating early that authors propose a generic framework then exemplified by thyroid nodules: Related to the main point, in comment 1.1. I suggested to state early on that the study is about thyroid nodules. however, in the revised version, they convey that "AI has made substantial progress as a clinical diagnostic tool for a variety of diseases", to then follow on with examples of thyroid nodules work (without describing that this is the focus, or arguing whether thyroid nodules would be representative of a wider range of disease), and following from those examples they infer that " Given sample size imbalance, AI models might be subject to unfairness by omitting features of the minority subgroups" which is a broad statement, generic, and not thyroid nodule specific. The same trend follows in the rest of the motivation section. In brief, they try to motivate and defend their work as a generic framework but all evidence is pulled from thyroid nodule works, and that does not hold/generalize. Please, either broaden the focus and build your case from a wider range of applications, or really formulate your proposal as a thyroid nodule work.

Response:

We sincerely appreciate the reviewer's comments regarding this matter. We have taken the reviewer's suggestions into careful consideration and have restructured our manuscript to align with our primary contribution: introducing a generalizable approach to address medical AI unfairness issue for imbalanced subgroups. From our perspective, this has been achieved through the following modifications and considerations:

Firstly, we have reorganized the introduction and incorporated a substantial amount of literature related to more general medical AI fairness topics beyond thyroid cancer, as mentioned in our response to the reviewer's issue 1.

Secondly, we introduced a universal 'Pareto fairness criterion' to address fairness issues, and specifically in this revision, we have added context regarding our proposed fairness criterion within the existing literature in the method section. Our 'Pareto fairness criterion' is an innovation, distinct from the current medical fairness literature, as it approaches fairness from a more causality-based perspective. Through learning curve experiments, it can reveal unfairness phenomena that are not intuitively observable using traditional fairness concepts.

Thirdly, our experimental results include two public datasets outside of thyroid cancer. We selected these two public datasets for validation because they encompass diseases and medical image types that are significantly different from our thyroid ultrasound data. Supported by empirical results provided in Fig. 6 and Fig. 7, we believe that these results can demonstrate the generalizability of our method.

We would greatly appreciate the reviewer's guidance if further evidence is required to establish that our work is intended for general applications in medical AI, and we sincerely look forward to the reviewer's feedback.

We list our modifications regarding the contextualization of our notation of fairness in extant literature as follows:

Line 378:

The performance disparity is often used to indicate model unfairness⁴⁹⁻⁵¹.

Line 389-390:

Performance disparity ... definitions of Pareto fairness⁵⁰.

Line 390-392:

Most studies on Pareto fairness ... detriment to others⁵²⁻⁵⁴.

Line 392-395:

Nonetheless, there are phenomena ... decision-making algorithms^{52,55,56}.

Line 398-399:

However, few studies approach this fairness issue from a causal perspective.

Line 402-403:

which is a causality-based fairness ... in literature.

Line 440-441:

Previous investigations into Pareto improvement ... while benefiting others^{52,54,57,58}.

Line 441-444:

Our QPI approach doesn't rigidly constrain ... increases in sample size.

In general lines, authors have addressed all my other comments.

Reviewers' Comments:

Reviewer #1:

Remarks to the Author:

I would like to thank the authors, again, for their very careful and thorough re-revision! All of my previous concerns have been addressed and I have no further comments. I consider the revised manuscript to be in excellent shape for publication.

Reviewer #2:

Remarks to the Author:

Authors have addressed my comments satisfactorily in the body of the manuscript.

I only have two minor concerns.

1. The abstract, which has been left untouched since the last review, is now outdated, and misses the new focus of the paper. The paper is now about unfairness, and a new method to address it, where the efficacy is proving by using thyroid images. the abstract reads as if the paper is about thyroid images.

This can be simply addressed, i think. For example, before the sentence starting with "We collected ...", they could place the paragraph starting with "To address this ..." which is found later on in the abstract.

Basically I am suggesting that the abstract first explain what is the problem, in general; second, what is the generic solution proposed; and third, how this solution was assessed with specific datasets.

2. The addition of the ISIC2019 and the Chexpert dataset is very welcome, however through the paper it reads as if most of the development is done with the thyroid dataset and then only some validation is done with the other two. The problem with this is that is is unclear what the purpose of the other two is. Perhaps authors could empasize that the two datasets are not for validation (as is indicated now) but rather to prove that the approach is generic? I belive this can be done with a few small changes through the paper

Reviewer #1 (Remarks to the Author):

I would like to thank the authors, again, for their very careful and thorough re-revision! All of my previous concerns have been addressed and I have no further comments. I consider the revised manuscript to be in excellent shape for publication.

Response:

We sincerely appreciate the reviewer's approval of our work.

Reviewer #2 (Remarks to the Author):

Authors have addressed my comments satisfactorily in the body of the manuscript.

I only have two minor concerns.

1. The abstract, which has been left untouched since the last review, is now outdated, and misses the new focus of the paper. The paper is now about unfairness, and a new method to address it, where the efficacy is proving by using thyroid images. the abstract reads as if the paper is about thyroid images.

This can be simply addressed, i think. For example, before the sentence starting with "We collected ...", they could place the paragraph starting with "To address this ..." which is found later on in the abstract.

Basically I am suggesting that the abstract first explain what is the problem, in general; second, what is the generic solution proposed; and third, how this solution was assessed with specific datasets.

Response:

We sincerely appreciate the reviewer's suggestions. In revised manuscript, we've modified the abstract according to the reviewer's suggestions and also the formatting instructions. The revised abstract follows exactly the structure that the reviewer suggests and meets the formatting requirements.

Line 2-12:

Artificial Intelligence (AI) models ... for equitable healthcare outcomes.

2. The addition of the ISIC2019 and the Chexpert dataset is very welcome, however through the paper it reads as if most of the development is done with the thyroid dataset and then only some validation is done with the other two. The problem with this is that is unclear what the purpose of the other two is. Perhaps authors could emphasize that the two datasets are not for validation (as is indicated now) but rather to prove that the approach is generic? I believe this can be done with a few small changes through the paper.

Response:

We sincerely thank the reviewer for the comments on this problem. In revised manuscript, we've emphasized this purpose in abstract and introduction.

Line 62:

... evaluated ...

Line 64:

... , to support the generalizability of our QPI approach.